# Trust Region Reward Optimization and Proximal Inverse Reward Optimization Algorithm [*]

**Yang Chen**[1][†][‡]   **Menglin Zou**[2][†]   **Jiaqi Zhang**[3]   **Yitan Zhang**[2]   **Junyi Yang**[2]
**Gaël Gendron**[2]   **Libo Zhang**[2]   **Jiamou Liu**[2]   **Michael J. Witbrock**[2]
[1] Shanghai Artificial Intelligence Laboratory   [2] University of Auckland   [3] Chongqing University
chenyang4@pjlab.org.cn

## Abstract

Inverse Reinforcement Learning (IRL) learns a reward function to explain expert demonstrations. Modern IRL methods often use the adversarial (minimax) formulation that alternates between reward and policy optimization, which often lead to *unstable* training. Recent non-adversarial IRL approaches improve stability by jointly learning reward and policy via energy-based formulations but lack formal guarantees. This work bridges this gap. We first present a *unified* view showing canonical non-adversarial methods explicitly or implicitly maximize the likelihood of expert behavior, which is equivalent to minimizing the expected return gap. This insight leads to our main contribution: *Trust Region Reward Optimization* (TRRO), a framework that guarantees *monotonic* improvement in this likelihood via a Minorization-Maximization process. We instantiate TRRO into *Proximal Inverse Reward Optimization* (PIRO), a practical and stable IRL algorithm. Theoretically, TRRO provides the IRL counterpart to the stability guarantees of Trust Region Policy Optimization (TRPO) in forward RL. Empirically, PIRO matches or surpasses state-of-the-art baselines in reward recovery, policy imitation with high sample efficiency on MuJoCo and Gym-Robotics benchmarks and a real-world animal behavior modeling task. [1]

## 1 Introduction

Learning optimal policies from fixed reward functions is reinforcement learning (RL); learning rewards from fixed expert policies is inverse reinforcement learning (IRL) [28]. Modern IRL methods [12, 40, 33] often take a minimax game formulation and a bi-level optimization procedure, where a reward function (min player) is adversarially optimized to differentiate between a best-response policy (max player, an RL subroutine) and the expert policy via their expected return gap (a.k.a. the *imitation gap* [39]). Due to the advantages of interpretability, robustness to dynamics shifts [1], and out-of-distribution generalization [6], these methods have been effectively applied in autonomous driving [21], robotics [7], and reward modeling in language models [38]. However, despite its theoretical grounding and practical appeal, adversarial training introduces optimization instability due to brittle approximations and high sensitivity to hyperparameters, hindering reliable reward recovery.

---

[*]Title used at submission and review: PIRO: Toward Stable Reward Learning for Inverse RL via Monotonic Policy Divergence Reduction.

[†]Main contributors. Yang Chen developed the theorems, completed the proofs, wrote the paper, and implemented the initial version of the algorithm. Menglin Zou led the experimental evaluation. Jiaqi Zhang and Junyi Yang validated the algorithm using toy models. Yitan Zhang conducted the experiments on robotics and animal behavior modeling tasks. The remaining authors contributed through critical discussions and feedback.

[‡]Corresponding author.

[1]The implementation is available at https://github.com/PolynomialTime/PIRO.

39th Conference on Neural Information Processing Systems (NeurIPS 2025).

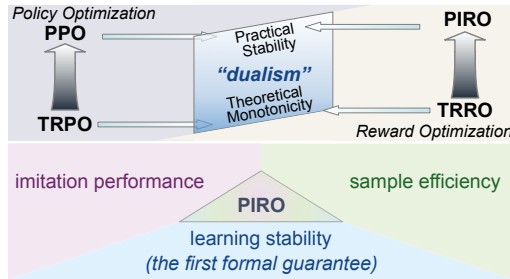

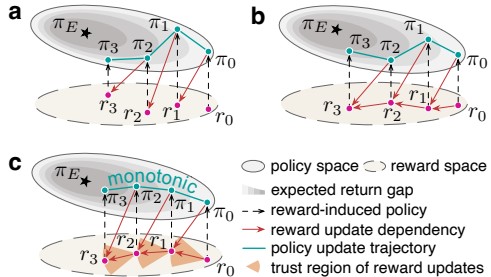

Figure 1: **Theoretical (top) and practical (bottom) contributions. Top:** PPO – rooted in TRPO's theory of monotonic policy improvement – has been (one of) the most successful RL algorithm(s). This work is motivated by a ***dualism:*** the mathematical beauty of TRPO should *not* exist in *isolation*, but in *conjugation* with its inverse problem space. We identify and formalize this inverse counterpart, completing the "right half" of this "symmetric picture". We believe this contribution advances RL theory and opens new avenues for designing robust IRL algorithms. See Sec. 4 for theoretical justifications. **Bottom:** PIRO, our practical algorithm, achieves a three-way balance among learning stability, imitation performance, and sample efficiency. To our knowledge, PIRO is the first IRL method that achieves state-of-the-art performance in imitation performance and learning stability with high sample efficiency. See Sec. 5 for the practical algorithm design and Sec. 7 for experiments.

Figure 2: **Comparing Adversarial IRL, Non-adversarial IRL and our Trust Region Reward Optimization (TRRO).** **(a)** Adversarial IRL methods frame reward learning as a game against a (nearly) best-response policy, often resulting in unstable training dynamics due to the inherent minimax structure. **(b)** Non-adversarial IRL methods bypass this game setup by coupling reward and policy via energy-based formulations and jointly update them by minimizing the expected return gap (a.k.a. the imitation gap). However, lacking principled control over reward update makes them sensitive to optimization errors. **(c)** TRRO reformulates non-adversarial IRL as a majorization-minimization (MM) process that identifies a trusted reward update in each step. This ensures a monotonic reduction in imitation gap and providing, to our knowledge, the first formal stability guarantee in IRL. (**Note:** This is a theoretical comparison assuming exact policy computation.)

Recent non-adversarial IRL approaches [32, 15, 30, 50, 51, 44] revive a line of early apprenticeship learning methods [27, 31]; they bypass the nested adversarial training by coupling the reward and policy via an energy-based model [17], jointly updating them to optimize some measure of fit to expert behavior. While improving empirical stability, they still lack principled control over reward updates. As a result, *a provably stable IRL mechanism, one that ensures consistent progress toward expert imitation, remains elusive*. This work aims to address this **gap**.

By leveraging the fact that the expected return gap between two policies equals the expected advantage value of one under the other [35, 25, 50], we develop a **unified view** of canonical non-adversarial IRL methods. We show that *they all, explicitly or implicitly, optimize the likelihood of expert behavior, which is equivalent to minimizing the imitation gap* (Sec. 3). This leads to our **key insight:** *IRL stability can be achieved by provably increasing the likelihood of expert demonstrations at every update step.* We realize this insight in a principled non-adversarial IRL framework and a practical algorithm that together offer a stable alternative to existing approaches.

Concretely, our contributions are summarized as follows, which are illustrated in Fig. 1:

• We propose **Trust Region Reward Optimization** (TRRO), a principled non-adversarial IRL framework that, to our knowledge, for the *first* time provides a formal guarantee on stability. As depicted in Fig. 2, it provides principled control on reward update via a Minorization-Maximization (MM) process, which iteratively optimizes a surrogate objective function to identify a trusted reward update that ensures a *monotonic* improvement in the likelihood of expert behavior (equivalent to reducing the imitation gap). (Sec. 4)

• We develop **Proximal Inverse Reward Optimization** (PIRO), a practical IRL algorithm that approximates the theoretical guarantee of TRRO through adaptive step sizes in place of the theory-informed small updates. PIRO achieves a balance among learning stability, imitation performance and sample efficiency. It can be easily implemented on top of Soft Actor-Critic [17] by adding a few stochastic gradient steps for the controlled reward update. (Sec. 5)

• We empirically demonstrate the strong performance of PIRO. Across MuJoCo and Gym Robotics tasks, PIRO offers substantially improved stability and high sample efficiency, while matches or exceeds state-of-the-art IRL methods in reward recovery and policy imitation. (Sec. 7)

TRRO/PIRO mirrors the success of Trust Region Policy Optimization (TRPO) [35] and its successor Proximal Policy Optimization (PPO) [36]. TRPO guarantees monotonic policy improvement in expected return with respect to a fixed reward function, while TRRO ensures monotonic reduction in the expected return gap with respect to the expert behavior. In this sense, TRRO/PIRO serves as the inverse RL counterpart to TRPO/PPO in forward RL.

## 2 Preliminaries

Consider a Markov decision process (MDP) defined by $(\mathcal{S}, \mathcal{A}, r, \eta, P, \gamma)$, where $\mathcal{S}$ and $\mathcal{A}$ are the state and action spaces, $\eta(\cdot)$ is the initial state distribution, $P : \mathcal{S} \times \mathcal{A} \times \mathcal{S} \to [0, 1]$ is the transition function, $r : \mathcal{S} \times \mathcal{A} \to \mathbb{R}$ is the reward function, and $\gamma \in (0, 1)$ is the discount factor. A stochastic policy $\pi : \mathcal{S} \times \mathcal{A} \to [0, 1]$ defines a probabilistic action selection at each state. We denote the *occupancy measure* of $\pi$ as $\rho^\pi(\mathbf{s}, \mathbf{a}) := \sum_{t=0}^\infty \gamma^t \Pr(\mathbf{s}_t = \mathbf{s}, \mathbf{a}_t = \mathbf{a}|\mathbf{s}_0 \sim \eta, \pi, P)$. Note that we will omit the normalizing constant $\frac{1}{1-\gamma}$ for $\rho^\pi(\mathbf{s}, \mathbf{a})$.

### 2.1 Maximum Entropy RL

MaxEnt RL characterizes the optimal behavior as a policy $\pi^*$ that maximizes the *policy entropy-augmented rewards*:

$$J(\pi, r) := \mathbb{E}_{\rho^\pi}[r(\mathbf{s}, \mathbf{a})] + \mathcal{H}(\pi), \ \mathcal{H}(\pi) := \mathbb{E}_{\rho^\pi}[-\log(\pi(\mathbf{a}|\mathbf{s}))]. \qquad \text{(MaxEnt-RL)}$$

Here, $\mathcal{H}(\pi)$ is the discounted *causal entropy* [53] of a policy $\pi$. In MaxEnt RL, an optimal policy $\pi^*$ follows an *energy-based model*:

$$\pi^*(\mathbf{a}|\mathbf{s}) = \exp(Q_r^{\pi^*}(\mathbf{s}, \mathbf{a}) - V_r^{\pi^*}(\mathbf{s})), \qquad (1)$$

where $Q_r^{\pi^*}$ is the optimal soft Q-function and $V_r^{\pi^*}$ is the optimal soft value function satisfying:

$$V_r^{\pi^*}(\mathbf{s}) = \log \sum_{\mathbf{a} \in \mathcal{A}} \exp(Q_r^{\pi^*}(\mathbf{s}, \mathbf{a})), \ Q_r^{\pi^*}(\mathbf{s}, \mathbf{a}) = r(\mathbf{s}, \mathbf{a}) + \gamma \mathbb{E}_{\mathbf{s}' \sim P(\cdot|\mathbf{s}, \mathbf{a})}[V_r^{\pi^*}(\mathbf{s}')]. \qquad (2)$$

Eq. (2) is the so-called *Soft Bellman Equation*. Given a reward function $r \in \mathcal{R} \subset \mathbb{R}^{\mathcal{S} \times \mathcal{A}}$ and a policy $\pi \in \Pi \subset [0, 1]^{\mathcal{S} \times \mathcal{A}}$, the soft Q-value can be computed by iteratively applying the *soft Bellman operator* $\mathcal{B}_r^\pi : \mathbb{R}^{\mathcal{S} \times \mathcal{A}} \to \mathbb{R}^{\mathcal{S} \times \mathcal{A}}$ defined as:

$$(\mathcal{B}_r^\pi Q)(\mathbf{s}, \mathbf{a}) = r(\mathbf{s}, \mathbf{a}) + \gamma \mathbb{E}_{\mathbf{s}' \sim P(\cdot|\mathbf{s}, \mathbf{a})}[V(\mathbf{s}')], \ V(\mathbf{s}) = \mathbb{E}_{\mathbf{a} \sim \pi(\cdot|\mathbf{s})}[Q(\mathbf{s}, \mathbf{a}) - \log \pi(\mathbf{a}|\mathbf{s})]. \qquad (3)$$

The operator $\mathcal{B}_r^\pi$ is contractive [18] and defines the soft Q-function $Q_r^\pi$ as a unique fixed point solution, i.e. $Q_r^\pi = \mathcal{B}_r^\pi Q_r^\pi$. An improved policy can be derived from $Q_r^\pi$ through

$$\pi'(\mathbf{a}|\mathbf{s}) \propto \exp(Q_r^\pi(\mathbf{a}, \mathbf{s})), \qquad (4)$$

which guarantees $Q_r^{\pi'}(\mathbf{a}|\mathbf{s}) \geq Q_r^\pi(\mathbf{a}|\mathbf{s})$ for all $(\mathbf{s}, \mathbf{a}) \in \mathcal{S} \times \mathcal{A}$. Starting from an arbitray policy $\pi$, repeated application of Eq. (3) and Eq. (4) gives the so-called *soft policy iteration* [18], which converges to the optimal policy $\pi^*$ that maximizes $J(\pi, r)$ in (MaxEnt-RL).

### 2.2 Maximum Entropy IRL

Suppose we do not know the reward function but have a set of demonstrations $\mathcal{D}_E = \{(\mathbf{s}_0, \mathbf{a}_0, \ldots)\}$ sampled from an expert policy $\pi_E$. MaxEnt IRL aims to recover the reward function that explains demonstrations by minimizing the expected return gap (a.k.a. *imitation gap* [39]) through solving the following optimization problem: [2]

$$\min_{\pi \in \Pi} \max_{r \in \mathcal{R}} J(\pi_E, r) - J(\pi, r) = \mathbb{E}_{\rho^{\pi_E}}[r(\mathbf{s}, \mathbf{a})] - (\mathbb{E}_{\rho^\pi}[r(\mathbf{s}, \mathbf{a})] + \mathcal{H}(\pi)). \qquad \text{(MaxEnt-IRL)}$$

In practice, $\mathbb{E}_{\rho^{\pi_E}}[r(\mathbf{s}, \mathbf{a})]$ is emprically estimated on expert demonstrations $\mathcal{D}_E$. The minimax formulation of (MaxEnt-IRL) suggests an adversarial solution structure: [3] an *outer loop* optimizes the reward function by differentiating expert and learned policies through maximizing the imitation gap (Line 4, Alg. 1) and an *inner loop* trains an optimal policy via a MaxEnt RL process (Line 3, Alg. 1). MaxEnt IRL has been well studied theoretically [53, 4] and has been practically applied [45, 13]. However, its nested structure can introduce significant training instability and computational burden, especially when state-action spaces are high-dimensional or continuous.

---

[2] We hereafter omit the constant expert policy entropy $\mathcal{H}(\pi_E)$ in $J(\pi_E, r)$.

[3] See Sec. 6 for the discussion on adversarial IRL methods.

| **Algorithm 1** Adversarial IRL | **Algorithm 2** Non-Adversarial IRL |
|---|---|
| 1: **Provided:** Expert demonstration $\mathcal{D}_E$, Reward parameter $\boldsymbol{\theta}_0$.
2: **for** $i$ in $1, \ldots, N$ **do**
   // A full RL process
3:   $\pi_i \leftarrow \texttt{MaxEntRL}(r_{\boldsymbol{\theta}_{i-1}})$.
4:   $\boldsymbol{\theta}_i \leftarrow \arg\max_{\boldsymbol{\theta}} J(\pi_E, r_{\boldsymbol{\theta}}) - J(\pi_i, r_{\boldsymbol{\theta}})$.
5: **end for** | 1: **Provided:** Expert demonstration $\mathcal{D}_E$, Reward parameter $\boldsymbol{\theta}_0$, Policy $\pi_0$.
2: **for** $i$ in $1, \ldots, N$ **do**
   // One round of soft policy iteration.
3:   $\pi_i(\mathbf{a}|\mathbf{s}) \propto \exp(Q_{r_{\boldsymbol{\theta}_{i-1}}}^{\pi_{i-1}}(\mathbf{s}, \mathbf{a}))$.
4:   $\boldsymbol{\theta}_i \leftarrow \boldsymbol{\theta}_{i-1} + \alpha_i \nabla_{\boldsymbol{\theta}}(J(\pi_E, r_{\boldsymbol{\theta}}) - J(\pi_i, r_{\boldsymbol{\theta}}))$.
5: **end for** |

## 2.3 Maximum Likelihood IRL

ML-IRL bypasses the nested loop in MaxEnt IRL by jointly updating the reward and policy via the energy-based model (Eq. (1)), thereby improving stability. Let $\pi_{\boldsymbol{\theta}}$ denote the optimal policy induced by a $\boldsymbol{\theta}$-parameterized reward function $r_{\boldsymbol{\theta}}$ with $\boldsymbol{\theta} \in \mathbb{R}^d$. ML-IRL aims to maximize the likelihood of expert behavior under $\pi_{\boldsymbol{\theta}}$ (equivalent to minimizing the KL divergence $D_{\mathrm{KL}}(\pi_E(\mathbf{a}|\mathbf{s})\|\pi_{\boldsymbol{\theta}}(\mathbf{a}|\mathbf{s})) := \mathbb{E}_{\rho^{\pi_E}}[\log \pi_E(\mathbf{a}|\mathbf{s}) - \log \pi_{\boldsymbol{\theta}}(\mathbf{a}|\mathbf{s})]$):

$$\max_{\boldsymbol{\theta}} \ell(\boldsymbol{\theta}) := \mathbb{E}_{\rho^{\pi_E}}[\log \pi_{\boldsymbol{\theta}}(\mathbf{a}|\mathbf{s})]. \tag{ML-IRL}$$

An important property of $\ell(\boldsymbol{\theta})$ is that it can be equivalently expressed as the imitation gap. [4]

**Proposition 1** (Lemma 1 in [50]). *The log-likelihood objective $\ell(\boldsymbol{\theta})$ in* (ML-IRL) *has the following equivalent form that implies the expression of its gradient:*

$$\ell(\boldsymbol{\theta}) = \mathbb{E}_{\rho^{\pi_E}}[r_{\boldsymbol{\theta}}(\mathbf{s}, \mathbf{a})] - \mathbb{E}_{\mathbf{s}_0 \sim \eta}[V_{r_{\boldsymbol{\theta}}}^{\pi_{\boldsymbol{\theta}}}(\mathbf{s}_0)] = J(\pi_E, r_{\boldsymbol{\theta}}) - J(\pi_{\boldsymbol{\theta}}, r_{\boldsymbol{\theta}}), \tag{5a}$$

$$\nabla_{\boldsymbol{\theta}} \ell(\boldsymbol{\theta}) = \mathbb{E}_{\rho^{\pi_E}}[\nabla_{\boldsymbol{\theta}} r_{\boldsymbol{\theta}}(\mathbf{s}, \mathbf{a})] - \mathbb{E}_{\rho^{\pi_{\boldsymbol{\theta}}}}[\nabla_{\boldsymbol{\theta}} r_{\boldsymbol{\theta}}(\mathbf{s}, \mathbf{a})]. \tag{6a}$$

Indeed, Proposition 1 is not so surprising, as it reflects a standard identity in RL theory: the expected return gap between two policies equals the expected advantage value ($Q(\mathbf{s}, \mathbf{a}) - V(\mathbf{s})$) of one policy under the occupancy measure of the other [23, 35, 25]; in MaxEnt RL, the advantage value corresponds to $\log \pi$ (see Eq. (1)). However, its implication for MF-IRL is noteworthy: it effectively bypasses the inner RL loop typically required in MaxEnt IRL. As a result, the nested-loop optimization is reduced to a *single-loop* structure: alternating between one round of soft policy iteration for policy improvement (Line 3, Alg. 2) and one gradient step for reward update (Line 4, Alg. 2).

To further mitigates instability, [50] employ a decaying gradient step size $\alpha_i = \frac{\alpha_0}{N^{\sigma}}$ for reward updates, where $N$ is the total number of iterations and $\sigma \in (0, 1)$ is a constant. Under the assumption of *exact policy computation* for $\pi_i$, [50, Theorem 2] show that with Alg. 2, $\ell(\boldsymbol{\theta})$ converges at rate $\mathcal{O}(N^{-1}) + \mathcal{O}(N^{-\sigma})$, and converges to the optimal value under linear reward functions. However, this setup still lacks a formal stability guarantee, as gradient-based reward updates with heuristic step sizes cannot ensure improvement in $\ell(\boldsymbol{\theta})$ at each step. Our key contribution fills this gap: a novel non-adversarial IRL framework that, under the similar assumption of exact policy computation, guarantees *monotonic* improvement in $\ell(\boldsymbol{\theta})$ through a carefully designed non-gradient reward update mechanism (Sec. 4).

## 3 A Unified View of Non-Adversarial IRL: IR, ER and Beyond

In this section, we show an interesting yet natural fact that a range of canonical non-adversarial IRL methods — both *implicit reward* (**IR**) methods that learn soft Q-functions (e.g., Soft Q Imitation Learning (SQIL) [32], Inverse Q Learning (IQ-Learn) [15]) and *explicit reward* (**ER**) methods that directly learn reward functions (e.g., $f$-IRL [30] and ML-IRL) — can be unified under the objective of maximizing the likelihood of expert behavior. As discussed further in Sec. 6, this unified view extends to a broader class of non-adversarial IRL methods that go beyond the settings of these canonical methods. This allows for unifying non-adversarial IRL methods under a general optimization procedure (Alg. 2), highlighting the generality of maximizing the likelihood as a principled objective and situates our framework (next section) within a broader methodological landscape.

---

[4]We provide the proof of Proposition 1 in Appendix A.1 using the notations in this paper.

For IR, we already know that the objectives of SQIL and IQ-Learn are regularized versions of [5]

$$\ell_Q(\boldsymbol{\omega}) := \mathbb{E}_{\rho^{\pi_E}}[r_{Q_{\boldsymbol{\omega}}}(\mathbf{s}, \mathbf{a})] - \mathbb{E}_{\mathbf{s}_0 \sim \eta}[V^*(\mathbf{s}_0)], \tag{5b}$$

where $V^*(\mathbf{s}) = \log \sum_{\mathbf{a} \in \mathcal{A}} \exp(Q_{\boldsymbol{\omega}}(\mathbf{s}, \mathbf{a}))$. Eq. (5b) can be derived by transforming $\ell(\boldsymbol{\theta})$ (Eq. (5a)) via replacing $r_{\boldsymbol{\theta}}$ with $r_{Q_{\boldsymbol{\omega}}}(\mathbf{s}, \mathbf{a}) := Q_{\boldsymbol{\omega}}(\mathbf{s}, \mathbf{a}) - \gamma \mathbb{E}_{\mathbf{s}' \sim P(\cdot | \mathbf{s}, \mathbf{a})}[V^*(\mathbf{s}')]$ – the implicit reward defined as the differences of $\boldsymbol{\omega}$-parameterized soft Q-values via the soft Bellman equation (Eq. (2)).

For ER, we show that the objective of a basic form of $f$-IRL — assuming state-only rewards and minimizing the KL divergence between expert and learner state marginals — is equivalent to $\ell(\boldsymbol{\theta})$, up to a constant. That is (proof of Eq. (6b) in Appendix B),

$$r_{\boldsymbol{\theta}}(\mathbf{s}) \implies \nabla_{\boldsymbol{\theta}} D_{\mathrm{KL}}(\rho^{\pi_E}(\mathbf{s}) \| \rho^{\pi_{\boldsymbol{\theta}}}(\mathbf{s})) \propto -\nabla_{\boldsymbol{\theta}} \ell(\boldsymbol{\theta}), \tag{6b}$$

where $\rho^{\pi}(\mathbf{s}) = \rho^{\pi}(\mathbf{s}, \mathbf{a})/\pi(\mathbf{a}|\mathbf{s})$ denotes the state marginal of the occupancy measure.

Pros and cons of IR/ER methods are well-documented [33]. IR offers higher computational efficiency, as Eq. (5b) depends *solely* on estimating the soft Q-function, which encodes both reward and policy. However, this coupling of reward and environment dynamics can lead to inaccuracies under dynamics shift, thereby limiting the reward transferability to new dynamics. In contrast, ER methods learn reward functions directly and avoid this entanglement, offering better robustness to dynamics shift. In light of this, our framework will adopt the ER formulation.

## 4   Trust Region Reward Optimization

In this section, we introduce *Trust Region Reward Optimization* (TRRO), a theoretical IRL framework that enforces stability by producing a guaranteed increase on the likelihood of expert behavior. To our knowledge, it provides the *first* formal theoretical stability guarantee for IRL.

To proceed, let $\boldsymbol{\theta}_{\mathrm{old}}$ denote the current reward parameter and assume we have the corresponding optimal policy $\pi_{\mathrm{old}}$. As argued in Sec. 2.3, gradient-based reward updates cannot rigorously ensure an improvement in $\ell(\boldsymbol{\theta})$. We thus consider a non-gradient-based approach. Our key idea is to restrict the search for $\boldsymbol{\theta}_{\mathrm{new}}$ within a region centered around $\boldsymbol{\theta}_{\mathrm{old}}$ such that all $\boldsymbol{\theta}$ in that region admit an increase on $\ell(\boldsymbol{\theta})$. To do so, we introduce the following local approximation to $\ell(\boldsymbol{\theta})$:

$$\ell_{\boldsymbol{\theta}_{\mathrm{old}}}(\boldsymbol{\theta}) := \mathbb{E}_{\rho^{\pi_E}}[r_{\boldsymbol{\theta}}(\mathbf{s}, \mathbf{a})] - \mathbb{E}_{\mathbf{s}_0 \sim \eta}[V_{r_{\boldsymbol{\theta}}}^{\pi_{\mathrm{old}}}(\mathbf{s}_0)] = J(\pi_E, r_{\boldsymbol{\theta}}) - J(\pi_{\mathrm{old}}, r_{\boldsymbol{\theta}}). \tag{5c}$$

**Proposition 2.** *Suppose $r_{\boldsymbol{\theta}}$ is differentiable. The surrogate function $\ell_{\boldsymbol{\theta}_{\mathrm{old}}}(\boldsymbol{\theta})$ in Eq. (5c) matches the original objective $\ell(\boldsymbol{\theta})$ in Eq. (5a) to first order, i.e., for any value $\boldsymbol{\theta}_{\mathrm{old}}$:*

$$\underbrace{\ell_{\boldsymbol{\theta}_{\mathrm{old}}}(\boldsymbol{\theta}_{\mathrm{old}}) = \ell(\boldsymbol{\theta}_{\mathrm{old}})}_{\equiv \mathbb{E}_{\rho^{\pi_E}}[r_{\boldsymbol{\theta}_{\mathrm{old}}}(\mathbf{s},\mathbf{a})] - \mathbb{E}_{\mathbf{s}_0 \sim \eta}[V_{r_{\boldsymbol{\theta}_{\mathrm{old}}}}^{\pi_{\mathrm{old}}}(\mathbf{s}_0)]} \quad and \quad \underbrace{\nabla_{\boldsymbol{\theta}} \ell_{\boldsymbol{\theta}_{\mathrm{old}}}(\boldsymbol{\theta})|_{\boldsymbol{\theta}=\boldsymbol{\theta}_{\mathrm{old}}} = \nabla_{\boldsymbol{\theta}} \ell(\boldsymbol{\theta})|_{\boldsymbol{\theta}=\boldsymbol{\theta}_{\mathrm{old}}}}_{\equiv \mathbb{E}_{\rho^{\pi_E}}[\nabla_{\boldsymbol{\theta}} r_{\boldsymbol{\theta}}(\mathbf{s},\mathbf{a})] - \mathbb{E}_{\rho^{\pi_{\mathrm{old}}}}[\nabla_{\boldsymbol{\theta}} r_{\boldsymbol{\theta}}(\mathbf{s},\mathbf{a})]|_{\boldsymbol{\theta}=\boldsymbol{\theta}_{\mathrm{old}}}} \quad . \tag{6c}$$

*Proof.* See annotated equivalence relationships above. □

Proposition 2 implies that a sufficiently small step $\boldsymbol{\theta}_{\mathrm{old}} \to \boldsymbol{\theta}_{\mathrm{new}}$, which increases $\ell_{\boldsymbol{\theta}_{\mathrm{old}}}(\boldsymbol{\theta})$, will also increases $\ell(\boldsymbol{\theta})$. However, it still does not provide guidance on the suitable step size for this update. Our theorem below addresses this by deriving an explicit lower bound on $\ell(\boldsymbol{\theta}_{\mathrm{new}})$ in terms of $\ell_{\boldsymbol{\theta}_{\mathrm{old}}}(\boldsymbol{\theta}_{\mathrm{new}})$ and the difference between $r_{\boldsymbol{\theta}_{\mathrm{old}}}$ and $r_{\boldsymbol{\theta}_{\mathrm{new}}}$.

**Theorem 3.** *Let $\epsilon_{\boldsymbol{\theta}_{\mathrm{old}}}(\boldsymbol{\theta}_{\mathrm{new}}) := \max_{\mathbf{s},\mathbf{a}} |r_{\boldsymbol{\theta}_{\mathrm{new}}}(\mathbf{s}, \mathbf{a}) - r_{\boldsymbol{\theta}_{\mathrm{old}}}(\mathbf{s}, \mathbf{a})|$. Assume $|\mathcal{A}| < \infty$ and $|r_{\boldsymbol{\theta}_{\mathrm{new}}}(\mathbf{s}, \mathbf{a})| \leq R, \forall \mathbf{s} \in \mathcal{S}, \mathbf{a} \in \mathcal{A}$. Then, the following inequality holds:*

$$\ell(\boldsymbol{\theta}_{\mathrm{new}}) \geq \ell_{\boldsymbol{\theta}_{\mathrm{old}}}(\boldsymbol{\theta}_{\mathrm{new}}) - C\epsilon_{\boldsymbol{\theta}_{\mathrm{old}}}(\boldsymbol{\theta}_{\mathrm{new}}), \text{ where the constant}$$
$$C = \frac{2|\mathcal{A}|}{(1-\gamma)^2} + \frac{(5-\gamma)|\mathcal{A}|R + (\gamma - \gamma^2 + 2)|\mathcal{A}| \log |\mathcal{A}|}{(1-\gamma)^4}. \tag{7}$$

*Proof.* In Appendix A.2. □

---

[5] See [32, Sec. 3.3] for SQIL and [15, Sec. 4] for IQ-Learn.

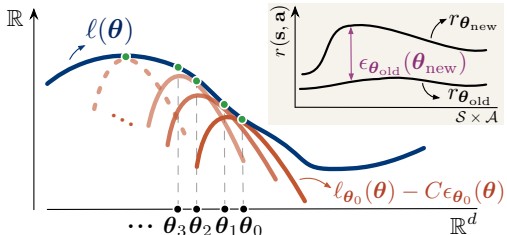

Figure 3: **Illustration of the mechanism of Trust Region Reward Optimization (TRRO).** The reward optimization follows a Minorization-Maximization process, iteratively optimizing a surrogate function that minorizes the original likelihood objective, thereby guaranteeing monotonic improvement in the likelihood of expert demonstrations (assuming exact policy optimization).

Since $\epsilon_{\boldsymbol{\theta}_{\text{old}}}(\boldsymbol{\theta}_{\text{old}}) = 0$, by continuity, there exists a $\boldsymbol{\theta}_{\text{new}}$ in the neighborhood of $\boldsymbol{\theta}_{\text{old}}$ such that $\ell(\boldsymbol{\theta}_{\text{new}}) \geq \ell_{\boldsymbol{\theta}_{\text{old}}}(\boldsymbol{\theta}_{\text{new}}) - C\epsilon_{\boldsymbol{\theta}_{\text{old}}}(\boldsymbol{\theta}_{\text{new}})$. This implies that maximizing the lower bound in Theorem 3 guarantees an increase (or at least no decrease) on $\ell(\boldsymbol{\theta})$, which leads to the following procedure that alternates between policy and reward update:

$$\pi = \arg\max_{\pi} J(\pi, r_{\boldsymbol{\theta}_{\text{old}}}) \overset{\pi \to \pi_{\text{old}}}{\underset{\boldsymbol{\theta}_{\text{old}} \leftarrow \boldsymbol{\theta}_{\text{new}}}{\rightleftharpoons}} \boldsymbol{\theta}_{\text{new}} = \arg\max_{\boldsymbol{\theta}} \ell_{\boldsymbol{\theta}_{\text{old}}}(\boldsymbol{\theta}) - C\epsilon_{\boldsymbol{\theta}_{\text{old}}}(\boldsymbol{\theta}). \qquad \text{(TRRO)}$$

This implies the following theoretical guarantee on stability.

**Corollary 4.** *Assume exact policy optimization. Staring from an arbitrary reward parameter $\boldsymbol{\theta}_0$, (TRRO) will yield a sequence of reward functions $r_{\boldsymbol{\theta}_0}, r_{\boldsymbol{\theta}_1}, r_{\boldsymbol{\theta}_2}, \ldots$ such that the corresponding likelihood of expert demonstrations **monotonically increases**: $\ell(\boldsymbol{\theta}_0) \leq \ell(\boldsymbol{\theta}_1) \leq \ell(\boldsymbol{\theta}_2) \leq \ldots$.*

As illustated in Fig. 3, TRRO is a type of Minorization-Maximization (MM) algorithms [20], where $\ell_{\boldsymbol{\theta}_{\text{old}}}(\boldsymbol{\theta}) - C\epsilon_{\boldsymbol{\theta}_{\text{old}}}(\boldsymbol{\theta})$ is a surrogate that minorizes $\ell(\boldsymbol{\theta})$ and matches it at $\boldsymbol{\theta} = \boldsymbol{\theta}_{\text{old}}$. [6] Maximizing the surrogate ensures progress on the original objective. In light of this, TRRO plays a role in inverse RL analogous to Trust Region Policy Optimization (TRPO) [35] in forward RL: while TRPO's theoretical framework uses the MM algorithm to ensure monotonic policy improvement in expcted return with respect to a fixed reward function, our TRRO ensures monotonic expected return gap (equivalent to the likelihood) reduction with respect to the given the expert behavior.

# 5 Proximal Inverse Reward Optimization Algorithm

In this section, we develop a practical algorithm, *Proximal Inverse Reward Optimization* (PIRO). It approximates the theoretical guarantee of TRRO, enabling adpatively larger reward update steps, efficient policy optimization and accommodating continuous state-action spaces. It operates under realistic constraint of finite expert demonstrations $\mathcal{D}_E = \{(\mathbf{s}_0, \mathbf{a}_0, \ldots)\}$.

**Adaptive reward update.** The original scale factor $C$ is often too large, leading to excessively small reward updates. [7] To mitigate this, we introduce an customizable coefficient $\mu > 0$ to relax the scale. Another issue is that $\epsilon_{\boldsymbol{\theta}_{\text{old}}}(\boldsymbol{\theta})$ is indifferentiable due to its definition as the maximum norm. To address this, we replace $\epsilon_{\boldsymbol{\theta}_{\text{old}}}(\boldsymbol{\theta})$ with the differentiable $L^2$ norm of reward differences and calculate it on the state-action space for the tabular cases or, more generally, estimate it on a subset $\hat{\mathcal{D}}_E \subset \mathcal{D}_E$ and a set of rollouts $\mathcal{D}_S$ sampled from $\pi_{\text{old}}$ for continuous control:

$$\bar{\epsilon}_{\boldsymbol{\theta}_{\text{old}}}(\boldsymbol{\theta}) \coloneqq \left( \sum_{(\mathbf{s}, \mathbf{a}) \in \hat{\mathcal{D}}_E \cup \mathcal{D}_S} \left( r_{\boldsymbol{\theta}_{\text{old}}}(\mathbf{s}, \mathbf{a}) - r_{\boldsymbol{\theta}}(\mathbf{s}, \mathbf{a}) \right)^2 \right)^{1/2}. \qquad (8)$$

Note that through $\bar{\epsilon}_{\boldsymbol{\theta}_{\text{old}}}(\boldsymbol{\theta})$, we also implicitly penalize the magnitude of the reward function (the $L^2$ norm $\|r_{\boldsymbol{\theta}}(\mathbf{s}, \mathbf{a})\|_2$), similar to the reward sparsity regularization in SQIL [32], which discourages assigning high rewards to state-action pairs absent in demonstrations.

The above approximations yield the following objective for each reward update step:

$$\boldsymbol{\theta}_{\text{new}} = \arg\max_{\boldsymbol{\theta}} L_{\boldsymbol{\theta}_{\text{old}}}(\boldsymbol{\theta}) \coloneqq \ell_{\boldsymbol{\theta}_{\text{old}}}(\boldsymbol{\theta}) - \mu\bar{\epsilon}_{\boldsymbol{\theta}_{\text{old}}}(\boldsymbol{\theta}). \qquad (9)$$

We minimize $L_{\boldsymbol{\theta}_{\text{old}}}(\boldsymbol{\theta})$ using gradient descent by estimating

$$\nabla_{\boldsymbol{\theta}} L_{\boldsymbol{\theta}_{\text{old}}}(\boldsymbol{\theta}) = \mathbb{E}_{\hat{\mathcal{D}}_E}[\nabla_{\boldsymbol{\theta}} r_{\boldsymbol{\theta}}(\mathbf{s}, \mathbf{a})] - \mathbb{E}_{\mathcal{D}_S}[\nabla_{\boldsymbol{\theta}} r_{\boldsymbol{\theta}}(\mathbf{s}, \mathbf{a})] - \mu\nabla_{\boldsymbol{\theta}}\bar{\epsilon}_{\boldsymbol{\theta}_{\text{old}}}(\boldsymbol{\theta}). \qquad (10)$$

---

[6] If $\ell_{\boldsymbol{\theta}_{\text{old}}}(\boldsymbol{\theta}) - C\epsilon_{\boldsymbol{\theta}_{\text{old}}}(\boldsymbol{\theta})$ reaches a local maximum at $\boldsymbol{\theta}_{\text{old}}$, a wider search range is needed – a known limitation of MM algorithms. This, however, is out of the scope of this paper.

[7] See Appendix D.2 for an experiment for the performance comparison between theoretical and adaptive $C$.

We adaptively adjust the coefficient $\mu$ as follows:

$$\text{If } \bar{\epsilon}_{\boldsymbol{\theta}_{\text{old}}}(\boldsymbol{\theta}) > \bar{\epsilon}^{\text{target}} \times x, \text{ then } \mu \leftarrow \mu \times y; \quad \text{If } \bar{\epsilon}_{\boldsymbol{\theta}_{\text{old}}}(\boldsymbol{\theta}) < \bar{\epsilon}^{\text{target}}/x, \text{ then } \mu \leftarrow \mu/y, \quad (11)$$

where $\bar{\epsilon}^{\text{target}} > 0$, $x, y > 1$ are predefined hyperparameters. The updated $\mu$ is used for the next reward update step. Sensitivity tests for $x, y, \bar{\epsilon}^{\text{target}}$ are in Sec. 7.6.

**Practical policy optimization.** In practice, we cannot expect exact policy optimization. For efficiency, similar to the setting in ML-IRL [50, 51], we calculate $\pi_{\text{old}}$ by performing several rounds of soft policy iterations through Soft Actor-Critic [18] under $r_{\boldsymbol{\theta}_{\text{old}}}$ and $\pi_{\text{old}}$.

**Final practical algorithm.** Finally, we obtain the following practical iterative procedure for PIRO:

$$\pi \leftarrow k \text{ SAC rounds with } r_{\boldsymbol{\theta}_{\text{old}}}, \pi_{\text{old}} \underset{\boldsymbol{\theta}_{\text{old}} \leftarrow \boldsymbol{\theta}_{\text{new}}}{\overset{\pi \rightarrow \pi_{\text{old}}}{\rightleftarrows}} \boldsymbol{\theta}_{\text{new}} \leftarrow n \text{ grad. steps with } \nabla_{\boldsymbol{\theta}} L_{\boldsymbol{\theta}_{\text{old}}}(\boldsymbol{\theta}). \quad \text{(PIRO)}$$

Note that (PIRO) degrades into Alg. 2 (the general procedure of non-adversarial IRL) if $k = n = 1$ and $\mu = 0$. This indicates that, in theory, PIRO improves stability at the cost of more frequent updates. However, our empirical evaluation in the next section (Tab. 1 and Fig. 4) reveals that this added computational effort does not compromise time efficiency, as the improved stability leads to faster convergence, effectively offsetting the additional update overhead.

To summarize, we show the training procedure of PIRO in Alg. 3.

---

**Algorithm 3** Proximal Inverse Reward Optimization (PIRO)

1: **Input:** Expert demostrations $\mathcal{D}_E$; Initialized reward parameter $\boldsymbol{\theta}_{\text{old}}$, policy $\pi_{\text{old}}$; Targets $\bar{\epsilon}^{\text{target}}$, coefficient $\mu$ and scalars $x, y > 1$; Loop control parameters $m, k, n > 0$.
2: **for** $i = 1$ to $m$ **do**
3:     $\pi_{\text{old}} \leftarrow k$ rounds of SAC based on $r_{\boldsymbol{\theta}_{\text{old}}}$ and $\pi_{\text{old}}$.
4:     **for** $j = 1$ to $n$ **do**
5:         Sample a batch $\hat{\mathcal{D}}_E \subset \mathcal{D}_E$.
6:         Rollout $\pi_{\text{old}}$ to sample a set of transitions $\mathcal{D}_S$.
7:         Estimate $\nabla_{\boldsymbol{\theta}} L_{\boldsymbol{\theta}_{\text{old}}}(\boldsymbol{\theta})$ on $\hat{\mathcal{D}}_E$ and $\mathcal{D}_S$.     ▷ Eq. (10)
8:         Update $\boldsymbol{\theta}$ to increase $L_{\boldsymbol{\theta}_{\text{old}}}(\boldsymbol{\theta})$ via $\nabla_{\boldsymbol{\theta}} L_{\boldsymbol{\theta}_{\text{old}}}(\boldsymbol{\theta})$.
9:     **end for**
10:    Adjust $\mu$ and Set $\boldsymbol{\theta}_{\text{old}} \leftarrow \boldsymbol{\theta}$.     ▷ Eq. (11)
11: **end for**
12: **Output:** reward $r_{\boldsymbol{\theta}_{\text{old}}}$ and policy $\pi_{\text{old}}$.

---

## 6 Related Work

**Adversarial IRL.** Predominant IRL methods follow an adversarial learning paradigm (see GAIL [19] and discriminator-actor-critic (DAC) [24]), with AIRL variants [10–12] and extensions [48, 47, 8, 9] as key representatives. As argued in [33], this also includes methods that do not explicitly adopt a min-max game formulation but implicitly learn from its adversarial dynamics, such as classic approaches like Apprenticeship Learning [1, 2] and Max-Ent IRL [52, 53]. Recent work [39] unifies these adversarial methods through the concept of Moment Matching (a.k.a. Integral Probability Metric) [26], offering a broader perspective on their underlying principles. Building on this, recent methods further improve adversarial IRL by providing sample-efficient policy update mechnisms such as FILTER [40] (resets the learner to expert states) and HyPE [33] (a hybrid-RL based IRL algorithm that trains on a mixture of online and expert data to curtail unnecessary exploration in policy updates). In contrast to all these methods, our approach is non-adversarial and features principled stable reward learning.

**Non-adversarial IRL.** We expand the discussion on non-adversarial IRL methods in the introduction and Sec. 3. Coherent Soft Imitation Learning (CSIL) [44] simplifies the idea of non-adversarial IRL with a two-stage procedure: it first extracts a reward function from a max-likelihood policy with a reference policy and then trains a policy based on this reward. BC-IRL [41] minimizes the mean squared loss rather than maximizing the likelihood, but with no guarantee on stability. Least-squares inverse Q-learning (LSIQ) [3] penalizes the reward function magnitude and give its theoretical support; PIRO does so implicitly in its practical implementation of reward update constraints. To handle distributional shift due to limited state-action coverage, some methods adopt the model-based paradigm and conservative updates — either on the policy (Offline ML-IRL [51]) or on the reward function (CLARE [49]). In contrast, our PIRO is model-free and leverages online rollouts. Another recent method, SFM [22], minimizes the imitation gap by matching expert Successor Features (i.e., predictions of future state occupancies under a policy). A technically related method is $P^2$IL [43], which applies the proximal point method to stabilize soft Q-function learning under linear MDP assumptions. Our method addresses general MDPs with explicit rewards.

Table 1: **Averaged Rewards** (five independent runs) on five MuJoCo and four Gym Robotics tasks.

| Task | Expert | IL | | Adv. IRL (Online) | | | | Adv. (Offline) | Non-Adv. Online | | | Non-Adv. Offline | | PIRO | *Gain* |
|---|---|---|---|---|---|---|---|---|---|---|---|---|---|---|---|
| | | BC | GAIL | MM | AIRL | FILTER | HyPE | DAC | IQ | ML-IRL | $f$-IRL | CSIL | P²IL | | |
| **MuJoCo** Ant-v4 | 5926.2 | 1631.5 | 996.9 | -304.0 | 991.4 | -376.3 | 2800.5 | 923.8 | 3589.8 | 5382.5 | 980.4 | 420.7 | 976.6 | **5967.2** | +584.7 |
| Humanoid-v4 | 5501.0 | 418.1 | 508.4 | 367.2 | 281.4 | 291.7 | 717.5 | 76.3 | 1847.5 | 5573.4 | 470.4 | – | – | **5954.9** | +381.5 |
| Walker2d-v4 | 5524.5 | 384.4 | 4158.1 | 70.4 | 72.8 | 77.7 | 1478.7 | -3.0 | 3023.0 | 4794.7 | 243.8 | 686.1 | 1054.0 | **5643.7** | +849.0 |
| Hopper-v4 | 3632.8 | 1034.4 | **3535.7** | 57.8 | 13.5 | 37.3 | 2593.7 | 3321.6 | 3424.5 | 3316.4 | 361.7 | 6.7 | 25.8 | 3362.0 | -173.7 |
| Halfcheetah-v4 | 12266.1 | 221.2 | 1298.8 | 20.3 | 2251.4 | 0.3 | 6473.4 | 9645.0 | 3825.5 | 11873.2 | -0.7 | -107.2 | -0.1 | **12587.4** | +714.2 |
| **Robotics** AntMaze-UMazeDense-v4 | 35.6 | 8.8 | 5.2 | 5.1 | 4.5 | 6.1 | 11.9 | – | 3.9 | 4.2 | 3.6 | – | 3.4 | **25.7** | +13.8 |
| AntMaze-MediumDense-v4 | 26.9 | 1.1 | 1.3 | 3.4 | 2.6 | 1.9 | 3.0 | – | 3.4 | 0.9 | 1.1 | – | 2.9 | **9.4** | +6.0 |
| AntMaze-LargeDense-v1 | 11.5 | 1.1 | 0.9 | 1.7 | 3.4 | 0.6 | 1.5 | – | 0.8 | 0.3 | 0.9 | – | 0.2 | **8.8** | +5.4 |
| AdroitHandePen-Human-v1 | 1062.5 | 44.1 | -8.7 | -344.3 | -593.9 | -685.4 | -866.7 | – | -751.9 | -251.2 | -65.3 | – | -61.2 | **254.0** | +209.9 |
| runtime per iteration | - | - | 3-14s | 8-79s | 5-8s | 9-41s | 11-70s | 135-142s | 7-57s | 93-166s | 16-85s | 68-90s | 20-111s | 96-178s | – |

**Note:** DAC, CSIL and P²IL are not evaluated on certain tasks due to compatibility issues cause by version conflicts. Specifically, the current implementations of DAC and P²IL are incompatible with the current Gymnasium Robotics suite, while P²IL and CSIL are incompatible with the Humanoid version used in testing other algorithms.

**Stable Inverse Optimal Control.** A line of work in inverse optimal control uses trust-region or Lyapunov-based methods [37, 5, 42] to ensure stability but requires knowledge of system dynamics and second-order optimization, limiting scalability. PIRO, in contrast, is model-free and relies only on first-order optimization, making it more practical for real-world applications.

# 7 Experiments

We focus on the following key performance indicators in the empirical evaluation: **(1)** reward recovery and policy imitation, **(2)** learning stability, **(3)** sample efficiency. We also test PIRO's capability of reward transfer to new environment dynamics and learning state-only rewards We evaluate alogrithms on five MuJoCo locomotion and four Gym-Robotics tasks (see Tab. 1). To examine PIRO's capability of real-world problem solving, we additionally provide a real-world case study on an animal behavior modeling task in Appendix E, where PIRO shows superior performance compared to baselines.

**Experimental Setup.** For MuJoCo tasks, we use the same demonstrations as $f$-IRL [30] and ML-IRL [50], keeping original hyperparameters except for standardized batch sizes and training steps to ensure fair comparison under identical computational budgets. Robotic tasks use expert trajectories from Minari Offline RL datasets [46]. We use a *single* expert trajectory per task in order to examine their imitation capability; the only exception is AdroitHandPen, where we use 10 expert trajectories instead of one to ensure convergence. Full implementation details, including hyperparameters, network architectures and trajectory lengths, are in Appendix C.

## 7.1 Reward Recovery and Policy Imitation

The reward performance is shown in Tab. 1. PIRO consistently outperforms or matches all baselines across nearly all tasks. The performance gains are especially pronounced in harder domains such as Humanoid, AntMaze, and AdroitHand, where PIRO shows substantial improvements over the best baseline. On average, PIRO demonstrates strong reward recovery and policy imitation. Although PIRO incurs a moderately higher computation time per iteration, this reflects its principled stable reward optimization mechanism: the increased runtime stems from controlled updates that ensure *stable policy improvement* (justified in the next experiment).

## 7.2 Learning Stability

We investigate learning stability by analyzing the learning curves across all experimental tasks, which are shown in Fig. 4. PIRO consistently outperforms ML-IRL and demonstrates significantly higher stability compared to other baselines throughout the learning process (except slightly weaker performance on AntMaze-MediumDense-v4). In challenging AntMaze environments, while PIRO exhibits fluctuation, it remains the only method capable of successfully imitating expert behavior, likely due to the complex environment dynamics that cause the failures of other algorithms.

## 7.3 Sample Efficiency

We assess sample efficiency by analyzing the convergence speed with respect to the environment steps, which can be observed in Fig. 4. PIRO consistently delivers competitive or faster convergence speed. Although in certain environments our method exhibits lower sample efficiency than some baselines (e.g., DAC on HalfCheetah-v4), PIRO ultimately achieves higher final rewards after convergence

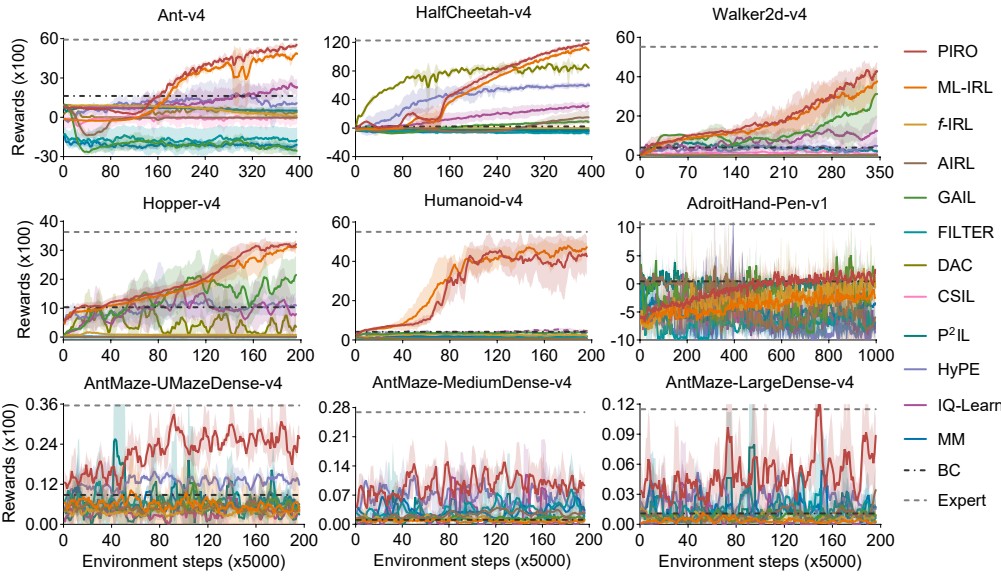

Figure 4: **Reward curves of algorithms** on MuJoCo locomotion tasks and Gym Robotics tasks.

and approaches expert-level performancem, while most baselines are far from expert performance after convergence. Moreover, in these environments PIRO demonstrates more stable improvements throughout training.

## 7.4 Learning State-only Rewards

As explored in [12], restricting rewards to be solely state-dependent mitigates ambiguity from *reward shaping* [29], that is, a class of reward transformations that yield the same optimal policies, making it impossible for an IRL algorithm to identify the true reward without prior knowledge of the environment. This also improves generalization across MDPs with different dynamics. Thanks to explicit reward learning, PIRO naturally supports state-only rewards by directly parameterizing $r_\theta(\mathbf{s})$, without the additional modifications required by implicit reward methods [15]. Empirically, we demonstrate PIRO's effectiveness in recovering state-only ground-truth rewards in Fig. 5.

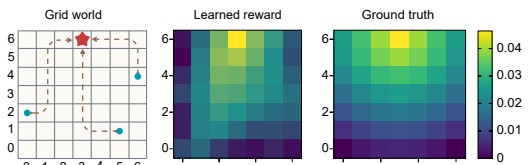

Figure 5: **Experiments on reward recovery in tasks with state-only rewards. Left:** The task is a $7 \times 7$ grid world, where the agent starts from a random initial position (blue circles) with the objective of reaching the target position (red star) via the shortest possible path. **Right:** The ground truth reward at each position is defined as the negative Euclidean distance to the terminal state. **Middle:** The reward recovered by PIRO and the ground-truth reward function is highly consistent with the ground truth reward. Cumulative rewards: $-9.24$ (expert) vs. $-8.48$ (PIRO).

## 7.5 Reward Transfer

To assess the transferability of the learned reward function, we evaluate whether a reward learned under the original environment dynamics can induce an effective policy when the dynamics change. LunarLander provides a testbed for this as we can alter its dynamics by "adding winds" in the simulated physical conditions. As shown in Fig. 6, the resulting policy performs well under the modified dynamics, demonstrating that PIRO recovers robust reward functions capable of generalizing across environmental changes.

## 7.6 Sensitivity Tests

To assess the robustness of PIRO with respect to hyperparameters controlling reward update magnitude, we conduct sensitivity tests on three key parameters: $\bar{\epsilon}^{\text{target}}$ and its associated scaling factors $x$ and $y$, which govern the adaptive adjustment of the regularization coefficient $\mu$ in Eq. (11). Specifi-

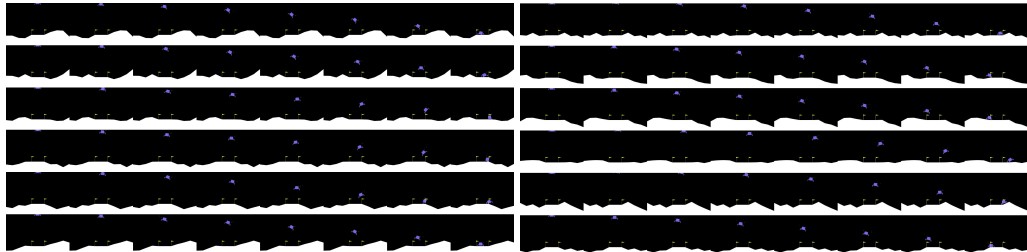

Figure 6: **Results for reward transfer to new environments with altered dynamics. Left panels:** Policy behavior learned by PIRO in the original LunarLander environment. PIRO succeeds in most cases. **Right panels:** Policy behavior under PIRO's learned reward function in LunarLander with altered dynamics (stochastic wind added). The policy is robust in general, despite some failure cases, e.g., row 3.

cally, we vary one parameter at a time while keeping all others fixed. Results are reported in Fig. 7, which suggest that the algorithm is not highly sensitive to the hyperparameters $x, y$; both can be set within the range $(1, 2)$ without significant impact. We also observe that setting the target value $\bar{\epsilon}^{\text{target}}$ within the range $(0.1, 1)$ generally does not significantly affect the reward performance.

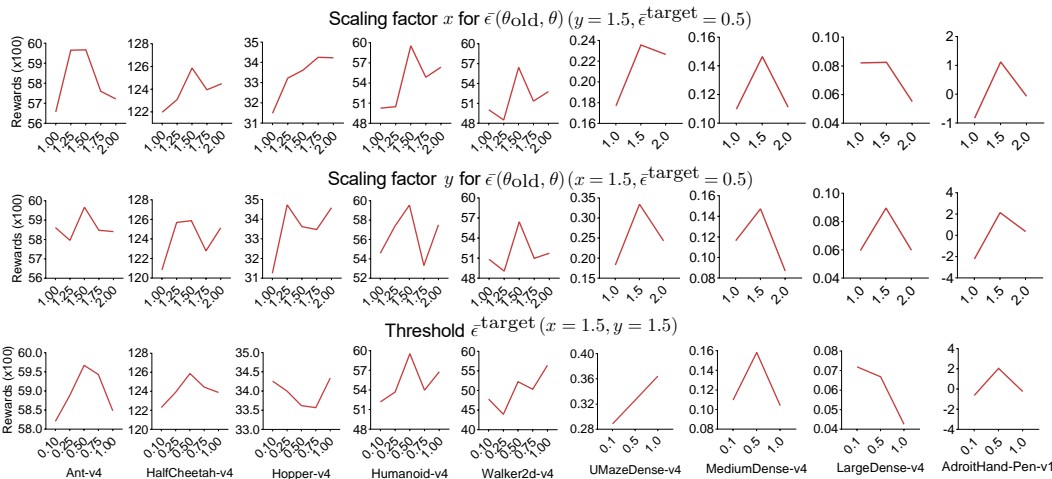

Figure 7: **Sensitivity test** for the parameter $\bar{\epsilon}^{\text{target}}$ and its scaling factors $x, y$.

## 8    Conclusion

We propose Proximal Inverse Reward Optimization (PIRO), a novel non-adversarial, practical IRL algorithm that stabilizes reward learning by approximating Trust Region Reward Optimization (TRRO) – a novel theoretical framework guaranteeing monotonic improvement in the likelihood of expert behavior. Experiments MuJoCo and Gym Robotics benchmarks show that PIRO achieves stable training, accurate and robust reward recovery, high sample efficiency, and good reward transfer capability. This work provides a theoretical foundation for stabilizing IRL, and we hope it provides a new perspective for designing more robust IRL algorithms.

**Limitations.** Despite its advantages, PIRO has limitations. First, while it stabilizes reward learning, the overall training stability also depends on a stable policy optimizer, especially in high-dimensional and complex-dynamics settings. Second, the dependency on on-policy sampling may reduce sample efficiency in environment interactions, potentially limiting scalability to sample-expensive tasks.

**Future work.** First, improving the efficiency of policy optimization by incorporating resets to expert states [40, 33] may substantially reduce computational cost. Second, exploring alternative policy alignment measures beyond likelihood (e.g., statistical divergences other than KL) may open new paradigms for stable IRL. Finally, on the application side, extending PIRO to real-world scenarios — such as learning reward models and policies for aligning large language models with human feedback — offers a promising path to improving agent performance in practice.

## Acknowledgments and Disclosure of Funding

This work was supported by a locally commissioned task from the Shanghai Municipal Government.

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

# Appendices

## A   Proofs

### A.1   Proof of Proposition 1

Let us first show that $\ell(\boldsymbol{\theta}) = \mathbb{E}_{\rho^{\pi_E}}[\log \pi_{\boldsymbol{\theta}}(\mathbf{a}|\mathbf{s})] = \mathbb{E}_{\rho^{\pi_E}}[r_{\boldsymbol{\theta}}(\mathbf{s}, \mathbf{a})] - \mathbb{E}_{\mathbf{s}_0 \sim \eta}[V_{r_{\boldsymbol{\theta}}}^{\pi_{\boldsymbol{\theta}}}(\mathbf{s}_0)] = J(\pi_E, r_{\boldsymbol{\theta}}) - J(\pi_{\boldsymbol{\theta}}, r_{\boldsymbol{\theta}})$ (Eq. (5a)). Let $d_t^{\pi}(\mathbf{s})$ denote the state distribution under a policy $\pi$. Note that $d_0^{\pi} \equiv \eta$, where $\eta$ is the fixed initial state distribution.

$$
\begin{aligned}
\ell(\boldsymbol{\theta}) &= \mathbb{E}_{\rho^{\pi_E}}[\log \pi_{\boldsymbol{\theta}}(\mathbf{a}|\mathbf{s})] \\
&= \mathbb{E}_{\rho^{\pi_E}}[Q_{r_{\boldsymbol{\theta}}}^{\pi_{\boldsymbol{\theta}}}(\mathbf{s}, \mathbf{a}) - V_{r_{\boldsymbol{\theta}}}^{\pi_{\boldsymbol{\theta}}}(\mathbf{s})] \\
&= \mathbb{E}_{\rho^{\pi_E}}[r_{\boldsymbol{\theta}}(\mathbf{s}, \mathbf{a}) + \gamma \mathbb{E}_{\mathbf{s}' \sim P(\cdot|\mathbf{s}, \mathbf{a})}[V_{r_{\boldsymbol{\theta}}}^{\pi_{\boldsymbol{\theta}}}(\mathbf{s}')] - V_{r_{\boldsymbol{\theta}}}^{\pi_{\boldsymbol{\theta}}}(\mathbf{s})] \\
&= \mathbb{E}_{\rho^{\pi_E}}[r_{\boldsymbol{\theta}}(\mathbf{s}, \mathbf{a})] - \sum_{t=0}^{\infty} \gamma^t \mathbb{E}_{\mathbf{s} \sim d_t^{\pi_E}, \mathbf{a} \sim \pi_E(\cdot|\mathbf{s})}\left[V_{r_{\boldsymbol{\theta}}}^{\pi_{\boldsymbol{\theta}}}(\mathbf{s}) - \gamma \mathbb{E}_{\mathbf{s}' \sim P(\cdot|\mathbf{s}, \mathbf{a})}[V_{r_{\boldsymbol{\theta}}}^{\pi_{\boldsymbol{\theta}}}(\mathbf{s})]\right] \\
&= \mathbb{E}_{\rho^{\pi_E}}[r_{\boldsymbol{\theta}}(\mathbf{s}, \mathbf{a})] - \left(\sum_{t=0}^{\infty} \gamma^t \mathbb{E}_{\mathbf{s} \sim d_t^{\pi_E}}[V_{r_{\boldsymbol{\theta}}}^{\pi_{\boldsymbol{\theta}}}(\mathbf{s})] - \sum_{t=0}^{\infty} \gamma^{t+1} \mathbb{E}_{\mathbf{s} \sim d_{t+1}^{\pi_E}}[V_{r_{\boldsymbol{\theta}}}^{\pi_{\boldsymbol{\theta}}}(\mathbf{s})]\right) \\
&= \mathbb{E}_{\rho^{\pi_E}}[r_{\boldsymbol{\theta}}(\mathbf{s}, \mathbf{a})] - \mathbb{E}_{\mathbf{s} \sim d_0^{\pi_E}}[V_{r_{\boldsymbol{\theta}}}^{\pi_{\boldsymbol{\theta}}}(\mathbf{s})] \\
&= \mathbb{E}_{\rho^{\pi_E}}[r_{\boldsymbol{\theta}}(\mathbf{s}, \mathbf{a})] - \mathbb{E}_{\mathbf{s} \sim \eta}[V_{r_{\boldsymbol{\theta}}}^{\pi_{\boldsymbol{\theta}}}(\mathbf{s})] \\
&= J(\pi_E, r_{\boldsymbol{\theta}}) - J(\pi_{\boldsymbol{\theta}}, r_{\boldsymbol{\theta}}).
\end{aligned} \tag{12}
$$

Note that in Eq. (12), we omit the constant policy entropy $\mathcal{H}(\pi_E)$ in $J(\pi_E, r_{\boldsymbol{\theta}})$.

We next show
$$
\nabla_{\boldsymbol{\theta}} \ell(\boldsymbol{\theta}) = \mathbb{E}_{\rho^{\pi_E}}[\nabla_{\boldsymbol{\theta}} r_{\boldsymbol{\theta}}(\mathbf{s}, \mathbf{a})] - \mathbb{E}_{\rho^{\pi_{\boldsymbol{\theta}}}}[\nabla_{\boldsymbol{\theta}} r_{\boldsymbol{\theta}}(\mathbf{s}, \mathbf{a})]
$$
in Eq. (6a). Let us begin with investigating the gradient of $Q_{r_{\boldsymbol{\theta}}}^{\pi_{\boldsymbol{\theta}}}(\mathbf{s}_t, \mathbf{a}_t)$:

$$
\begin{aligned}
&\nabla_{\boldsymbol{\theta}} Q_{r_{\boldsymbol{\theta}}}^{\pi_{\boldsymbol{\theta}}}(\mathbf{s}_t, \mathbf{a}_t) \\
&\overset{(a)}{=} \nabla_{\boldsymbol{\theta}} r_{\boldsymbol{\theta}}(\mathbf{s}_t, \mathbf{a}_t) + \gamma \sum_{\mathbf{s}_{t+1}} P(\mathbf{s}_{t+1}|\mathbf{s}_t, \mathbf{a}_t) \nabla_{\boldsymbol{\theta}} V_{r_{\boldsymbol{\theta}}}^{\pi_{\boldsymbol{\theta}}}(\mathbf{s}_{t+1}) \\
&\overset{(b)}{=} \nabla_{\boldsymbol{\theta}} r_{\boldsymbol{\theta}}(\mathbf{s}_t, \mathbf{a}_t) + \gamma \sum_{\mathbf{s}_{t+1}} P(\mathbf{s}_{t+1}|\mathbf{s}_t, \mathbf{a}_t) \sum_{\mathbf{a}_{t+1}} \frac{\exp(Q_{r_{\boldsymbol{\theta}}}^{\pi_{\boldsymbol{\theta}}}(\mathbf{s}_{t+1}, \mathbf{a}_{t+1}))}{\sum_{\mathbf{a}'} \exp(Q_{r_{\boldsymbol{\theta}}}^{\pi_{\boldsymbol{\theta}}}(\mathbf{s}_{t+1}, \mathbf{a}'))} \nabla_{\boldsymbol{\theta}} Q_{r_{\boldsymbol{\theta}}}^{\pi_{\boldsymbol{\theta}}}(\mathbf{s}_{t+1}, \mathbf{a}_{t+1}) \\
&= \nabla_{\boldsymbol{\theta}} r_{\boldsymbol{\theta}}(\mathbf{s}_t, \mathbf{a}_t) + \gamma \sum_{\mathbf{s}_{t+1}} P(\mathbf{s}_{t+1}|\mathbf{s}_t, \mathbf{a}_t) \sum_{\mathbf{a}_{t+1}} \pi_{\boldsymbol{\theta}}(\mathbf{a}_{t+1}|\mathbf{s}_{t+1}) \nabla_{\boldsymbol{\theta}} Q_{r_{\boldsymbol{\theta}}}^{\pi_{\boldsymbol{\theta}}}(s_{t+1}, a_{t+1}) \\
&= \cdots
\end{aligned} \tag{13}
$$

Equality $(a)$ uses the soft Bellman equation, while Equality $(b)$ follows the energy-based formulation of the policy. Notably, both $\nabla_{\boldsymbol{\theta}} Q_{r_{\boldsymbol{\theta}}}^{\pi_{\boldsymbol{\theta}}}(\mathbf{s}_t, \mathbf{a}_t)$ and $\nabla_{\boldsymbol{\theta}} V_{r_{\boldsymbol{\theta}}}^{\pi_{\boldsymbol{\theta}}}(\mathbf{s}_t)$ exhibit recursive forms, where the gradient $\nabla_{\boldsymbol{\theta}} r_{\boldsymbol{\theta}}(\mathbf{s}_t, \mathbf{a}_t)$ accumulates as an expectation alongside the expansion of $Q_{r_{\boldsymbol{\theta}}}^{\pi_{\boldsymbol{\theta}}}$ and $V_{r_{\boldsymbol{\theta}}}^{\pi_{\boldsymbol{\theta}}}$. Continuing this recursive expansion, we derive:

$$
\nabla_{\boldsymbol{\theta}} Q_{r_{\boldsymbol{\theta}}}^{\pi_{\boldsymbol{\theta}}}(\mathbf{s}_t, \mathbf{a}_t) = \mathbb{E}_{\substack{\mathbf{s}_{l+1} \sim P(\cdot|\mathbf{s}_l, \mathbf{a}_l), \\ \mathbf{a}_{l+1} \sim \pi_{\boldsymbol{\theta}}(\cdot|\mathbf{s}_{l+1})}}\left[\sum_{l=t}^{\infty} \gamma^{l-t} \nabla_{\boldsymbol{\theta}} r_{\boldsymbol{\theta}}(\mathbf{s}_l, \mathbf{a}_l)\right]. \tag{14}
$$

$$
\nabla_{\boldsymbol{\theta}} V_{r_{\boldsymbol{\theta}}}^{\pi_{\boldsymbol{\theta}}}(\mathbf{s}_t) = \mathbb{E}_{\substack{\mathbf{a}_\ell \sim \pi_{\boldsymbol{\theta}}(\cdot|\mathbf{s}_\ell), \\ \mathbf{s}_{\ell+1} \sim P(\cdot|\mathbf{s}_\ell, \mathbf{a}_\ell)}}\left[\sum_{\ell=t}^{\infty} \gamma^{\ell-t} \nabla_{\boldsymbol{\theta}} r_{\boldsymbol{\theta}}(\mathbf{s}_\ell, \mathbf{a}_\ell)\right]. \tag{15}
$$

Then, we have

$$
\nabla_{\boldsymbol{\theta}} V_{r_{\boldsymbol{\theta}}}^{\pi_{\boldsymbol{\theta}}}(\mathbf{s}_0) = \mathbb{E}_{\substack{\mathbf{a}_t \sim \pi_{\boldsymbol{\theta}}(\cdot|\mathbf{s}_t), \\ \mathbf{s}_{t+1} \sim d_{t+1}^{\pi_{\boldsymbol{\theta}}}}}\left[\sum_{\ell=0}^{\infty} \gamma^t \nabla_{\boldsymbol{\theta}} r_{\boldsymbol{\theta}}(\mathbf{s}_t, \mathbf{a}_t)\right]. \tag{16}
$$

Finally, according to Eq. (12), we have

$$\nabla_{\boldsymbol{\theta}}\ell(\boldsymbol{\theta}) = \mathbb{E}_{\rho^{\pi_E}}[\nabla_{\boldsymbol{\theta}}r_{\boldsymbol{\theta}}(\mathbf{s},\mathbf{a})] - \mathbb{E}_{\mathbf{s}_0 \sim \eta}[\nabla_{\boldsymbol{\theta}}V_{r_{\boldsymbol{\theta}}}^{\pi_{\boldsymbol{\theta}}}(\mathbf{s}_0)]$$

$$= \mathbb{E}_{\rho^{\pi_E}}[\nabla_{\boldsymbol{\theta}}r_{\boldsymbol{\theta}}(\mathbf{s},\mathbf{a})] - \mathbb{E}_{\mathbf{s}_0 \sim \eta, \mathbf{a}_t \sim \pi_{\boldsymbol{\theta}}(\cdot|\mathbf{s}_t), \mathbf{s}_{t+1} \sim d_{t+1}^{\pi_{\boldsymbol{\theta}}}}\left[\sum_{\ell=0}^{\infty}\gamma^t \nabla_{\boldsymbol{\theta}}r_{\boldsymbol{\theta}}(\mathbf{s}_t, \mathbf{a}_t)\right] \quad (17)$$

$$= \mathbb{E}_{\rho^{\pi_E}}[\nabla_{\boldsymbol{\theta}}r_{\boldsymbol{\theta}}(\mathbf{s},\mathbf{a})] - \mathbb{E}_{\rho^{\pi_{\boldsymbol{\theta}}}}[\nabla_{\boldsymbol{\theta}}r_{\boldsymbol{\theta}}(\mathbf{s},\mathbf{a})].$$

## A.2 Proof of Theorem 3

We begin by presenting some useful lemmas that tell us how much the policy discrepancy (Lemma 5), state margin discrepancy (Lemma 6), log policy discrepancy (Lemma 7), $Q$ and $V$ functions (Lemma 8), log policy (Lemma 9), and the expected entropy discrepancy (Lemma 10) grows based on the reward difference. In all these lemmas, we use the following notations:

- $r_1(\mathbf{s},\mathbf{a})$ and $r_2(\mathbf{s},\mathbf{a})$ are two reward functions,
- $\pi_1(\cdot|\mathbf{s})$ and $\pi_2(\cdot|\mathbf{s})$ are optimal policies under $r_1$ and $r_2$ under the MaxEnt RL framework, respectively.
- $\epsilon := \max_{(\mathbf{s},\mathbf{a})}|r_1(\mathbf{s},\mathbf{a}) - r_2(\mathbf{s},\mathbf{a})|$ denotes the reward difference.
- $|r_i(\mathbf{s},\mathbf{a})| \leq R, \forall, i \in \{1,2\}, \mathbf{s} \in \mathcal{S}, \mathbf{a} \in \mathcal{A}$.

**Lemma 5.** *The total variation distance between $\pi_1(\cdot|\mathbf{s})$ and $\pi_2(\cdot|\mathbf{s})$ is upper-bounded as follows:*

$$D_{\mathrm{TV}}(\pi_1(\cdot s), \pi_2(\cdot|\mathbf{s})) = \frac{1}{2}\|\pi_1(\cdot|\mathbf{s}) - \pi_2(\cdot|\mathbf{s})\|_1 = \frac{1}{2}\sum_{\mathbf{a} \in \mathcal{A}}|\pi_1(\mathbf{a}|\mathbf{s}) - \pi_2(\mathbf{a}|\mathbf{s})| \leq \frac{|\mathcal{A}|\epsilon}{2(1-\gamma)}. \quad (18)$$

*Proof.* We start by analyzing the sensitivity of the policy to changes in soft $Q$-function. The difference in $\pi_1(\mathbf{a}|\mathbf{s})$ and $\pi_2(\mathbf{a}|\mathbf{s})$ arises from the difference in their respective soft $Q$-functions, $Q_1(\mathbf{s},\mathbf{a})$ and $Q_2(\mathbf{s},\mathbf{a})$. Expanding the policies gives:

$$\pi_1(\mathbf{a}|\mathbf{s}) - \pi_2(\mathbf{a}|\mathbf{s}) = \frac{\exp(Q_1(\mathbf{s},\mathbf{a}))}{\sum_{a'}\exp(Q_1(\mathbf{s},\mathbf{a}'))} - \frac{\exp(Q_2(\mathbf{s},\mathbf{a}))}{\sum_{\mathbf{a}'}\exp(Q_2(\mathbf{s},\mathbf{a}'))}. \quad (19)$$

This softmax-like function is $\frac{1}{\alpha}$-Lipschitz continuous [14] with $\alpha$ being the temperature in the energy-based model (w.l.o.g., we assume $\alpha = 1$ in this paper). This means small changes in $Q$ lead to proportionally small changes in the softmax output. This allows us to approximate the policy difference for small deviations in $Q$. Thus, the policy difference can be bounded as:

$$|\pi_1(\mathbf{a}|\mathbf{s}) - \pi_2(\mathbf{a}|\mathbf{s})| \leq |Q_1(\mathbf{s},\mathbf{a}) - Q_2(\mathbf{s},\mathbf{a})|. \quad (20)$$

Summing over actions, the (doubled) total variation distance becomes:

$$\|\pi_1(\cdot|\mathbf{s}) - \pi_2(\cdot|\mathbf{s})\|_1 = \sum_{\mathbf{a} \in \mathcal{A}}|\pi_1(\mathbf{a}|\mathbf{s}) - \pi_2(\mathbf{a}|\mathbf{s})| \leq \sum_{\mathbf{a} \in \mathcal{A}}|Q_1((\mathbf{s},\mathbf{a})) - Q_2((\mathbf{s},\mathbf{a}))|. \quad (21)$$

We bound $|Q_1(\mathbf{s},\mathbf{a}) - Q_2(\mathbf{s},\mathbf{a})|$ by:

$$\max_{\mathbf{s},\mathbf{a}}|Q_1(\mathbf{s},\mathbf{a}) - Q_2(\mathbf{s},\mathbf{a})|$$

$$\leq \epsilon + \gamma \max_{(\mathbf{s},\mathbf{a})}\mathbb{E}_{\mathbf{s}' \sim P(\cdot|(\mathbf{s},\mathbf{a}))}[V_1(\mathbf{s}') - V_2(\mathbf{s}')]$$

$$\leq \epsilon + \gamma \max_{\mathbf{s}',\mathbf{a}'}\left|\log\sum_{\mathbf{a}'}\exp(Q_1(\mathbf{s}',\mathbf{a}')) - \log\sum_{\mathbf{a}'}\exp(Q_2(\mathbf{s}',\mathbf{a}'))\right| \quad (22)$$

$$\overset{(a)}{\leq} \epsilon + \gamma \max_{(\mathbf{s},\mathbf{a})}|Q_1(\mathbf{s},\mathbf{a}) - Q_2(\mathbf{s},\mathbf{a})|,$$

where inequality $(a)$ uses the fact that for any two sets of values $\{x_i\}$ and $\{y_i\}$,

$$\left|\log\sum_i\exp(x_i) - \log\sum_i\exp(y_i)\right| \leq \max_i|x_i - y_i|. \quad (23)$$

Rearranging Eq. (22) and performing some algebra yields:

$$\max_{(\mathbf{s},\mathbf{a})} |Q_1(\mathbf{s}, \mathbf{a}) - Q_2(\mathbf{s}, \mathbf{a})| \leq \frac{\epsilon}{1-\gamma}.$$

Finally, according to Eq. (21) summing over action space introduces scaling:

$$\|\pi_1(\cdot|\mathbf{s}) - \pi_2(\cdot|\mathbf{s})\|_1 \leq |\mathcal{A}|\,|Q_1(\mathbf{s}, \mathbf{a}) - Q_2(\mathbf{s}, \mathbf{a})| \quad \leq \frac{|\mathcal{A}|\epsilon}{1-\gamma}. \tag{24}$$

$\square$

**Lemma 6.** *Let $d_t^{\pi_1}(\mathbf{s})$ and $d_t^{\pi_2}(\mathbf{s})$ denote the state marginal distributions at time $t$ under each policy, starting from the same initial distribution $\eta$. Then, for any $t \geq 0$,*

$$D_{\mathrm{TV}}(d_t^{\pi_1}, d_t^{\pi_2}) = \frac{1}{2}\|d_t^{\pi_2} - d_t^{\pi_1}\|_1 \leq t D_{\mathrm{TV}}^{\max}(\pi_1, \pi_2), \tag{25}$$

*where*

$$D_{\mathrm{TV}}^{\max}(\pi_1, \pi_2) := \max_{\mathbf{s}} D_{\mathrm{TV}}(\pi_1(\cdot|\mathbf{s}), \pi_2(\cdot|\mathbf{s})) = \frac{\epsilon}{1-\gamma} \quad \text{(Lemma 5)} \tag{26}$$

*is the worst-case total variation distance between $\pi_2$ and $\pi_1$ over all states.*

*Proof.* We proceed by induction on $t$.

**Base case** ($t = 0$): At $t = 0$, $d_0^{\pi_2} = d_0^{\pi_1} = \eta$ (the initial distribution), so

$$\|d_0^{\pi_2} - d_0^{\pi_1}\|_1 = 0, \tag{27}$$

which satisfies the bound.

**Inductive step**: Suppose that at time $t$,

$$D_{\mathrm{TV}}(d_t^{\pi_1}, d_t^{\pi_2}) \leq t D_{\mathrm{TV}}^{\max}(\pi_1, \pi_2). \tag{28}$$

We now show that the same holds at time $t + 1$.

The state marginals evolve according to

$$d_{t+1}^{\pi}(\mathbf{s}') = \sum_{\mathbf{s}} d_t^{\pi}(\mathbf{s}) \sum_{\mathbf{a}} \pi(\mathbf{a}|\mathbf{s}) P(\mathbf{s}'|\mathbf{s}, \mathbf{a}).$$

Thus,

$$d_{t+1}^{\pi_2}(\mathbf{s}') - d_{t+1}^{\pi_1}(\mathbf{s}') = \sum_{\mathbf{s}} \sum_{\mathbf{a}} \left( d_t^{\pi_2}(\mathbf{s})\pi_2(\mathbf{a}|\mathbf{s}) - d_t^{\pi_1}(\mathbf{s})\pi_1(\mathbf{a}|\mathbf{s}) \right) P(\mathbf{s}'|\mathbf{s}, \mathbf{a}). \tag{29}$$

Taking the $L^1$ norm and using the triangle inequality,

$$\|d_{t+1}^{\pi_2} - d_{t+1}^{\pi_1}\|_1 \leq \sum_{\mathbf{s}} \|d_t^{\pi_2}(\mathbf{s})\pi_2(\cdot|\mathbf{s}) - d_t^{\pi_1}(\mathbf{s})\pi_1(\cdot|\mathbf{s})\|_1. \tag{30}$$

Now expand the difference inside:

$$d_t^{\pi_2}(\mathbf{s})\pi_2(\mathbf{a}|\mathbf{s}) - d_t^{\pi_1}(\mathbf{s})\pi_1(a|\mathbf{s}) = (d_t^{\pi_2}(\mathbf{s}) - d_t^{\pi_1}(\mathbf{s}))\pi_2(\mathbf{a}|\mathbf{s}) + d_t^{\pi_1}(\mathbf{s})\left(\pi_2(\mathbf{a}|\mathbf{s}) - \pi_1(\mathbf{a}|\mathbf{s})\right). \tag{31}$$

Using triangle inequality again:

$$\|d_t^{\pi_2}(\mathbf{s})\pi_2(\cdot|\mathbf{s}) - d_t^{\pi_1}(\mathbf{s})\pi_1(\cdot|\mathbf{s})\|_1 \leq |d_t^{\pi_2}(\mathbf{s}) - d_t^{\pi_1}(\mathbf{s})| + d_t^{\pi_1}(\mathbf{s})\|\pi_2(\cdot|\mathbf{s}) - \pi_1(\cdot|\mathbf{s})\|_1. \tag{32}$$

Thus,

$$\|d_{t+1}^{\pi_2} - d_{t+1}^{\pi_1}\|_1 \leq \sum_{\mathbf{s}} |d_t^{\pi_2}(\mathbf{s}) - d_t^{\pi_1}(\mathbf{s})| + \sum_{\mathbf{s}} d_t^{\pi_1}(\mathbf{s})\|\pi_2(\cdot|\mathbf{s}) - \pi_1(\cdot|\mathbf{s})\|_1. \tag{33}$$

The first term of the right-hand side is simply

$$\|d_t^{\pi_2} - d_t^{\pi_1}\|_1 = 2 D_{\mathrm{TV}}(d_t^{\pi_1}, d_t^{\pi_2}), \tag{34}$$

and the second term is at most

$$2D_{\mathrm{TV}}^{\max}(\pi_2, \pi_1), \tag{35}$$

since $d_t^{\pi_1}$ is a distribution and $\|\pi_2(\cdot|\mathbf{s}) - \pi_1(\cdot|\mathbf{s})\|_1 \leq 2D_{\mathrm{TV}}^{\max}(\pi_2, \pi_1)$ for all $\mathbf{s}$.

Therefore,

$$D_{\mathrm{TV}}(\rho_{t+1}^{\pi_1}, d_{t+1}^{\pi_2}) = \frac{1}{2}\|d_{t+1}^{\pi_2} - d_{t+1}^{\pi_1}\|_1 \leq D_{\mathrm{TV}}(d_t^{\pi_1}, d_t^{\pi_2}) + D_{\mathrm{TV}}^{\max}(\pi_2, \pi_1). \tag{36}$$

Applying the inductive hypothesis:

$$D_{\mathrm{TV}}(d_t^{\pi_1}, d_t^{\pi_2}) \leq t D_{\mathrm{TV}}^{\max}(\pi_1, \pi_2), \tag{37}$$

we conclude

$$D_{\mathrm{TV}}(d_{t+1}^{\pi_1}, d_{t+1}^{\pi_2}) \leq (t+1) D_{\mathrm{TV}}^{\max}(\pi_1, \pi_2). \tag{38}$$

Thus, the claim holds for $t+1$, completing the induction. $\square$

**Lemma 7.** *Under MaxEnt RL, let $\pi_1(\mathbf{a}|\mathbf{s})$ and $\pi_2(\mathbf{a}|\mathbf{s})$ be two policies defined over a finite action set $\mathcal{A}$, induced by reward functions $r_1(\mathbf{s}, \mathbf{a})$ and $r_2(\mathbf{s}, \mathbf{a})$ respectively. Assume that for all $\mathbf{s}, \mathbf{a}$, the rewards are uniformly bounded by a constant $R > 0$, i.e., $|r_i(\mathbf{s}, \mathbf{a})| \leq R$, for $i = 1, 2$. Let $\epsilon = \max_{\mathbf{s}, \mathbf{a}} |r_1(\mathbf{s}, \mathbf{a}) - r_2(\mathbf{s}, \mathbf{a})|$. Then, the log-policy difference is bounded as:*

$$\|\log \pi_1(\cdot|\mathbf{s}) - \log \pi_2(\cdot|\mathbf{s})\|_\infty \leq \frac{2\epsilon}{1-\gamma}. \tag{39}$$

*Proof.* We start from the softmax policy expression:

$$\log \pi_i(\mathbf{a}|\mathbf{s}) = Q_i(\mathbf{s}, \mathbf{a}) - \log \sum_{\mathbf{a}' \in \mathcal{A}} \exp(Q_i(\mathbf{s}, \mathbf{a}')), \quad i = 1, 2. \tag{40}$$

So the difference is:

$$\log \pi_1(\mathbf{a}|\mathbf{s}) - \log \pi_2(\mathbf{a}|\mathbf{s}) = Q_1(\mathbf{s}, \mathbf{a}) - Q_2(\mathbf{s}, \mathbf{a}) -$$
$$\left[ \log \sum_{\mathbf{a}'} \exp(Q_1(\mathbf{s}, \mathbf{a}')) - \log \sum_{\mathbf{a}'} \exp(Q_2(\mathbf{s}, \mathbf{a}')) \right]. \tag{41}$$

Following from the Lipschitz continuity of the $\log \sum \exp(\cdot)$ function with Lipschitz constant 1 under $L^\infty$-norm, we have

$$\left| \log \sum_{\mathbf{a}} \exp(Q_1(\mathbf{s}, \mathbf{a})) - \log \sum_{\mathbf{a}} \exp(Q_2(\mathbf{s}, \mathbf{a})) \right| \leq \|Q_1 - Q_2\|_\infty = \frac{\epsilon}{1-\gamma} \quad \text{(Lemma 5)}. \tag{42}$$

Combining everything:

$$|\log \pi_1(\mathbf{a}|\mathbf{s}) - \log \pi_2(\mathbf{a}|\mathbf{s})|$$
$$\leq |Q_1(\mathbf{s}, \mathbf{a}) - Q_2(\mathbf{s}, \mathbf{a})| + \left| \log \sum_{\mathbf{a}} \exp(Q_1(\mathbf{s}, \mathbf{a})) - \log \sum_{\mathbf{a}} \exp(Q_2(\mathbf{s}, \mathbf{a})) \right|$$
$$\leq 2\|Q_1 - Q_2\|_\infty$$
$$= \frac{2\epsilon}{1-\gamma}. \tag{43}$$

$\square$

**Lemma 8.** *Under MaxEnt RL, we have*

$$\|Q\|_\infty \leq \frac{R + \gamma \log |\mathcal{A}|}{1-\gamma} \tag{44}$$

$$\|V\|_\infty \leq \frac{R + \log |\mathcal{A}|}{1-\gamma} \tag{45}$$

*Proof.*

$$V(\mathbf{s}) \le \log \sum_{\mathbf{a}} \exp(Q(\mathbf{s}, \mathbf{a})) \le \|Q\|_\infty + \log |\mathcal{A}|. \tag{46}$$

Since $Q(\mathbf{s}, \mathbf{a}) = r(\mathbf{s}, \mathbf{a}) + \gamma \mathbb{E}_{\mathbf{s}' \sim P}[V(\mathbf{s}')]$, we have

$$\|Q\|_\infty \le R + \gamma(\|Q\|_\infty + \log |\mathcal{A}|). \tag{47}$$

Rearranging, we obtain Eq. (44) and hence Eq. (45). $\qquad\square$

**Lemma 9.** *Under MaxEnt RL, we have*

$$\|\log \pi\|_\infty \le \frac{2R + (1+\gamma)\log|\mathcal{A}|}{1-\gamma}. \tag{48}$$

*Proof.* This directly follows Lemma 8 because

$$|\log \pi(\mathbf{a}|\mathbf{s})| = |Q(\mathbf{s}, \mathbf{a}) - V(\mathbf{s})| \le |Q(\mathbf{s}, \mathbf{a})| + |V(\mathbf{s})|.$$

$\square$

**Lemma 10.** *The discounted entropy difference is bounded by*

$$
\left| \mathbb{E}_{\pi_2}\left[ \sum_{t=0}^\infty \gamma^t \log \pi_2(\mathbf{a}_t|\mathbf{s}_t) \right] - \mathbb{E}_{\pi_1}\left[ \sum_{t=0}^\infty \gamma^t \log \pi_1(\mathbf{a}_t|\mathbf{s}_t) \right] \right|
$$
$$
\le \frac{2|\mathcal{A}|\epsilon}{(1-\gamma)^2} + \frac{(2R + (1+\gamma)\log|\mathcal{A}|)|\mathcal{A}|\epsilon}{(1-\gamma)^3} + \frac{(2R + (1+\gamma)\log|\mathcal{A}|)|\mathcal{A}|\gamma\epsilon}{(1-\gamma)^4}. \tag{49}
$$

*Proof.* We express the expected discounted sum as

$$\mathbb{E}_\pi\left[ \sum_{t=0}^\infty \gamma^t \log \pi(\mathbf{a}_t|\mathbf{s}_t) \right] = \sum_{t=0}^\infty \gamma^t \mathbb{E}_{\mathbf{s} \sim d_t^\pi}\left[ \sum_{\mathbf{a}} \pi(\mathbf{a}|\mathbf{s}) \log \pi(\mathbf{a}|\mathbf{s}) \right].$$

Now consider the difference:

$$\sum_{t=0}^\infty \gamma^t \left( \mathbb{E}_{\mathbf{s} \sim d_t^{\pi_2}}\left[ \sum_{\mathbf{a}} \pi_2(\mathbf{a}|\mathbf{s}) \log \pi_2(\mathbf{a}|\mathbf{s}) \right] - \mathbb{E}_{\mathbf{s} \sim d_t^{\pi_1}}\left[ \sum_{\mathbf{a}} \pi_1(\mathbf{a}|\mathbf{s}) \log \pi_1(\mathbf{a}|\mathbf{s}) \right] \right)$$

$$= \sum_{t=0}^\infty \gamma^t \left( \underbrace{\mathbb{E}_{\mathbf{s} \sim d_t^{\pi_2}}\left[ \sum_{\mathbf{a}} (\pi_2(\mathbf{a}|\mathbf{s}) \log \pi_2(\mathbf{a}|\mathbf{s}) - \pi_1(\mathbf{a}|\mathbf{s}) \log \pi_1(\mathbf{a}|\mathbf{s})) \right]}_{\text{(first term)}} \right.$$

$$\left. + \underbrace{\left( \mathbb{E}_{\mathbf{s} \sim d_t^{\pi_2}} - \mathbb{E}_{\mathbf{s} \sim d_t^{\pi_1}} \right)\left[ \sum_{\mathbf{a}} \pi_1(\mathbf{a}|\mathbf{s}) \log \pi_1(\mathbf{a}|\mathbf{s}) \right]}_{\text{(second term)}} \right). \tag{50}$$

We bound each term:

- The first term is bounded by

$$\text{(first term)} \le \mathbb{E}_{\mathbf{s} \sim d_t^{\pi_2}}\big[ \|\pi_2(\cdot|\mathbf{s}) - \pi_1(\cdot|\mathbf{s})\|_1 \cdot \|\log \pi_2\|_\infty +$$
$$\|\log \pi_2(\mathbf{a}|\mathbf{s}) - \log \pi_1(\mathbf{a}|\mathbf{s})\|_1 \big]$$
$$\le \frac{|\mathcal{A}|\epsilon}{1-\gamma} \frac{2R + (1+\gamma)\log|\mathcal{A}|}{1-\gamma} + \frac{2|\mathcal{A}|\epsilon}{1-\gamma} \quad \text{(Lemmas 5, 9, 7)} \tag{51}$$
$$\le \frac{(2R + (1+\gamma)\log|\mathcal{A}|)|\mathcal{A}|\epsilon}{(1-\gamma)^2} + \frac{2|\mathcal{A}|\epsilon}{1-\gamma}.$$

- The second term is bounded by

$$\text{(second term)} \le \|d_t^{\pi_1} - d_t^{\pi_2}\|_1 \cdot |\mathcal{A}| \|\log \pi_1\|_\infty$$

$$\le \frac{t\epsilon}{1-\gamma} \cdot |\mathcal{A}| \cdot \frac{2R + (1+\gamma)\log|\mathcal{A}|}{1-\gamma} \quad \text{(Lemmas 6, 9)} \tag{52}$$

$$\le \frac{(2R + (1+\gamma)\log|\mathcal{A}|)|\mathcal{A}\epsilon}{(1-\gamma)^2} \cdot t.$$

Summing over $t$ and applying $\sum_{t=0}^{\infty} \gamma^t = \frac{1}{1-\gamma}$ and $\sum_{t=0}^{\infty} \gamma^t t = \frac{\gamma}{(1-\gamma)^2}$ completes the proof. $\qquad\square$

**We next prove Theorem 3.**

*Proof.* Substracting $\ell(\boldsymbol{\theta}_{\text{new}})$ from $\ell_{\boldsymbol{\theta}_{\text{old}}}(\boldsymbol{\theta}_{\text{new}})$ gives

$$\ell_{\boldsymbol{\theta}_{\text{old}}}(\boldsymbol{\theta}_{\text{new}}) - \ell(\boldsymbol{\theta}_{\text{new}}) = \mathbb{E}_{\rho^{\pi_E}} \left[ V_{r_{\boldsymbol{\theta}_{\text{new}}}}^{\pi_{\boldsymbol{\theta}_{\text{new}}}}(\mathbf{s}) - V_{r_{\boldsymbol{\theta}_{\text{new}}}}^{\pi_{\boldsymbol{\theta}_{\text{old}}}}(\mathbf{s}) + \underbrace{Q_{r_{\boldsymbol{\theta}_{\text{new}}}}^{\pi_{\boldsymbol{\theta}_{\text{old}}}}(\mathbf{s}, \mathbf{a}) - Q_{r_{\boldsymbol{\theta}_{\text{new}}}}^{\pi_{\boldsymbol{\theta}_{\text{new}}}}(\mathbf{s}, \mathbf{a})}_{\le 0 \text{ because } \pi_{\boldsymbol{\theta}_{\text{new}}} \text{ is optimal to } r_{\boldsymbol{\theta}_{\text{new}}}} \right] \tag{53}$$

$$\le \mathbb{E}_{\mathbf{s} \sim \rho^{\pi_E}} \left[ V_{r_{\boldsymbol{\theta}_{\text{new}}}}^{\pi_{\boldsymbol{\theta}_{\text{new}}}}(\mathbf{s}) - V_{r_{\boldsymbol{\theta}_{\text{new}}}}^{\pi_{\boldsymbol{\theta}_{\text{old}}}}(\mathbf{s}) \right].$$

To bound $\ell_{\boldsymbol{\theta}_{\text{old}}}(\boldsymbol{\theta}_{\text{new}}) - \ell(\boldsymbol{\theta}_{\text{new}})$, it is suffices to bound $V_{r_{\boldsymbol{\theta}_{\text{new}}}}^{\pi_{\boldsymbol{\theta}_{\text{new}}}}(\mathbf{s}) - V_{r_{\boldsymbol{\theta}_{\text{new}}}}^{\pi_{\boldsymbol{\theta}_{\text{old}}}}(\mathbf{s})$. To do so, let us first investigate the definition of $V_{r_{\boldsymbol{\theta}}}^{\pi_{\boldsymbol{\theta}}}(\mathbf{s})$ with $\pi_{\boldsymbol{\theta}}$ optimal to $r_{\boldsymbol{\theta}}$:

$$V_r^{\pi}(\mathbf{s}) = \mathbb{E}_{\pi} \left[ \sum_{t=0}^{\infty} \gamma^t \left( r(\mathbf{s}_t, \mathbf{a}_t) - \log \pi(\mathbf{a}_t|\mathbf{s}_t) \right) \Big| \mathbf{s}_0 = \mathbf{s} \right],$$

which indicates that the value function $V_{r_{\boldsymbol{\theta}}}^{\pi_{\boldsymbol{\theta}}}(\mathbf{s})$ can be split into two terms:

1. **Reward term:** $\mathbb{E}_{\pi_{\boldsymbol{\theta}}} \left[ \sum_{t=0}^{\infty} \gamma^t r_{\boldsymbol{\theta}}(\mathbf{s}_t, \mathbf{a}_t) \right]$.

2. **Entropy term:** $-\mathbb{E}_{\pi_{\boldsymbol{\theta}}} \left[ \sum_{t=0}^{\infty} \gamma^t \log \pi_{\boldsymbol{\theta}}(\mathbf{a}_t|\mathbf{s}_t) \right]$.

Thus, we can decompose $V_{r_{\boldsymbol{\theta}_{\text{new}}}}^{\pi_{\boldsymbol{\theta}_{\text{new}}}}(\mathbf{s}) - V_{r_{\boldsymbol{\theta}_{\text{new}}}}^{\pi_{\boldsymbol{\theta}_{\text{old}}}}(\mathbf{s})$ into two terms:

$$V_{r_{\boldsymbol{\theta}_{\text{new}}}}^{\pi_{\boldsymbol{\theta}_{\text{new}}}}(\mathbf{s}) - V_{r_{\boldsymbol{\theta}_{\text{new}}}}^{\pi_{\boldsymbol{\theta}_{\text{old}}}}(\mathbf{s}) = \Delta_{\text{reward}}(\mathbf{s}) + \Delta_{\text{entropy}}(\mathbf{s}), \tag{54}$$

where

$$\Delta_{\text{reward}}(\mathbf{s}) := \mathbb{E}_{\pi_{\boldsymbol{\theta}_{\text{new}}}} \left[ \sum_{t=0}^{\infty} \gamma^t r_{\boldsymbol{\theta}_{\text{new}}}(\mathbf{s}_t, \mathbf{a}_t) \Big| \mathbf{s}_0 = \mathbf{s} \right] - \mathbb{E}_{\pi_{\boldsymbol{\theta}_{\text{old}}}} \left[ \sum_{t=0}^{\infty} \gamma^t r_{\boldsymbol{\theta}_{\text{new}}}(\mathbf{s}_t, \mathbf{a}_t) \Big| \mathbf{s}_0 = \mathbf{s} \right], \tag{55}$$

$$\Delta_{\text{entropy}}(\mathbf{s}) := \mathbb{E}_{\pi_{\boldsymbol{\theta}_{\text{old}}}} \left[ \sum_{t=0}^{\infty} \gamma^t \log \pi_{\boldsymbol{\theta}_{\text{old}}}(\mathbf{a}_t|\mathbf{s}_t) \Big| \mathbf{s}_0 = \mathbf{s} \right] - \mathbb{E}_{\pi_{\boldsymbol{\theta}_{\text{new}}}} \left[ \sum_{t=0}^{\infty} \gamma^t \log \pi_{\boldsymbol{\theta}_{\text{new}}}(\mathbf{a}_t|\mathbf{s}_t) \Big| \mathbf{s}_0 = \mathbf{s} \right]. \tag{56}$$

We first bound the term $\Delta_{\text{reward}}(\mathbf{s})$:

$$\Delta_{\text{reward}}(\mathbf{s})$$

$$= \sum_{t=0}^{\infty} \gamma^t \left( \sum_{\mathbf{s}} d_t^{\pi_{\boldsymbol{\theta}_{\text{new}}}}(\mathbf{s}) \sum_{\mathbf{a}} \pi_{\boldsymbol{\theta}_{\text{new}}}(\mathbf{a}|\mathbf{s}) r_{\boldsymbol{\theta}_{\text{new}}}(\mathbf{s}, \mathbf{a}) - \sum_{\mathbf{s}} d_t^{\pi_{\boldsymbol{\theta}_{\text{old}}}}(\mathbf{s}) \sum_{\mathbf{a}} \pi_{\boldsymbol{\theta}_{\text{old}}}(\mathbf{a}|\mathbf{s}) r_{\boldsymbol{\theta}_{\text{new}}}(\mathbf{s}, \mathbf{a}) \right)$$

$$\leq \sum_{t=0}^{\infty} \gamma^t \left| \sum_{\mathbf{s}} d_t^{\pi_{\boldsymbol{\theta}_{\text{new}}}}(\mathbf{s}) \left( \sum_{\mathbf{a}} \left( \pi_{\boldsymbol{\theta}_{\text{new}}}(\mathbf{a}|\mathbf{s}) - \pi_{\boldsymbol{\theta}_{\text{old}}}(\mathbf{a}|\mathbf{s}) \right) r_{\boldsymbol{\theta}_{\text{new}}}(\mathbf{s}, \mathbf{a}) \right) \right| +$$

$$\sum_{t=0}^{\infty} \gamma^t \left| \sum_{\mathbf{s}} \left( d_t^{\pi_{\boldsymbol{\theta}_{\text{new}}}}(\mathbf{s}) - d_t^{\pi_{\boldsymbol{\theta}_{\text{old}}}}(\mathbf{s}) \right) \sum_{\mathbf{a}} \pi_{\boldsymbol{\theta}_{\text{old}}}(\mathbf{a}|\mathbf{s}) r_{\boldsymbol{\theta}_{\text{new}}}(\mathbf{s}, \mathbf{a}) \right| \quad \text{(triangle inequality)}$$

$$\leq \sum_{t=0}^{\infty} \gamma^t \sum_{\mathbf{s}} d_t^{\pi_{\boldsymbol{\theta}_{\text{new}}}}(\mathbf{s}) \cdot 2R D_{\text{TV}}(\pi_{\boldsymbol{\theta}_{\text{new}}}(\cdot|\mathbf{s}), \pi_{\boldsymbol{\theta}_{\text{old}}}(\cdot|\mathbf{s})) + \sum_{t=0}^{\infty} \gamma^t R \| d_t^{\pi_{\boldsymbol{\theta}_{\text{new}}}} - d_t^{\pi_{\boldsymbol{\theta}_{\text{old}}}} \|_1 \quad \text{(Lemma 6)}$$

$$\leq \sum_{t=0}^{\infty} \gamma^t \left( 2R D_{\text{TV}}^{\max}(\pi_{\boldsymbol{\theta}_{\text{new}}}, \pi_{\boldsymbol{\theta}_{\text{old}}}) + 2t R D_{\text{TV}}^{\max}(\pi_{\boldsymbol{\theta}_{\text{new}}}, \pi_{\boldsymbol{\theta}_{\text{old}}}) \right)$$

$$\leq \sum_{t=0}^{\infty} \gamma^t \left( 2(t+1) R D_{\text{TV}}^{\max}(\pi_{\boldsymbol{\theta}_{\text{new}}}, \pi_{\boldsymbol{\theta}_{\text{old}}}) \right)$$

$$\leq \sum_{t=0}^{\infty} \gamma^t \left( (t+1) \frac{R|\mathcal{A}| \epsilon_{\boldsymbol{\theta}_{\text{old}}}(\boldsymbol{\theta}_{\text{new}})}{1 - \gamma} \right)$$

$$= \frac{R|\mathcal{A}| \epsilon_{\boldsymbol{\theta}_{\text{old}}}(\boldsymbol{\theta}_{\text{new}})}{1 - \gamma} \left( \sum_{t=0}^{\infty} \gamma^t t + \sum_{t=0}^{\infty} \gamma^t \right)$$

$$= \frac{R|\mathcal{A}| \epsilon_{\boldsymbol{\theta}_{\text{old}}}(\boldsymbol{\theta}_{\text{new}})}{1 - \gamma} \left( \frac{\gamma}{(1 - \gamma)^2} + \frac{1}{1 - \gamma} \right)$$

$$= \frac{R|\mathcal{A}| \epsilon_{\boldsymbol{\theta}_{\text{old}}}(\boldsymbol{\theta}_{\text{new}})}{(1 - \gamma)^3}. \tag{57}$$

The bound of the term $\Delta_{\text{entropy}}(\mathbf{s})$ directly follows Lemma 10:

$$\Delta_{\text{entropy}}(\mathbf{s}) \leq \left( \frac{2|\mathcal{A}|}{(1 - \gamma)^2} + \frac{(2R + (1 + \gamma) \log |\mathcal{A}|)|\mathcal{A}|}{(1 - \gamma)^3} \right.$$

$$\left. + \frac{(2R + (1 + \gamma) \log |\mathcal{A}|)|\mathcal{A}| \gamma}{(1 - \gamma)^4} \right) \epsilon_{\boldsymbol{\theta}_{\text{old}}}(\boldsymbol{\theta}_{\text{new}}). \tag{58}$$

Finally, combining Eq. (57) and Eq. (58), we complete the proof by

$$\ell_{\boldsymbol{\theta}_{\text{old}}}(\boldsymbol{\theta}_{\text{new}}) - \ell(\boldsymbol{\theta}_{\text{new}})$$

$$\leq \mathbb{E}_{\mathbf{s} \sim \rho^{\pi_E}} \left[ V_{r_{\boldsymbol{\theta}_{\text{new}}}}^{\pi_{\boldsymbol{\theta}_{\text{new}}}}(\mathbf{s}) - V_{r_{\boldsymbol{\theta}_{\text{new}}}}^{\pi_{\boldsymbol{\theta}_{\text{old}}}}(\mathbf{s}) \right]$$

$$\leq \Delta_{\text{reward}}(\mathbf{s}) + \Delta_{\text{entropy}}(\mathbf{s}) \quad \text{(Eq. (58)+Eq. (57))} \tag{59}$$

$$= \left( \frac{2|\mathcal{A}|}{(1 - \gamma)^2} + \frac{(5 - \gamma)R|\mathcal{A}| + (\gamma - \gamma^2 + 2)|\mathcal{A}| \log |\mathcal{A}|}{(1 - \gamma)^4} \right) \epsilon_{\boldsymbol{\theta}_{\text{old}}}(\boldsymbol{\theta}_{\text{new}}).$$

$\square$

# B  A Unified View of Non-adversarial IRL

Let $\text{Cov}_{p(\mathbf{x})}(\kappa_1(\mathbf{x}), \kappa_2(\mathbf{x})) := \mathbb{E}_{p(\mathbf{x})}[\kappa_1(\mathbf{x}) \cdot \kappa_2(\mathbf{x})] - \mathbb{E}_{p(\mathbf{x})}[\kappa_1(\mathbf{x})] \cdot \mathbb{E}_{p(\mathbf{x})}[\kappa_2(\mathbf{x})]$ denote the covariance of two functions $\kappa_1(\mathbf{x}), \kappa_2(\mathbf{x})$ under the distribution $p(\mathbf{x})$. We first show an equivalent expression of $\ell(\boldsymbol{\theta})$.

**Lemma 11.** *The likelihood objective has the following equivalent expression:*

$$\ell(\boldsymbol{\theta}) = \mathbb{E}_{\rho^{\pi_E}}[r_{\boldsymbol{\theta}}(\mathbf{s}, \mathbf{a})] - \mathbb{E}_{\rho^{\pi_{\boldsymbol{\theta}}}}[r_{\boldsymbol{\theta}}(\mathbf{s}, \mathbf{a})]$$

$$= \mathbb{E}_{\rho^{\pi_{\boldsymbol{\theta}}}}\left[\frac{\rho^{\pi_E}(\mathbf{s}, \mathbf{a})}{\rho^{\pi_{\boldsymbol{\theta}}}(\mathbf{s}, \mathbf{a})} r_{\boldsymbol{\theta}}(\mathbf{s}, \mathbf{a})\right] - \underbrace{\mathbb{E}_{\rho^{\pi_{\boldsymbol{\theta}}}}\left[\frac{\rho^{\pi_E}(\mathbf{s}, \mathbf{a})}{\rho^{\pi_{\boldsymbol{\theta}}}(\mathbf{s}, \mathbf{a})}\right]}_{\equiv 1} \times \mathbb{E}_{\rho^{\pi_{\boldsymbol{\theta}}}}[r_{\boldsymbol{\theta}}(\mathbf{s}, \mathbf{a})] \tag{60}$$

$$= \operatorname{Cov}_{\rho^{\pi_{\boldsymbol{\theta}}}(\mathbf{s}, \mathbf{a})}\left(\frac{\rho^{\pi_E}(\mathbf{s}, \mathbf{a})}{\rho^{\pi_{\boldsymbol{\theta}}}(\mathbf{s}, \mathbf{a})}, \nabla_{\boldsymbol{\theta}} r_{\boldsymbol{\theta}}(\mathbf{s}, \mathbf{a})\right).$$

We next show that the KL-based $f$-IRL [30] essentially maximizes the likelihood of expert demonstrations (minimize the imitation gap). Recall from the main text that $f$-IRL assumes a state-only reward function, $r_{\boldsymbol{\theta}}(\mathbf{s})$, and seeks to match the expert's state marginal distribution by minimizing an $f$-divergence objective:

$$L_f(\boldsymbol{\theta}) \coloneqq D_f(\rho^{\pi_E}(\mathbf{s}) \| \rho^{\pi_{\boldsymbol{\theta}}}(\mathbf{s})), \tag{61}$$

where $\rho^{\pi}(\mathbf{s})$ denotes the state marginal of the occupancy measure such that $\rho^{\pi}(\mathbf{s}, \mathbf{a}) = \rho^{\pi}(\mathbf{s})\pi(\mathbf{a}|\mathbf{s})$. It has been shown in [30, Appendix A2] that if $D_f$ is taken as the KL divergence, then $\nabla_{\boldsymbol{\theta}} L_f(\boldsymbol{\theta})$ can be reduced to the following analytical form:

$$\nabla_{\boldsymbol{\theta}} L_f(\boldsymbol{\theta}) = \mathbb{E}_{\tau \sim \rho^{\pi_{\boldsymbol{\theta}}}(\tau)}\left[\mathbb{E}_{\mathbf{s} \sim \tau}\left[\frac{\rho^{\pi_E}(\mathbf{s})}{\rho^{\pi_{\boldsymbol{\theta}}}(\mathbf{s})}\right] \mathbb{E}_{\mathbf{s} \sim \tau}[r_{\boldsymbol{\theta}}(\mathbf{s})]\right] -$$

$$\mathbb{E}_{\tau \sim \rho^{\pi_{\boldsymbol{\theta}}}(\tau)}\left[\mathbb{E}_{\mathbf{s} \sim \tau}\left[\frac{\rho^{\pi_E}(\mathbf{s})}{\rho^{\pi_{\boldsymbol{\theta}}}(\mathbf{s})}\right]\right] \times \mathbb{E}_{\tau \sim \rho^{\pi_{\boldsymbol{\theta}}}(\tau)}[\mathbb{E}_{\mathbf{s} \sim \tau}[r_{\boldsymbol{\theta}}(\mathbf{s})]], \tag{62}$$

where $\rho^{\pi_{\boldsymbol{\theta}}}(\tau)$ denotes the trajectory distribution under the reward $r_{\boldsymbol{\theta}}$ and $\mathbb{E}_{\mathbf{s} \sim \tau}[\cdot]$ denotes the expectation w.r.t. states over the cumulative state visitation frequency determined by a given trajectory. To show that KL-based $f$-IRL essentially maximizes $\ell(\boldsymbol{\theta})$, it suffices to show that Eq. (62) is propotional to Eq. (60). To proceed, we first notice that $\mathbb{E}_{\tau \sim \rho^{\pi_{\boldsymbol{\theta}}}(\tau)}[\mathbb{E}_{\mathbf{s} \sim \tau}[\cdot]] \equiv (1 - \gamma)\mathbb{E}_{\mathbf{s} \sim \rho^{\pi_{\boldsymbol{\theta}}}(\mathbf{s})}[\cdot]$ as both represent the state marginal of the occupancy measure. Given this equivalence relationship, we can reduce the second term in the right-hand side of Eq. (62) to the following term:

$$\mathbb{E}_{\tau \sim \rho^{\pi_{\boldsymbol{\theta}}}(\tau)}\left[\mathbb{E}_{\mathbf{s} \sim \tau}\left[\frac{\rho^E(\mathbf{s})}{\rho^{\pi_{\boldsymbol{\theta}}}(\mathbf{s})}\right]\right] \times \mathbb{E}_{\tau \sim \rho^{\pi_{\boldsymbol{\theta}}}(\tau)}[\mathbb{E}_{\mathbf{s} \sim \tau}[r_{\boldsymbol{\theta}}(\mathbf{s})]]$$

$$= (1 - \gamma)\mathbb{E}_{\mathbf{s} \sim \rho^{\pi_{\boldsymbol{\theta}}}(\mathbf{s})}\left[\frac{\rho^E(\mathbf{s})}{\rho^{\pi_{\boldsymbol{\theta}}}(\mathbf{s})}\right] \times \mathbb{E}_{\mathbf{s} \sim \rho^{\pi_{\boldsymbol{\theta}}}(\mathbf{s})}[r_{\boldsymbol{\theta}}(\mathbf{s})] \tag{63}$$

$$= (1 - \gamma)\mathbb{E}_{\mathbf{s} \sim \rho^{\pi_{\boldsymbol{\theta}}}(\mathbf{s})}[r_{\boldsymbol{\theta}}(\mathbf{s})].$$

We next investigate the first term in the right-hand side of Eq. (62):

$$\mathbb{E}_{\tau \sim \rho^{\pi_{\boldsymbol{\theta}}}(\tau)}\left[\mathbb{E}_{\mathbf{s} \sim \tau}\left[\frac{\rho^{\pi_E}(\mathbf{s})}{\rho^{\pi_{\boldsymbol{\theta}}}(\mathbf{s})}\right] \mathbb{E}_{\mathbf{s} \sim \tau}[r_{\boldsymbol{\theta}}(\mathbf{s})]\right]$$

$$= \mathbb{E}_{\tau \sim \rho^{\pi_{\boldsymbol{\theta}}}(\tau)}\left[-\operatorname{Cov}_{\tau}\left(\frac{\rho^{\pi_E}(\mathbf{s})}{\rho^{\pi_{\boldsymbol{\theta}}}(\mathbf{s})}, \nabla_{\boldsymbol{\theta}} r_{\boldsymbol{\theta}}(\mathbf{s})\right) + \mathbb{E}_{\mathbf{s} \sim \tau}\left[\frac{\rho_E(\mathbf{s})}{\rho^{\pi_{\boldsymbol{\theta}}}(\mathbf{s})}\nabla_{\boldsymbol{\theta}} r_{\boldsymbol{\theta}}(\mathbf{s})\right]\right]$$

$$= \mathbb{E}_{\tau \sim \rho^{\pi_{\boldsymbol{\theta}}}(\tau)}\left[-\operatorname{Cov}_{\tau}\left(\frac{\rho^{\pi_E}(\mathbf{s})}{\rho^{\pi_{\boldsymbol{\theta}}}(\mathbf{s})}, \nabla_{\boldsymbol{\theta}} r_{\boldsymbol{\theta}}(\mathbf{s})\right)\right] + \mathbb{E}_{\tau \sim \rho^{\pi_{\boldsymbol{\theta}}}(\tau)}\left[\mathbb{E}_{\mathbf{s} \sim \tau}\left[\frac{\rho^{\pi_E}(\mathbf{s})}{\rho^{\pi_{\boldsymbol{\theta}}}(\mathbf{s})}\nabla_{\boldsymbol{\theta}} r_{\boldsymbol{\theta}}(\mathbf{s})\right]\right] \tag{64}$$

$$= -\operatorname{Cov}_{\rho^{\pi_{\boldsymbol{\theta}}}(\mathbf{s})}\left(\frac{\rho^{\pi_E}(\mathbf{s})}{\rho^{\pi_{\boldsymbol{\theta}}}(\mathbf{s})}, \nabla_{\boldsymbol{\theta}} r_{\boldsymbol{\theta}}(\mathbf{s})\right) + (1 - \gamma)\mathbb{E}_{\rho^{\pi_E}(\mathbf{s})}[\nabla_{\boldsymbol{\theta}} r_{\boldsymbol{\theta}}(\mathbf{s})].$$

Combining Eq. (63) and Eq. (64), we have

$$\nabla_{\boldsymbol{\theta}} L_f(\boldsymbol{\theta}) = -\operatorname{Cov}_{\rho^{\pi_{\boldsymbol{\theta}}}(\mathbf{s})}\left(\frac{\rho^{\pi_E}(\mathbf{s})}{\rho^{\pi_{\boldsymbol{\theta}}}(\mathbf{s})}, \nabla_{\boldsymbol{\theta}} r_{\boldsymbol{\theta}}(\mathbf{s})\right) + (1 - \gamma)\mathbb{E}_{\rho^{\pi_E}(\mathbf{s})}[\nabla_{\boldsymbol{\theta}} r_{\boldsymbol{\theta}}(\mathbf{s})] - (1 - \gamma)\mathbb{E}_{\mathbf{s} \sim \rho^{\pi_{\boldsymbol{\theta}}}(\mathbf{s})}[r_{\boldsymbol{\theta}}(\mathbf{s})]$$

$$= -\operatorname{Cov}_{\rho^{\pi_{\boldsymbol{\theta}}}(\mathbf{s})}\left(\frac{\rho^{\pi_E}(\mathbf{s})}{\rho^{\pi_{\boldsymbol{\theta}}}(\mathbf{s})}, \nabla_{\boldsymbol{\theta}} r_{\boldsymbol{\theta}}(\mathbf{s})\right) + (1 - \gamma)\operatorname{Cov}_{\rho^{\pi_{\boldsymbol{\theta}}}(\mathbf{s})}\left(\frac{\rho^{\pi_E}(\mathbf{s})}{\rho^{\pi_{\boldsymbol{\theta}}}(\mathbf{s})}, \nabla_{\boldsymbol{\theta}} r_{\boldsymbol{\theta}}(\mathbf{s})\right)$$

$$= -\gamma\operatorname{Cov}_{\rho^{\pi_{\boldsymbol{\theta}}}(\mathbf{s})}\left(\frac{\rho^{\pi_E}(\mathbf{s})}{\rho^{\pi_{\boldsymbol{\theta}}}(\mathbf{s})}, \nabla_{\boldsymbol{\theta}} r_{\boldsymbol{\theta}}(\mathbf{s})\right). \tag{65}$$

Therefore, if the reward is state-only, i.e., $r_{\boldsymbol{\theta}}(\mathbf{s})$, we have $\nabla_{\boldsymbol{\theta}} L_f(\boldsymbol{\theta}) \propto -\nabla_{\boldsymbol{\theta}} \ell(\boldsymbol{\theta})$. This completes the proof for Eq. (6b) in the main text.

## C   Detailed Experimental Setup

### C.1   Experimental Setup for PIRO

Training procedure is given in Alg. 3. Network architecture and hyperparameter setup for each task are listed in Tab. 2 and Tab. 3.

Table 2: Network architecture and hyperparameter setup for MuJoCo tasks.

|  | Hopper | Walker2D | Ant | Humanoid | Cheetah |
|---|---|---|---|---|---|
| Expert demo. (**s**-**a** pairs) | 1000 | 1000 | 1000 | 1000 | 1000 |
| Reward network (hidden layers) | 128, 128 | 128, 128 | 128, 128 | 128, 128 | 128, 128 |
| Batch size (**s**-**a** pairs) | 256 | 256 | 256 | 256 | 256 |
| Reward learning rate | 1e-4 | 1e-4 | 1e-4 | 1e-4 | 1e-4 |
| SAC epochs per iteration | 5 | 5 | 5 | 5 | 5 |
| Entropy coefficient $\alpha$ | 0.2 | 0.2 | 0.2 | 0.2 | 0.2 |
| Threshold $\bar{\epsilon}^{\text{target}}$ | 0.5 | 0.5 | 0.5 | 0.5 | 0.5 |
| Scaling factor $x_\epsilon$ for $\bar{\epsilon}$ | 1.5 | 1.5 | 1.5 | 1.5 | 1.5 |
| Scaling factor $y_\epsilon$ for $\bar{\epsilon}$ | 1.5 | 1.5 | 1.5 | 1.5 | 1.5 |
| SAC rounds per iteration ($k$) | 1 | 1 | 1 | 1 | 1 |
| Reward gradient steps per iteration ($n$) | 1 | 1 | 1 | 1 | 1 |

Table 3: Network architecture and hyperparameter setup for AntMaze and Adroit tasks.

|  | AntMaze-U | AntMaze-M | AntMaze-L | HandPen |
|---|---|---|---|---|
| Expert demo. (**s**-**a** pairs) | 700 | 1000 | 1000 | 2000 |
| Reward network (hidden layers) | 128, 128 | 128, 128 | 128, 128 | 256, 256 |
| Batch size (**s**-**a** pairs) | 256 | 256 | 256 | 256 |
| Reward learning rate | 1e-4 | 1e-4 | 1e-4 | 3e-5 |
| SAC epochs per iteration | 5 | 5 | 5 | 5 |
| Entropy coefficient $\alpha$ | 0.2 | 0.2 | 0.2 | 0.2 |
| Threshold $\bar{\epsilon}^{\text{target}}$ | 0.5 | 0.5 | 0.5 | 0.5 |
| Scaling factor $x_\epsilon$ for $\bar{\epsilon}$ | 1.5 | 1.5 | 1.5 | 1.5 |
| Scaling factor $y_\epsilon$ for $\bar{\epsilon}$ | 1.5 | 1.5 | 1.5 | 1.5 |
| SAC rounds per iteration ($k$) | 1 | 1 | 1 | 1 |
| Reward gradient steps per iteration ($n$) | 1 | 1 | 1 | 1 |

### C.2   Pre-trained Expert Policy Model and Expert Demonstrations

The sources of pre-trained policy models or offline trajectory datasets for experts are provided in Tab. 4. In MuJoCo tasks, we use these high-quality pre-trained policy models to sample expert demonstrations. In Robotic tasks, we directly use the expert trajectories from the Minari Offline Reinforcement Learning datasets [46].

Table 4: The sources of expert policies or demonstrations.

| Task | Source |
|---|---|
| MuJoCo Tasks | Same as expert policies used in $f$-IRL [30] and ML-IRL [50] |
| UMazeDense | https://minari.farama.org/datasets/D4RL/antmaze/umaze-v1/ |
| MediumDense | https://minari.farama.org/datasets/D4RL/antmaze/medium-play-v1/ |
| LargeDense | https://minari.farama.org/datasets/D4RL/antmaze/large-play-v1/ |
| AdroitHandPen | https://minari.farama.org/datasets/D4RL/pen/human-v2/ |

# D   Additional Experimental Results

## D.1   Hardware Information

Hardware specifications are provided in Tab. 5.

Table 5: Hardware configuration used in experiments.

| Hardware | Specifications |
| --- | --- |
| CPU | AMD EPYC 7713 64-Core Processor @ 2 GHz |
| GPU | NVIDIA A100-SXM4-80GB @ 1215 MHz |
| Memory | 2 TB |

## D.2   Comparison Between Theoretical and Adaptive $C$ on CartPole

To further validate our theoretical analysis, we conduct an additional experiment on `CartPole`, where $|\mathcal{A}| = 2$, $R = 1$, and $\gamma = 0.9$. According to Eq. (7), we have that the exact theoretical value $C \approx 111{,}373.55$.

We compare this theoretical $C$ against the adaptive $C$ method (bounded in $[0.001, 10]$). As shown in Figure 8, the adaptive method substantially reduces KL divergence throughout training (mean $226.3$ vs. $648.9$) while also achieving significantly higher final rewards, both undiscounted ($313.6$ vs. $19.1$) and discounted ($10.0$ vs. $7.1$). When using theoretical $C$, the reward performance does improve within the acceptable training range, but the progress is neither as fast nor as stable as with the adaptive $C$.

These results highlight the practical benefit of adaptively adjusting $C$ during training, despite the theoretical guarantees provided by the closed-form expression. In particular, adaptive $C$ allows stable and sample-efficient learning while avoiding the instability caused by the overly large theoretical constant.

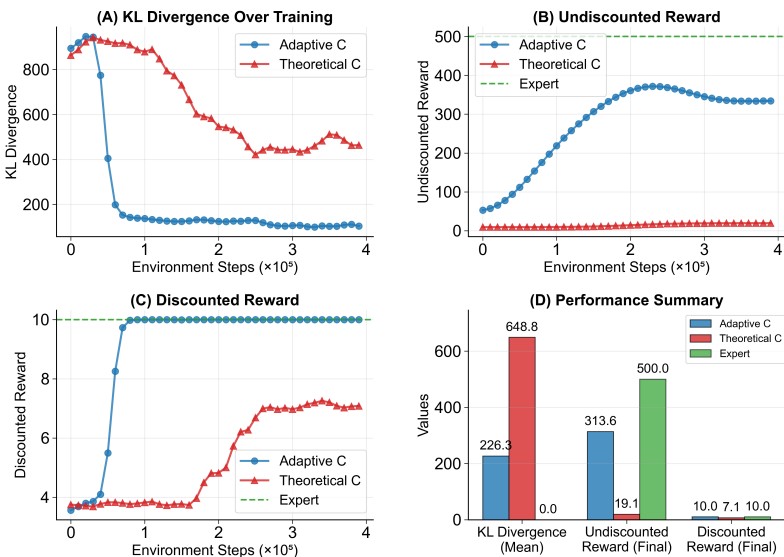

Figure 8: **Comparison between theoretical and adaptive $C$ on** `CartPole`.

# E   A Real-World Case Study: Meerkat Behavior Modeling

## E.1   Dataset Details

As a real-world case study, we apply PIRO to an animal behavior modeling task using a dataset of twenty 12-minute annotated videos capturing the spatial-temporal actions of a meerkat mob in a zoo

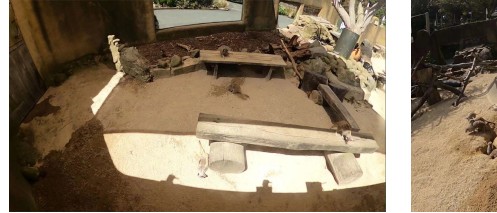
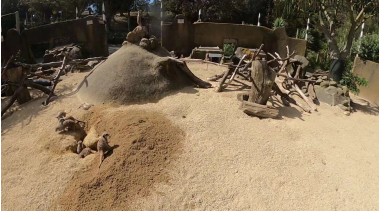

Camera view of the entrance and foraging area

Camera view of the mound and backside of the enclosure

Figure 9: Example images of the camera views.

habitat [34]. To obtain the meerkat behavior, Rogers et al. [34] used two GoPro Max cameras set on the back wall of the meerkat enclosure, focusing on two hubs of activity (Fig. 9). The current zone, coordinates, and behavior of every visible meerkat are labeled for every timestep. Fig. 10 illustrates the full set of behaviors. In addition, each meerkat is identified by a unique identifier during a sequence, keeping track of the same individuals. The heatmap of meerkat's activity is shown in Fig. 11 and the region division for each camera is shown in Fig. 12.

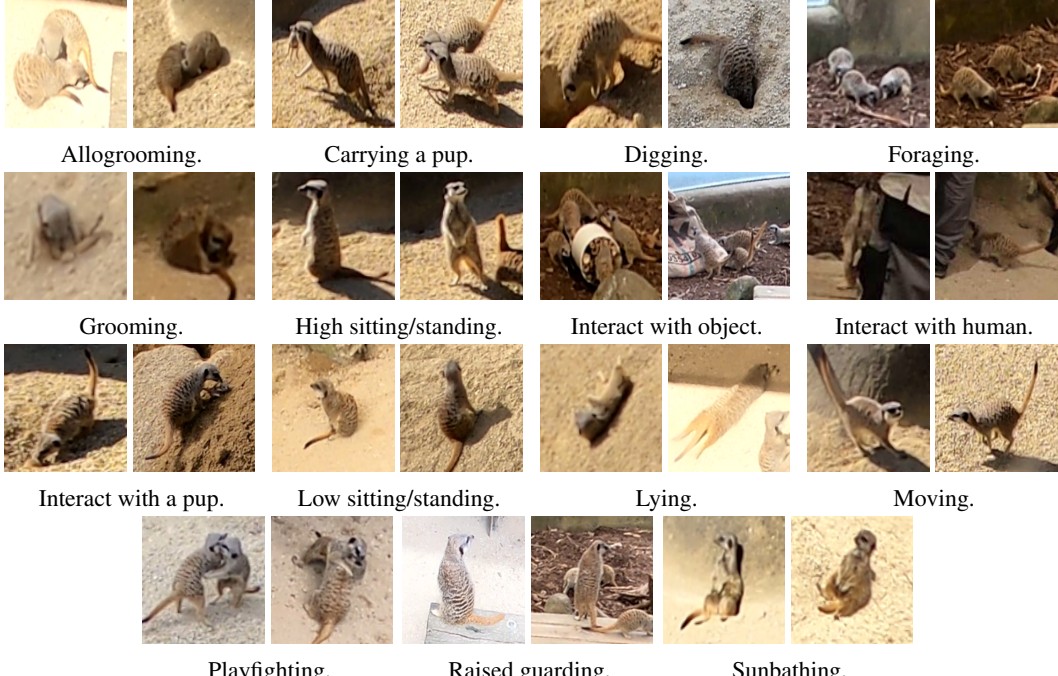

Allogrooming.      Carrying a pup.      Digging.      Foraging.

Grooming.    High sitting/standing.    Interact with object.    Interact with human.

Interact with a pup.    Low sitting/standing.    Lying.    Moving.

Playfighting.      Raised guarding.      Sunbathing.

Figure 10: Fifteen types of the meerkat behaviors.

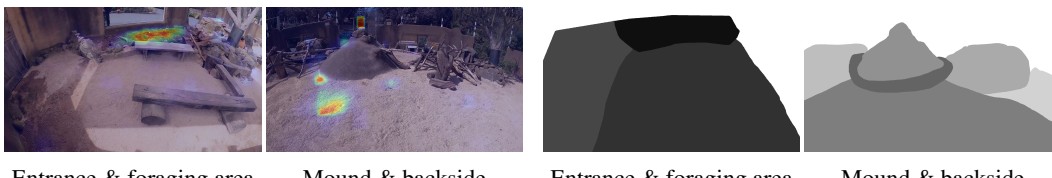

Entrance & foraging area    Mound & backside      Entrance & foraging area    Mound & backside

Figure 11: The frequency of meerkat activity in various regions corresponds to the heatmap from the camera perspective. The areas where meerkats are frequently active are highlighted.

Figure 12: Different colors are labelled for each area to visually illustrate the division of meerkat activity zones.

## E.2  Experimental Results for Policy Divergence Reduction

The dataset includes 25 discrete actions (15 behaviors + 10 actions the represent moving between zones in the habitat) and state representations based on zones (10 total) and social context (counts of close and distant neighbors). The goal is to learn a behavior model that predicts the actions of an individual meerkat, assuming a shared policy across individuals [16]. We extract independent demonstration trajectories of 30 consecutive transitions per individual.

Since *ground-truth rewards* are unavailable in this real-world setting, we evaluate policy imitation using frequencies of transition across habitat zones. Visualizations of the expert's frequencies, PIRO's outputs and baseline results are provided in Fig. 13.

PIRO consistently outperforms baselines in learning stability, as reflected in its lowest error rate. AIRL and IQ-Learn also demonstrate low errors, but these errors remain noticeably higher compared to PIRO. This highlights PIRO's capability to reproduce meerkat trajectories with high similarity.

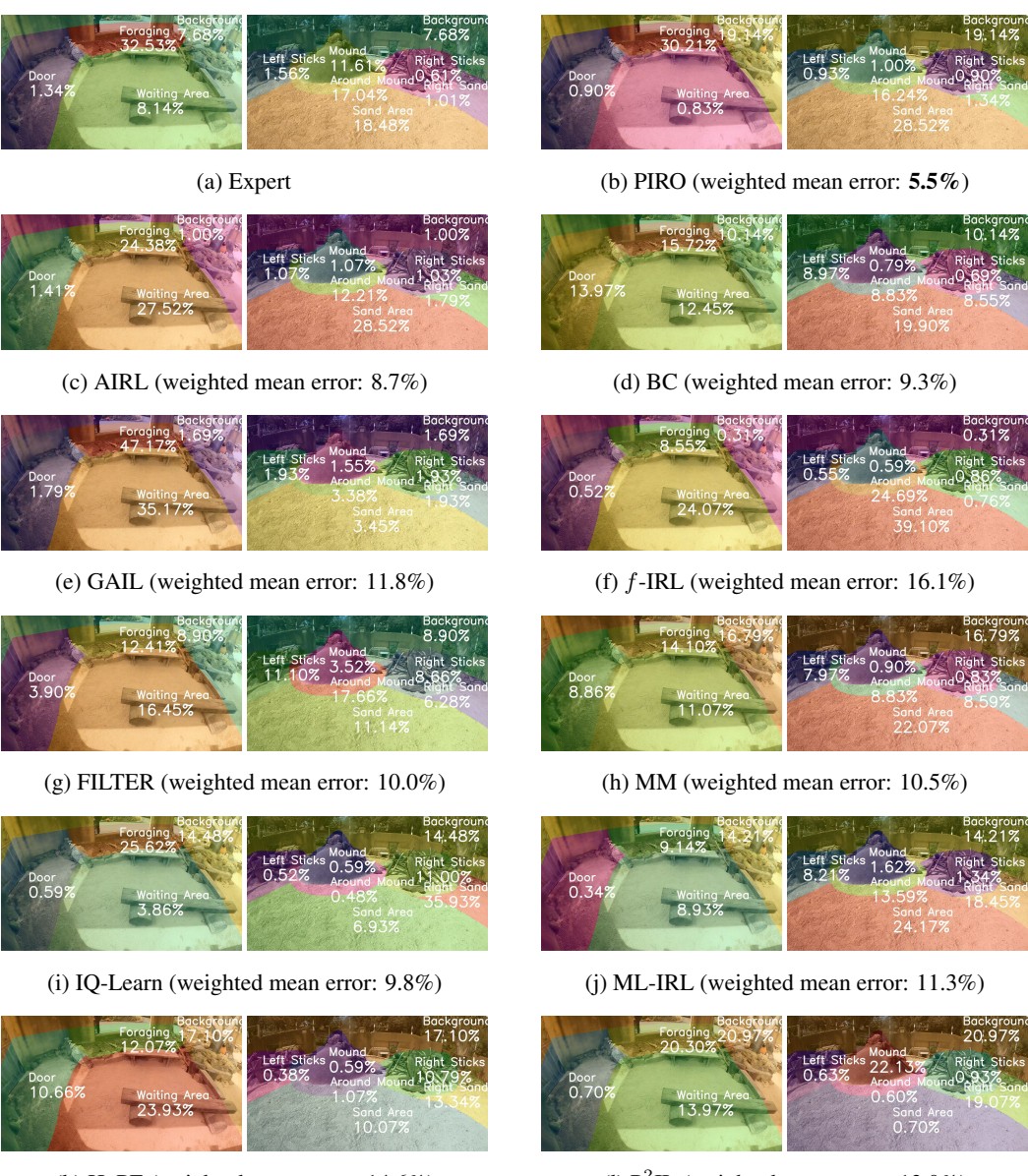

(a) Expert

(b) PIRO (weighted mean error: **5.5%**)

(c) AIRL (weighted mean error: 8.7%)

(d) BC (weighted mean error: 9.3%)

(e) GAIL (weighted mean error: 11.8%)

(f) $f$-IRL (weighted mean error: 16.1%)

(g) FILTER (weighted mean error: 10.0%)

(h) MM (weighted mean error: 10.5%)

(i) IQ-Learn (weighted mean error: 9.8%)

(j) ML-IRL (weighted mean error: 11.3%)

(k) HyPE (weighted mean error: 14.6%)

(l) P$^2$IL (weighted mean error: 13.0%)

Figure 13: **Regional visitation frequency map** generated by analyzing real meerkat trajectories alongside those produced by algorithms. PIRO achieves the lowest weighted mean error.

