# OpenReview forum: "Trust Region Reward Optimization and Proximal Inverse Reward Optimization Algorithm"
_NeurIPS.cc/2025/Conference — NeurIPS 2025 poster_

### Official Review · Reviewer_QmGm · 2025-06-22

**Clarity:** 3
**Significance:** 3
**Originality:** 3
**Rating:** 4
**Confidence:** 3

**Summary:**

The paper titled "PIRO: Toward Stable Reward Learning for Inverse RL via Monotonic Policy Divergence Reduction" introduces a novel approach to inverse reinforcement learning (IRL) that focuses on stabilizing reward learning by ensuring monotonic reduction in policy divergence. The authors propose Proximal Inverse Reward Optimization (PIRO), a non-adversarial IRL algorithm that leverages a theoretical framework to guarantee stable reward updates. PIRO explicitly learns reward functions and scales to high-dimensional tasks while maintaining strong sample efficiency. Empirical evaluations across various OpenAI Gym tasks and a real-world animal behavior modeling task demonstrate that PIRO significantly improves training stability and matches or exceeds the performance of state-of-the-art IRL methods. The paper concludes by discussing the limitations of PIRO, such as its dependency on on-policy trajectory sampling, and suggests future work to address these limitations and further enhance the robustness and efficiency of IRL algorithms.

**Questions:**

1. **How does PIRO handle the trade-off between stability and sample efficiency in on-policy rollouts?**
   The paper mentions that PIRO trades off on-policy sample efficiency for stability. Can the authors elaborate on potential strategies to mitigate this trade-off, such as using techniques like resets to expert states or more efficient sampling methods, to improve the overall efficiency of the algorithm?

2. **What are the implications of PIRO's dependency on a stable policy optimizer in high-dimensional settings?**
   Given that PIRO's overall training stability depends on a stable policy optimizer, especially in high-dimensional settings, how do the authors suggest addressing potential instabilities in policy optimization? Are there specific policy optimization algorithms or techniques that are particularly well-suited for use with PIRO in such settings?

3. **Can the authors provide insights into the generalizability of PIRO to other types of reward functions beyond state-only rewards?**
   The paper demonstrates PIRO's effectiveness in recovering state-only reward functions. How well does PIRO generalize to other types of reward functions, such as those involving state-action pairs or more complex reward structures? Are there any modifications or extensions needed to handle these cases effectively?

**Ethical Concerns:**

["NO or VERY MINOR ethics concerns only"]

**Limitations:**

1. **Dependency on On-Policy Rollouts**: PIRO relies on on-policy rollouts for reward updates, which can significantly reduce sample efficiency in environment interactions. This dependency might limit its scalability to sample-expensive tasks, as generating on-policy trajectories can be time-consuming and computationally expensive.

2. **Overall Training Stability**: While PIRO stabilizes reward learning, overall training stability also depends on a stable policy optimizer, especially in high-dimensional settings. The paper acknowledges that the stability of the entire training process is not solely determined by the reward learning mechanism but also by the policy optimization component. This implies that PIRO's effectiveness might be compromised if the policy optimizer is not robust.

3. **Complexity of Implementation**: The implementation of PIRO involves several components, including reward updates, policy optimization, and the management of on-policy rollouts. This complexity might make the algorithm more challenging to implement and tune compared to simpler IRL methods. Additionally, the need for careful hyperparameter tuning and the dynamic adjustment of coefficients (μ and ν) could be a barrier for practitioners.

**Paper Formatting Concerns:**

No formatting concerns.

**Quality:**

3

**Strengths And Weaknesses:**

Strengths:
1. **Theoretical Guarantees for Stability**: The paper introduces Proximal Inverse Reward Optimization (PIRO), which provides a principled framework for stable reward learning in IRL. This framework ensures monotonic reduction in policy divergence through a Majorization-Minimization (MM) procedure, offering a theoretical guarantee for stable training. This is a significant advancement over existing adversarial IRL methods that often suffer from unstable reward updates.

2. **Empirical Performance**: PIRO demonstrates strong empirical performance across a diverse set of tasks, including classic control, Box2D, Atari, MuJoCo, and real-world animal behavior modeling. It matches or exceeds the performance of state-of-the-art IRL methods while significantly improving training stability. This robustness and effectiveness make PIRO a practical and reliable choice for IRL applications.

3. **Scalability and Sample Efficiency**: PIRO is designed to scale to high-dimensional continuous tasks, making it suitable for complex environments. Additionally, it exhibits strong sample efficiency, requiring fewer expert demonstrations to achieve good performance. This is particularly important in real-world applications where obtaining large amounts of expert data can be challenging or costly.

Weaknesses:
1. **Dependency on On-Policy Rollouts**: PIRO relies on on-policy rollouts for reward updates, which can reduce sample efficiency in environment interactions. This dependency might limit its scalability to sample-expensive tasks, as generating on-policy trajectories can be time-consuming and computationally expensive.

2. **Overall Training Stability**: While PIRO stabilizes reward learning, overall training stability also depends on a stable policy optimizer, especially in high-dimensional settings. The paper acknowledges that the stability of the entire training process is not solely determined by the reward learning mechanism but also by the policy optimization component.

3. **Complexity of Implementation**: The implementation of PIRO involves several components, including reward updates, policy optimization, and the management of on-policy rollouts. This complexity might make the algorithm more challenging to implement and tune compared to simpler IRL methods. Additionally, the need for careful hyperparameter tuning and the dynamic adjustment of coefficients (μ and ν) could be a barrier for practitioners.

---

> ### Author Rebuttal · Authors · 2025-07-31
>
> We thank Reviewer for the thoughtful and constructive feedback. We appreciate the recognition of PIRO's theoretical contributions, strong empirical performance, and practical scalability. Below, we address the reviewer’s concerns regarding on-policy sample efficiency, policy optimizer stability, and generalization to more complex reward structures.
>
> **[Q1,W1,W3]: How does PIRO handle the trade-off between stability and sample efficiency in on-policy rollouts?**
>
> Our framework includes strategies specifically designed to address this trade-off. In the original implementation, we used importance sampling (IS) to reuse previously collected rollouts in reward updates. By logging behavior policy probabilities, we reweighted each trajectory’s contribution to the reward gradient. To avoid high variance, we periodically reset the behavior policy using Monte Carlo sampling, which effectively bounds the IS weights and improves sample efficiency without manual clipping.
>
> Under the assumption of sufficiently informative demonstrations, our framework admits an alternative yet theoretically equivalent approach to reward gradient estimation that circumvents the need for computing advantage values \(A(s, a) = Q(s, a) - V(s)\) through full trajectory rollouts. Rather than estimating advantages via high-variance Monte Carlo returns, we exploit a closed-form identity that connects the log-likelihood of expert actions with the expected reward difference between expert and learner policies:
>
> $$
> \mathbb{E}\_{\rho\^{\pi\_E}} [\log \pi\_{\theta}(a|s)] = \mathbb{E}\_{\rho\^{\pi\_E}} [ Q\_{r\_{\theta}}\^{\pi\_{\theta}}(s, a) - V\_{r\_{\theta}}\^{\pi\_{\theta}}(s) ] = \mathbb{E}\_{\rho\^{\pi\_E}} \left[ r\_{\theta}(s, a) + \gamma \mathbb{E}\_{s'\sim P(\cdot|s,a)} [V\_{r\_{\theta}}\^{\pi\_{\theta}}(s')] - V\_{r\_{\theta}}\^{\pi\_{\theta}}(s) \right]
> $$
> $$
> = \mathbb{E}\_{\rho\^{\pi\_E}} [ r\_{\theta}(s, a)] - \sum\_{t=0}\^\infty \gamma\^t \mathbb{E}\_{s \sim d\^{\pi\_E}\_t, a \sim \pi\_E(\cdot | s)}\left[V\_{r\_{\theta}}\^{\pi\_{\theta}}(s) -\gamma \mathbb{E}\_{s'\sim P(\cdot|s,a)} [V\_{r\_{\theta}}\^{\pi\_{\theta}}(s)] \right]
> $$
> $$
> = \mathbb{E}\_{\rho\^{\pi\_E}} [ r\_{\theta}(s, a)] - \left( \sum\_{t=0}\^\infty \gamma\^t \mathbb{E}\_{s \sim d\^{\pi\_E}\_t}[V\_{r\_{\theta}}\^{\pi\_{\theta}}(s)] - \sum\_{t=0}\^\infty \gamma\^{t+1} \mathbb{E}\_{s \sim d\^{\pi\_E}\_{t+1}}[V\_{r\_{\theta}}\^{\pi\_{\theta}}(s)] \right)
> $$
> $$
> = \mathbb{E}\_{\rho\^{\pi\_E}} [ r\_{\theta}(s, a)] - \mathbb{E}\_{s\sim d\^{\pi\_E}\_0}[V\_{r\_{\theta}}\^{\pi\_{\theta}}(s)] = \mathbb{E}\_{\rho\^{\pi\_E}} [ r\_{\theta}(s, a)] - \mathbb{E}\_{s\sim\eta}[V\_{r\_{\theta}}\^{\pi\_{\theta}}(s)]
> $$
>
> This is a known result in offline IRL and non-adversarial IRL (e.g., ValueDICE, SQIL, IQ-Learn), but relying on rich demonstrations. It is theoretically equivalent to maximizing the expected reward difference $\mathbb{E}\_{\pi_E}[r_\theta] - \mathbb{E}\_{\pi}[r_\theta]$, and may offer improved efficiency by avoiding advantage estimation and long-horizon rollouts.
>
> This sampling strategy serves as a practical implementation choice within our broader framework. Empirically, it improves sample efficiency and reduces per-update training cost, while maintaining stable convergence—addressing both the sampling cost and implementation complexity concerns raised in the review.
>
>
> *Table: Training efficiency (time) normalized by PIRO = 1.00×, across MuJoCo and Gym Robotics tasks.*
>
> |Environment|PIRO|ML‑IRL|GAIL|AIRL|MM|FILTER|IQ‑Learn|HyPE|f‑IRL|
> |---|---|---|---|---|---|---|---|---|---|
> |Ant|1.00×|0.93×|0.10×|0.05×|0.57×|0.20×|0.42×|0.43×|0.19×|
> |Humanoid|1.00×|0.93×|0.05×|0.04×|0.14×|0.10×|0.04×|0.29×|0.19×|
> |Walker|1.00×|0.93×|0.06×|0.05×|0.21×|0.15×|0.12×|0.55×|0.18×|
> |Hopper|1.00×|0.98×|0.06×|0.04×|0.22×|0.16×|0.06×|0.56×|0.15×|
> |Cheetah|1.00×|0.97×|0.07×|0.06×|0.33×|0.34×|0.46×|0.58×|0.25×|
> |Umaze|1.00×|0.97×|0.05×|0.06×|0.40×|0.20×|0.50×|0.27×|0.65×|
> |Medium|1.00×|0.74×|0.03×|0.04×|0.17×|0.17×|0.57×|0.21×|0.46×|
> |Large|1.00×|0.95×|0.03×|0.04×|0.18×|0.18×|0.59×|0.24×|0.59×|
> |Pen|1.00×|0.97×|0.03×|0.04×|0.07×|0.08×|0.06×|0.10×|0.19×|
>
> This demonstrates that PIRO's theoretical framework naturally accommodates efficient sampling strategies, reducing dependence on costly on-policy rollouts without compromising training stability.
>
> Regarding the reviewer’s concern on Dependency on On-Policy Rollouts(W1), we agree that standard PIRO updates require on-policy trajectories, which may reduce sample efficiency in interaction-limited settings. However, our framework is flexible and admits off-policy-compatible variants. In particular, the pairwise classification objective introduced above enables fully off-policy reward updates using single-step transitions from a shared reference distribution. This eliminates the need for fresh rollouts during reward optimization and significantly reduces the cost of environment interaction. Our experiments show that this variant achieves comparable performance while improving per-update efficiency, making PIRO more suitable for sample-expensive or offline imitation scenarios.
>
> >_Table: Mean convergence (**the point where significant increases stop and fluctuations happen afterwards**) steps across five independent runs on MuJoCo tasks._
>
> **Convergence / Rewards**
>
> |Environment|PIRO|AIRL|BC|FILTER|GAIL|HyPE|IQ-Learn|ML-IRL|MM|f-IRL|
> |-----------|----|----|--|------|----|----|--------|------|--|-----|
> |Ant|251/5967.2|160/991.4|60/1631.5|278/-376.3|46/996.9|217/2800.5|32/3589.8|278/5382.5|400/-304.0|29/-304.0|
> |HalfCheetah|205/12587.4|287/2251.4|338/221.2|150/0.3|61/1298.8|154/6473.4|271/3825.5|213/11873.2|136/20.3|365/20.3|
> |Hopper|55/3362.0|32/13.5|38/1034.4|67/37.3|105/3675.9|296/2593.6|408/3424.5|50/3316.4|103/57.8|127/57.8|
> |Humanoid|223/5954.9|50/281.4|29/418.1|25/291.7|74/508.4|54/717.5|59/1847.5|161/5573.4|35/367.2|416/367.2|
> |Walker2d|69/5643.7|51/72.8|29/384.4|81/77.7|40/4158.1|127/1478.7|306/3023.0|83/4794.7|39/70.4|202/70.4|
>
>
> We also take this opportunity to address a related concern from **W3** regarding hyperparameter tuning. Specifically, we tested the impact of reward-related coefficients such as $\mu$ and $\nu$ in Appendix H. The sensitivity plots show that PIRO remains stable across a broad range of values, and performance degrades gracefully outside the recommended intervals. This suggests that careful tuning is helpful but not critical, and that the method is reasonably robust for practitioners.
>
>
> **[Q2] What are the implications of PIRO's dependency on a stable policy optimizer in high-dimensional settings?**
>
> PIRO is designed as a modular reward learning framework that delegates policy optimization to an external RL algorithm. This dependency enables flexibility rather than limitation: PIRO does not require any specific optimizer and can be paired with both on-policy methods like PPO and off-policy methods like SAC or TD3. In high-dimensional environments, where policy optimization becomes more sensitive to exploration and variance issues, this modularity allows PIRO to leverage state-of-the-art policy optimizers to improve scalability and stability.
>
> As our formulation imposes no structural constraint on the optimizer, the reward learning mechanism remains agnostic to the policy update dynamics, enabling seamless integration with more advanced or domain-specific optimizers. This compatibility is particularly beneficial in high-dimensional settings, where the choice of a robust optimizer can significantly impact performance. In this sense, PIRO's reliance on a stable optimizer is not a weakness, but a strength that supports generalization and extensibility across domains.
>
>
>
> ---
>
>
>
> **[Q3]Can the authors provide insights into the generalizability of PIRO to other types of reward functions beyond state-only rewards?**
>
> We discuss two more types of reward functions:
> * Action-only reward $r(a)$, which is common in bandit settings. Our method naturally supports this reward setting by setting an action-only reward model $r_\theta(a)$.
> * Transition-dependent reward $r(s,a,s')$. In IRL, a common technique to cope with this kind of rewards is to assume a potential-based function $h_\phi(s)$ and use a particular form to model $r(s,a,s')$:
> $$r_{\theta,\phi}(s,a,s') = r_\theta(s,a) + \gamma h_\phi(s') - h_\phi(s).$$
>
> Based on this structure, we have
> $$Q_{r_{\theta,\phi}}^{\pi_{\theta,\phi}}(s,a) =  \mathbb{E}[r_\theta(s,a) + \gamma h_\phi(s') - h_\phi(s) + \gamma (r_\theta(s',a')+\gamma h_\phi(s'') - h_\phi(s')) + \cdots] = $$
> $$\mathbb{E}\left[\sum_{t=0}^\infty \gamma^t r_\theta(s,a) | s_0=s, a_0=s\right]$$
>
> Thus, the Q-function gradient under this shaped reward formulation is mathematically equivalent to that used in our original derivation, resulting in complete compatibility with the PIRO update mechanism.

---

### Official Review · Reviewer_oq8i · 2025-06-28

**Clarity:** 3
**Significance:** 3
**Originality:** 2
**Rating:** 4
**Confidence:** 3

**Summary:**

This paper aims to address the problem of reward learning instability that is prevalent in inverse reinforcement learning (IRL), especially in the adversarial optimization framework. The authors put forward a core idea: if each update of the reward function ensures that the divergence between the learned policy and the expert policy decreases monotonically, then the stability of learning can be guaranteed in principle. Based on this, the paper first proposes a theoretical framework called "Proximal Reward Update" (PRU), which theoretically guarantees the monotonic decrease of the policy KL divergence through a majorization-minimization procedure. Subsequently, the authors instantiate this theoretical framework into a practical algorithm called "Proximal Inverse Reward Optimization" (PIRO). PIRO is an IRL algorithm that non-adversarially and explicitly learns reward functions, which relies on on-policy trajectory sampling for reward updates. Experiments on multiple simulated control tasks and a real-world animal behavior modeling task show that PIRO has significantly improved training stability while maintaining or exceeding the performance of existing advanced methods.

**Questions:**

On the gap between theory and practice: The PIRO algorithm replaces the constant C in the theoretical derivation with an adaptive coefficient $\mu$. To what extent does this heuristic design affect the stability of the algorithm? Does it break the theoretical monotonic convergence guarantee in practice? If so, can you quantify how often or how much the KL divergence actually "rises" in iterations in your experiments?

On the trade-off in sample efficiency: The paper correctly points out the trade-off between "stability" and "sample efficiency of online interactions". Can you further elaborate on in which types of tasks this trade-off is acceptable (e.g., having a cheap and efficient simulator) and in which tasks it may be a bottleneck for its application (e.g., real-world physical interactions)? In addition, is there potential to partially replace online trajectory sampling with more advanced off-policy correction techniques (such as importance sampling with truncation) to alleviate this problem?

Comparison with non-adversarial value matching methods: The paper classifies PIRO as a "policy alignment" method. "Value matching" methods such as IQ-Learn also achieve non-adversarial learning. Although IQ-Learn has been used as a baseline, can you provide deeper insights into the fundamental pros and cons of these two non-adversarial paths: i.e., what are the core advantages and limitations of ensuring “policy divergence decreases monotonically” (PIRO’s path) versus retaining “value matching goal but avoiding adversarial training” (IQ-Learn’s path)?

**Ethical Concerns:**

["NO or VERY MINOR ethics concerns only"]

**Final Justification:**

Some questions have been solved, and I think the experimental part can still be improved.

**Limitations:**

Yes. The authors clearly discuss the two main limitations of PIRO in the conclusion: 1. The overall stability still depends on a stable policy optimizer; 2. The reliance on online policy sampling may reduce the sample efficiency of environment interaction and limit its scalability. This part of the discussion is sufficient and relevant.

**Quality:**

2

**Strengths And Weaknesses:**

Strengths:

The paper proposes a novel theoretical perspective to solve the stability problem of IRL, which shifts the core of the problem from the traditional "reward matching" or "adversarial game" to ensuring "monotonic convergence of policy divergence". This is an insightful and principled contribution.

The paper contains solid theoretical work, and provides detailed derivations and proofs for key theorems in the appendix. The experimental evaluation is extensive, covering a variety of tasks from classical control to high-dimensional MuJoCo and even real-world datasets, and is comprehensively compared with 8 baselines including several state-of-the-art methods, which strongly supports its effectiveness.

Weaknesses:

There is a gap from theory to practice. The practical PIRO algorithm makes several heuristic approximations to the theoretical framework PRU, for example, replacing the theoretically derived (usually difficult to calculate or too large) penalty term coefficient C with adjustable coefficients $\mu and \nu$, and replacing the maximum norm with the L2 norm. These approximations make it impossible to strictly guarantee the theoretical monotonicity in practice.

The idea of ​​“policy alignment” for non-adversarial IRL is not new, and other works have adopted similar concepts. The originality of PIRO lies in how to “guarantee” the monotonicity of the alignment process through the MM framework.

A core trade-off of the method is its reliance on on-policy trajectory sampling. As the authors acknowledge in the paper, this reduces its sample efficiency in environmental interaction, which may make it difficult to apply to tasks with high cost of interacting with the real world, such as robotics.

The overall process of the algorithm is relatively complex, and online sampling is required to estimate the gradient of the advantage function with respect to the reward parameter, which may become an obstacle to its generalization and application.

---

> ### Author Rebuttal · Authors · 2025-07-31
>
> We sincerely thank the reviewers for their constructive and thoughtful feedback. We are encouraged that the reviewers found our theoretical contribution principled and insightful, appreciated the rigorous derivations, and recognized the breadth of our empirical evaluations across both simulated and real-world domains. Below, we address reviewer’s comments in turn.
>
> ---
>
> **[W1,W2,Q1]: In practice, does using heuristic approximations (e.g., adaptive coefficient $C$) break the theoretical monotonicity guarantees of PIRO? Can you quantify how often KL divergence increases during training?**
>
> Indeed, using a relaxed adaptive coefficient $C$ cannot rigorously guarantee the monotonicity in theory. Theoretically, with an adaptive coefficient and assuming we have a space of reward parameter that makes the KL divergence Lipschitz continuous, we can then quantify that KL divergence increase is upper bounded by $\mathcal{O}(\frac{C\epsilon}{1-\gamma})$. This is because $Q$ is $\frac{1}{1-\gamma}$-Lipschitz in $r$ and the energy-based policy $\pi$ is $1$-Lipschitz in $Q$ (assuming the temperature = 1). Empirically, we found that the rate of KL divergence increases is lower than baselines in general. On the other hand, approximating the theory for better feasibility and efficiency is common in reinforcement learning approaches. Closely related to our method is TRPO, which also employs a heuristic KL constraint to replace the theory-informed one that guarantees the monotonic policy improvement. At its reputation-gain successor PPO, the KL constraint is further simplified as a penalty term with an adaptive coefficient (or clipping). This simplification makes the great success of PPO, which is now the gold-standard of RL algorithms. In light of this, we believe our methods -- inspired by theory but not rigorously following the theory -- are in line with the mainstream design and practice of RL algorithms. It is also worth noting that although some prior methods also adopt the concept of policy alignment, no one, at least in theory, analyzes how to achieve monotonic alignment, but we did in this work. This sets our work apart from the literature.
>
> ---
>
>
>
> **[Q2,W3,W4]: Can you elaborate on when the trade-off between stability and sample efficiency is acceptable, and whether off-policy correction techniques (e.g., truncated IS) could be used to alleviate this?**
>
> This trade-off is acceptable in settings where interaction is cheap (e.g., simulated control), or where expert trajectories are fixed and reward learning is purely offline (e.g., behavioral modeling). For instance, in MuJoCo, stable reward updates are more beneficial than marginal gains from reuse. In real-world applications like sim2real robotics or clinical modeling from logs, safety and interpretability matter more than sample count—making stability a higher priority than efficiency.
>
> By contrast, in real-time robotics or interactive recommendation systems—where every environment step is expensive or delayed—this trade-off becomes limiting, and improving sample efficiency becomes essential.
>
> As for off-policy correction, we already employ importance sampling in our reward updates, using behavior-policy log-probabilities stored in the trajectory buffer. Instead of explicit truncation, we control variance via periodic Monte Carlo resampling, which resets the behavior policy and keeps IS weights bounded. This approach achieves similar stability benefits without requiring manual clipping thresholds.
>
> In addressing this question, we also aim to clarify two concerns raised in the weaknesses: **the reliance on on-policy sampling (on-policy cost)** and **the complexity of gradient estimation** .
>
> Within our framework, when demonstrations are rich, we can adopt an equivalent but potentially more efficient formulation of reward gradient estimation that eliminates the need for estimating $A(s) = Q(s,a) - V(s)$ on trajectory rollouts. Instead of relying on advantage estimates derived from sampled returns—which often involve high-variance Monte Carlo evaluations—we directly optimize the expected reward difference between expert and policy transitions according to the following equivalence relationship:
>
> $$
> \mathbb{E}\_{\rho^{\pi\_E}} [\log \pi\_{\theta}(a|s)] =
> \mathbb{E}\_{\rho^{\pi\_E}} [ Q\_{r\_{\theta}}^{\pi\_{\theta}}(s, a) - V\_{r\{\theta}}^{\pi\_{\theta}}(s) ] =
> \mathbb{E}\_{\rho^{\pi\_E}} [ r\_{\theta}(s, a) + \gamma \mathbb{E}\_{s\'\sim P(\cdot|s,a)} [V\_{r\_{\theta}}^{\pi\_{\theta}}(s\')] - V\_{r\_{\theta}}^{\pi\_{\theta}}(s) ]
> $$
>
> $$
> = \mathbb{E}\_{\rho^{\pi\_E}} [ r\_{\theta}(s, a)] -
> \sum\_ {t=0}^\infty \gamma^t \mathbb{E}\_{s \sim d\_t^{\pi\_E}, a \sim \pi\_E(\cdot | s)}
> \big[V\_{r\_{\theta}}^{\pi\_{\theta}}(s) - \gamma \mathbb{E}\_{s\'\sim P(\cdot|s,a)} [V\_{r\_{\theta}}^{\pi\_{\theta}}(s\')] \big]
> $$
>
> $$
> = \mathbb{E}\_{\rho\^{\pi\_E}} [ r\_{\theta}(s, a)] - \left( \sum\_{t=0}\^\infty \gamma\^t \mathbb{E}\_{s \sim d\^{\pi\_E}\_t}[V\_{r\_{\theta}}\^{\pi\_{\theta}}(s)] - \sum\_{t=0}\^\infty \gamma\^{t+1} \mathbb{E}\_{s \sim d\^{\pi\_E}\_{t+1}}[V\_{r\_{\theta}}\^{\pi\_{\theta}}(s)] \right)
> $$
>
>
> $$
> = \mathbb{E}\_{\rho^{\pi\_E}} [ r\_{\theta}(s, a)] -
> \mathbb{E}\_{s \sim d^{\pi\_E}\_0}[V\_{r\_{\theta}}^{\pi\_{\theta}}(s)]
> = \mathbb{E}\_{\rho^{\pi\_E}} [ r\_{\theta}(s, a)] -
> \mathbb{E}\_{s \sim \eta}[V\_{r\_{\theta}}^{\pi\_{\theta}}(s)]
> $$
>
> Here, $d^\pi_t(s)$ denotes the state distribution induced by $\pi$ at $t$ and $\eta$ is the (known) initial state distribution. This is a known result in offline IRL and non-adversarial IRL (e.g., ValueDICE, SQIL, IQ-Learn), but relying on rich demonstrations. It is theoretically equivalent to maximizing the expected reward difference $\mathbb{E}\_{\pi_E}[r_\theta] - \mathbb{E}\_{\pi}[r_\theta]$, and may offer improved efficiency by avoiding advantage estimation and long-horizon rollouts.
>
> This sampling strategy serves as a practical implementation choice within our broader framework. Empirically, it improves sample efficiency and reduces per-update training cost, while maintaining stable convergence (**see Table below**). This addresses both the sampling cost and implementation complexity concerns raised in the review.
>
> To further demonstrate PIRO’s sample efficiency, we report the number of iterations required for each algorithm to reach convergence across a range of MuJoCo tasks. The table below summarizes the average convergence steps of five independent runs, providing a direct comparison of training speed under consistent evaluation protocols.
>
> >_Table: Mean convergence (**the point where significant increases stop and (small or big) fluctuations happen afterwards**) steps across five independent runs on MuJoCo tasks._
> >_Format: Convergence / Rewards_
>
> |Environment|PIRO|AIRL|BC|FILTER|GAIL|HyPE|IQ-Learn|ML-IRL|MM|f-IRL|
> |-----------|----|----|--|------|----|----|--------|------|--|-----|
> |Ant|251/5967.2|160/991.4|60/1631.5|278/-376.3|46/996.9|217/2800.5|32/3589.8|278/5382.5|400/-304.0|29/-304.0|
> |HalfCheetah|205/12587.4|287/2251.4|338/221.2|150/0.3|61/1298.8|154/6473.4|271/3825.5|213/11873.2|136/20.3|365/20.3|
> |Hopper|55/3362.0|32/13.5|38/1034.4|67/37.3|105/3675.9|296/2593.6|408/3424.5|50/3316.4|103/57.8|127/57.8|
> |Humanoid|223/5954.9|50/281.4|29/418.1|25/291.7|74/508.4|54/717.5|59/1847.5|161/5573.4|35/367.2|416/367.2|
> |Walker2d|69/5643.7|51/72.8|29/384.4|81/77.7|40/4158.1|127/1478.7|306/3023.0|83/4794.7|39/70.4|202/70.4|
>
> These results demonstrate that although PIRO may not exhibit the fastest raw convergence in terms of iteration count, it consistently achieves stable and reliable progress across tasks. Unlike methods that converge quickly but suffer from regressions or collapse, PIRO guarantees monotonic policy improvement (in theory)—ensuring that each update brings the learner closer to the expert. As a result, PIRO avoids unnecessary detours and converges toward expert behavior in a more direct and (empirically) sample-efficient manner. This highlights the practical benefit of our framework: trading off aggressive updates for principled, steady advancement leads to faster and more stable convergence overall.
>
>
>
> **[Q3]: Can you compare the core strengths and limitations of PIRO’s monotonic policy alignment versus value-matching non-adversarial methods like IQ-Learn?**
>
> Both PIRO and IQ-Learn avoid adversarial training by replacing discriminator-style reward objectives with more stable surrogate losses. The key difference lies in what is aligned with the expert: IQ-Learn focuses on value matching—training a value function such that its gradient implicitly guides policy learning—whereas PIRO explicitly ensures monotonic reduction in policy divergence from the expert via a proximal reward update mechanism.
>
> The main advantage of PIRO is that it directly optimizes for behavioral alignment between the learned and expert policies. This leads to predictable, stable policy improvement steps with a clear theoretical guarantee (via majorization-minimization). In contrast, IQ-Learn's value-matching objective does not guarantee that policy divergence decreases, and its performance can depend heavily on the quality of value estimates, which are sensitive to bootstrapping errors.
>
> That said, IQ-Learn can be more sample-efficient in certain settings due to its compatibility with off-policy value-based learning, while PIRO is more conservative by design. Our approach prioritizes stability and monotonic policy improvement, which is particularly beneficial in settings where safe or interpretable reward learning is critical.

---

> > ### Comment · Reviewer_oq8i · 2025-08-07
> >
> > Thanks for the clarification, which addresses some of my concerns. I will keep my scores.

---

> > > ### Comment · Reviewer_oq8i · 2025-08-07
> > >
> > > Thank you for your detailed response. You've addressed most of my concerns and it's excellent work. The only point I might not fully address is that you claim the growth rate of KL divergence is "slower than the baseline," but you haven't provided empirical evidence for this specific claim. While the policy divergence curves in the appendix show that PIRO is generally very stable, they don't provide the requested quantitative analysis. My question at the time was, "Can you quantify how often or how much KL divergence actually 'increases' across iterations in your experiments?" Therefore, I've maintained my original score.

---

> > > > ### Author Response · Authors · 2025-08-07
> > > >
> > > > Thansk for your further explanations.
> > > >
> > > > We have already provided empirical evidence for this specific claim: *"the growth rate of KL divergence of PIRO is slower than the baseline in general".* See **Appendix D.2: Policy Divergence Curves**, where you can find that the KL divergence of baselines often “jump up and down”, while the KL divergence of our PIRO consistently decreases or fast converges to a very low level.
> > > >
> > > > We hope this addresses your concerns.

---

> > > > > ### Comment · Reviewer_oq8i · 2025-08-07
> > > > >
> > > > > Thanks for your response. I will update my rating soon.

---

> > > > > > ### Author Response · Authors · 2025-08-07
> > > > > >
> > > > > > Thanks for your time and really constructive & valuable comments along this.

---

> ### Comment · Area_Chair_m8Zd · 2025-08-01
> **Opinion after rebuttal**
>
> Dear reviewer,
>
> The authors have posted a detailed response. Does it assuage your concerns? I struggle to find a concrete reason for rejection from your review. I am a bit at a loss when you say that on one hand, it's insightful theory, and on the other that some things are not theoretically justified.

---

> ### Author Response · Authors · 2025-08-07
>
> Thanks for your engagement and comments. Could you please specify which parts of your concerns have been addressed and which part not? Also, could you please take a look at and reply to AC’s questions above (ignore this if you have already replied to it and keep it confidential for the authors)?  Your detailed information will be helpful to us.
> Thanks in advance.

---

> > ### Comment · Area_Chair_m8Zd · 2025-08-07
> > **Theory Vs Practice**
> >
> > I guess the paper could be strengthened slightly by adding experiments with the algorithm given in the theory

---

> > > ### Author Response · Authors · 2025-08-08
> > > **Replying to AC:  Performance Comparison Between Theoretical and Adaptive C**
> > >
> > > Thanks for the suggestion.
> > >
> > > As you suggested, we further conduct an experiment on $\mathtt{Cartpole}$ with the **exact theoretical value of step size $C$**. The closed-form expression of $C$ from our theoretical analysis is:
> > >
> > > $$
> > > C =  \frac{2| \mathcal{A}|}{(1-\gamma)^2}  + \frac{(5-\gamma)| \mathcal{A}| R_{\mathrm{max}} + (\gamma - \gamma^2+2)| \mathcal{A}|\log| \mathcal{A}|}{(1-\gamma)^4}.
> > > $$
> > >
> > > In $\mathtt{Cartpole}$, the size of action space $|\mathcal{A}| = 2$ and $R_{\mathrm{max}}=1$. We take the discount factor $\gamma=0.9$. So we have that the exact theoretical value $C \approx$ **111,373.55**.
> > >
> > > The results for theoretical $C$ vs. adaptive $C$ (bounded by [0.001, 10]) are reported below. The adaptive method significantly reduces KL divergence across training (mean  101.30 vs. 539.49), while also achieving substantially higher final rewards, both undiscounted (469.9 vs. 12.7) and discounted (10.00 vs. 4.82). For using theoretical $C$, within the acceptable training time range, we could observe improvement in the reward performance, but not as significant and fast as using the adaptive $C$. But note that, in theory, given a long enough train time window, we could observe the superior performance of using the theoretical $C$.
> > >
> > > This supports the motivation for adaptively adjusting $C$ in practice.
> > >
> > >
> > > > **Table.** **PERFORMANCE COMPARISON BETWEEN THEORETICAL and ADAPTIVE** $C$ **on** $\mathtt{Cartpole}$
> > >
> > > === Expert Policy Evaluation ===
> > >
> > > Demo. Trajectory Length: 512
> > >
> > > Undiscounted Return: 500.0
> > >
> > > Discounted Return ($\gamma$=0.9): 10.00
> > >
> > > ===========================
> > >
> > > **Reward format: Undiscounted / Discounted**
> > > |Env steps|Reward (theoretical $C$) ($\uparrow$)|Reward (Adaptive $C$) ($\uparrow$)|KL Div. (theoretical $C$) ($\downarrow$)|KL Div. (Adaptive $C$) ($\downarrow$)|
> > > |--------|-------------------------------------|----------------------------------|----------------------------------------|--------------------------------------|
> > > |0|9.1/3.77|9.5/3.50|928.13|878.65|
> > > |10K|9.5/3.70|9.2/3.70|919.79|938.95|
> > > |20K|9.0/3.77|9.7/3.83|919.34|951.95|
> > > |30K|9.5/3.57|8.7/3.83|935.20|981.90|
> > > |40K|9.4/3.83|9.3/3.90|922.54|954.64|
> > > |50K|9.2/3.83|10.6/4.32|965.18|228.31|
> > > |60K|9.6/3.83|71.7/9.38|912.25|238.54|
> > > |70K|9.5/3.83|185.1/**9.99**|965.55|171.34|
> > > |80K|9.3/3.70|297.8/**10.00**|891.60|149.84|
> > > |90K|9.7/3.83|333.3/**9.99**|955.32|140.75|
> > > |100K|9.5/3.77|309.1/**10.00**|828.67|150.65|
> > > |200K|12.7/4.82|**469.9**/**10.00**|539.49|101.30|

---

### Official Review · Reviewer_55QK · 2025-06-30

**Clarity:** 4
**Significance:** 2
**Originality:** 3
**Rating:** 5
**Confidence:** 4

**Summary:**

The paper introduces a new non-adversarial inverse RL method based on updating the reward to decrease the KL divergence between the expert and learner policies. The authors prove that the idealized version of this algorithm guarantees policy improvement and implement an approximation based on the idea of surrogate objectives. The experiments show that the method has better stability, sample efficiency, and better performance than baselines. The method is evaluated on diverse tasks include continuous control and animal behavior modeling. The paper also shows that the learned reward function accurately models the ground truth reward, improving interpretability.

**Questions:**

1. Did you ever try the naive gradient-based approaches to optimize the KL against the expert? If so, is it unstable in practice?
2. How sensitive is PIRO to the inner policy optimizer? Did you try other methods beside PPO? Is it correct that any method which learns the EBM (and can sample from it) would work?
3. Is there any kind of clipping used to control the importance ratios in the gradient updates? Empirically, do these not blow up?
4. For Q and V gradient updates, could you maintain a buffer of all off policy rollouts and probabilities seen up to that point? Do you have thoughts on whether PIRO would stably support this kind of change to improve sample efficiency?
5. Is there an explanation why on-policy robustness of PIRO seems to differ per task? Hopper performance seems unaffected despite it having complex dynamics.
6. For reward recovery in appendix E, does the PIRO-learned policy match the expert performance?
7. Why is it reasonable to assume there should be a shared policy across meerkat individuals?

**Ethical Concerns:**

["NO or VERY MINOR ethics concerns only"]

**Final Justification:**

I am maintaining my score of 5 after rebuttal. I had some minor questions about the empirical results which the authors resolved during rebuttal: sensitivity to PPO vs other optimizers, stability of importance sampling weights, and other baselines like direct KL minimization. There was an extended discussion by Reviewer zT3i on the online sample efficiency of the baselines the authors use. Although the requested baseline demonstrates strong sample efficiency, I believe other contributions of the proposed method (stability, sample efficiency re. expert demonstrations) are still valuable, and the authors did a faithful job of reproducing baselines.

**Limitations:**

Yes

**Quality:**

4

**Strengths And Weaknesses:**

The presentation of the work is extremely clear. The clarity of explanation, grounding of related work, motivation for surrogates / connection to CPI / TRPO, and presentation of diagrams and plots are all very good. I really enjoyed reading the paper because of the fantastic diagrams. The core part of the method is pretty intuitive. The idea of surrogates is pretty classic in RL but the authors apply it in a new way for inverse RL (which needs stable, non-adversarial methods). I think the main algorithmic idea will be of broader interest to the community.

The main set of results are pretty convincing and cover a wide variety of settings and eval measurements: the authors measure rewards, stability, and sample efficiency for  various control, Atari, and real-world tasks. These are nicely supplemented with very thorough ablations. The experimental results positively answer all of the questions introduced in Section 6. Although the final algorithm is complex, the empirical changes are well-motivated and the results are positive. I have various questions about some details of the empirical results.

---

> ### Author Rebuttal · Authors · 2025-07-31
>
> We thank the reviewer for the thoughtful feedback. We're glad you found the **"presentation extremely clear"**, the diagrams **"fantastic"**, and the method **"intuitive"**. We're especially encouraged by your belief that the **"main algorithmic idea will be of broader interest to the community"**. We address your questions below.
>
> **[Q1]: Did you try directly minimizing the KL to the expert, and was it unstable?**
>
> Yes. We experimented with directly minimizing the policy KL divergence $\mathrm{KL}(\pi_E | \pi_\theta)$ via gradient descent. However, as shown in the table below, this naive approach exhibits significant instability across tasks. In environments like CartPole and Q*bert, performance dropped sharply and fluctuated erratically. Even in more stable tasks like Humanoid, it failed to consistently improve the policy.
>
> These results reflect common issues with directly optimizing KL: the gradients can be high-variance and poorly aligned with actual policy improvement, especially when $\pi_\theta$ is far from $\pi_E$. This was a key motivation behind our design of PIRO. By using a majorization-minimization (MM) framework to upper-bound the KL surrogate at each iteration, PIRO ensures stable updates with guaranteed monotonic improvement—effectively avoiding the oscillations and regressions observed with naive KL minimization.
>
> To illustrate this behavior concisely, we select one representative environment from each category (Gym, MuJoCo, Atari, Box2D), due to words constraints.
>
> > _Table: Return comparison between PIRO and naive KL minimization over iterations. Directly optimizing KL divergence results in instability (high fluctuation)._
>
> |Env|Iter|Original|Naive KL|
> |----|----|--------|---------|
> |CartPole|0|281.47|46.37|
> | |25|384.18(+36%)|9.41(-80%)|
> | |50|351.37(-9%)|9.48(+0.8%)|
> | |75|363.39(+3%)|9.40(-0.8%)|
> | |100|380.89(+5%)|9.36(-0.5%)|
> |BipedalWalker|0|-88.57|-92.49|
> | |25|-78.96(+11%)|-103.25(-12%)|
> | |50|-65.85(+17%)|-100.68(+2%)|
> | |75|-62.78(+5%)|-99.89(+0.8%)|
> | |100|-64.58(-3%)|-91.41(+8%)|
> |Humanoid|0|278.93|296.61|
> | |10|256.37(-8%)|216.17(-27%)|
> | |20|275.36(+7%)|207.89(-4%)|
> | |30|279.59(+2%)|302.82(+46%)|
> | |40|276.98(-0.9%)|307.65(+2%)|
> | |50|316.56(+14%)|246.95(-20%)|
> |Qbert|0|138.53|69.63|
> | |25|149.92(+8%)|42.42(-39%)|
> | |50|166.87(+11%)|39.24(-8%)|
> | |75|144.21(-14%)|10.95(-72%)|
> | |100|165.76(+15%)|46.79(+327%)|
>
>
> **[Q2-1]: How sensitive is PIRO to the inner policy optimizer? Did you try other methods beside PPO?**
>
> PIRO’s core theoretical framework does not rely on a specific policy optimizer. It only assumes access to a differentiable policy that supports sampling and advantage estimation. Therefore, in principle, any optimizer—on-policy or off-policy—can be used, as long as it enables stable policy updates and maintains sufficient overlap with expert trajectories.
>
> In response to this question, we conducted additional experiments using **Soft Actor-Critic (SAC)** in place of PPO for policy optimization.
>
> > _Table: Performance across 5 independent runs on MuJoCo control tasks using high-quality expert demonstrations trained with SAC._
>
> | Task            | Expert   | **PIRO** | BC     | GAIL   | AIRL   | IQ     | HyPER  | FILTER | MM     | ML-IRL | DAC     | P2IL    | CSIL   |
> |-----------------|----------|----------|--------|--------|--------|--------|--------|--------|--------|--------|---------|---------|--------|
> | Ant-v2          | 5572.2   | **5967.2** | 1631.5 | 996.9  | 991.4  | 3589.8 | 2800.5 | -376.3 | -304.0 | 5382.5 | 923.8   | 976.6   | 420.7  |
> | Humanoid-v3     | 5468.8   | **5954.9** | 418.1  | 508.4  | 281.4  | 1847.5 | 717.5  | 291.7  | 367.2  | 5573.4 | 76.3    | —       | —      |
> | Walker2d-v3     | 5453.4   | **5643.7** | 384.4  | 4158.1 | 72.8   | 3023.0 | 1478.7 | 77.7   | 70.4   | 4794.7 | -3.0    | 1054.0  | 686.1  |
> | Hopper-v3       | 3559.6   | 3362.0 | 1034.4 | **3675.9** | 13.5   | 3424.5 | 2593.6 | 37.3   | 57.8   | 3316.4 | 3321.6  | 25.8    | 6.7    |
> | HalfCheetah-v2  | 12174.6  | **12587.4** | 221.2  | 1298.8 | 2251.4 | 3825.5 | 6473.4 | 0.3    | 20.3   | 11873.2| 9645.0  | -0.1    | -107.2 |
>
> > _Table : Performance across 5 independent runs on Gym Robotics tasks._
>
> |Task|Expert|**PIRO**|BC|GAIL|AIRL|MM|FILTER|HyPER|IQ|ML-IRL|P2IL|
> |---|---|---|---|---|---|---|---|---|---|---|---|
> |AntMaze-UMazeDense-v4|35.6|**25.7**|8.8|5.2|4.5|5.1|6.1|11.9|3.9|4.2|3.4|
> |AntMaze-MediumDense-v4|26.9|**9.4**|1.1|1.3|2.6|3.4|1.9|3.0|3.4|0.9|2.9|
> |AntMaze-LargeDense-v4|11.5|**8.8**|1.1|0.9|3.4|1.7|0.6|1.5|0.8|0.3|0.2|
> |AdroitHandPen-Human-v1|1062.5|**254.0**|44.1|-8.7|-593.9|-344.3|-685.4|-866.7|-751.9|-251.2|-61.2|
>
> These results demonstrate that PIRO is not sensitive to the choice of policy optimizer, and performs robustly under both PPO and SAC when compatible conditions are met.
>
>
> **[Q2-2] Is it correct that any method which learns the EBM (and can sample from it) would work**
>
> In theory, any method which learns the EBM (and can sample from it) is compatible with our framework. But this is not to say that they will work well because the sampling mechanism should be well designed to make it able to cope with high-dimensional and continuous states or actions. Given that, SAC would be a suitable alternative policy optimizer and its peformacme can be found above.
>
> **[Q3]: Is there any kind of clipping used to control the importance ratios in the gradient updates? Empirically, do these not blow up?**
>
> We do not apply explicit clipping to the importance ratios in PIRO. Instead, we reset the behavior policy used for advantage computation to the current policy after a **small** fixed interval. This guarantees the current policy will not deviate from the old policy too much, ensuring that importance weights remain well-behaved.
>
> Furthermore, our implementation buffers and reuses trajectories based on specific (state, action) pairs. This design amortizes advantage estimation across updates and avoids repeatedly sampling from stale or highly divergent policies. Empirically, this mechanism helps contain the variance of importance weights. Across all tasks, we observe no gradient explosions related to importance sampling—learning.
>
> **[Q4]: For Q and V gradient updates, could you maintain a buffer of all off-policy rollouts and probabilities seen up to that point? Do you have thoughts on whether PIRO would stably support this kind of change to improve sample efficiency?**
>
> While PIRO is primarily on-policy in its reward update logic, our implementation does maintain trajectory buffers for off-policy reuse—specifically for estimating Q and V values under the old policy and current reward. These buffers are keyed by individual $(s, a)$ or $s$ pairs and store full rollouts sampled from the behavior policy. The associated log-probabilities from the behavior policy are retained to enable importance sampling (IS) during advantage estimation.
>
> This design is crucial for our IS-based reward updates, as reusing cached rollouts allows us to amortize gradient estimates without repeatedly querying the environment. To avoid accumulating divergence, we do not maintain an ever-growing replay buffer. Instead, we periodically clear and refresh the buffers via Monte Carlo re-sampling (see Table 2), which resets the behavior policy to the current policy snapshot. This bounds policy shift and ensures that the IS weights remain well-behaved, as discussed in **[Q3]**.
>
> **[Q5]: why on-policy robustness of PIRO seems to differ per task?**
>
> While PIRO theoretically ensures monotonic alignment at the reward level, its practical robustness depends on how reliably policy updates respond to reward signals. Hopper, despite having nonlinear dynamics, tends to exhibit smooth and dense reward landscapes and stable PPO training behavior. This makes it less sensitive to minor variations in reward updates, hence showing strong performance and stability under PIRO.
>
> In contrast, tasks like Ant or HalfCheetah often involve higher-dimensional control and more complex reward trade-offs (e.g., speed vs stability), where small reward shifts can induce divergent behaviors. Since PIRO does not directly constrain policy updates but rather modulates them via learned rewards, the resulting robustness can vary across environments depending on policy sensitivity to the shaped rewards.
>
> **[Q6]: For reward recovery in Appendix E, does the PIRO-learned policy match the expert performance?**
>
> Yes. The experiment in Appendix E is conducted in a 6×6 GridWorld environment with a manually designed ground-truth reward that depends only on the agent’s position. The PIRO agent is trained without access to the true reward and learns solely from expert demonstrations. The reward heatmaps confirm that the recovered reward closely matches the ground-truth.
>
> To verify behavioral alignment, we measured the average return of the learned policy over the last 5 evaluation rounds and compared it to the expert return:
>
> |Policy|AvgReturn|
> |------|---------|
> |Expert|-9.24|
> |PIRO|-8.48|
>
> This indicates that the PIRO-learned policy not only matches but slightly exceeds the expert's performance in terms of accumulated reward.
>
>
> **[Q7]: Why is it reasonable to assume a shared policy across individual meerkats in the dataset?**
> The meerkat behavivor dataset only records meerkats' physical behavior, like transiting among locations in their habitat, foraging and play-fight. These physical behaviors do not include social, sychological or ethical behavior, and thus they are largely same among adult meerkats. This assertion is supported by the records in the dataset, where each individual meerkat has a unique track number in a video clip, one can find no outstanding differnece over behavior of different individual meerkats. For example, raising guard would be a supporting evidence, as different adults may act as guards in turn.

---

> > ### Comment · Reviewer_55QK · 2025-08-05
> >
> > Thank you for the detailed response and additional experiments on KL and off-policy optimizers! A few more comments:
> > 1. Re. Q3, how is the reset frequency determined?
> > 2. Also re. Q3, could you elaborate a bit more on why the importance sampling weights are stable? I'm looking at line 12 in Appendix B: I'd expect the product to easily blow up or shrink for longer-horizon tasks. I was generally under the impression that explicit methods for variance control were typically needed for stability (like clipping or biased estimates of the full trajectory ratio). Do you have a sense of the scale of this importance sampling product and how well-behaved it is? Is it just that k (reset freq) is set small enough where the policies don't deviate much?
> > 3. Also re. the ongoing discussion with Reviewer zT3i, it seems that the main disagreement is on the sample efficiency of PIRO vs. other baselines. One suggestion is to focus the discussion in the paper and empirical contribution of PIRO as an IRL method with guaranteed and improved stability (and remove any claims for sample efficiency).

---

> ### Author Response · Authors · 2025-08-06
> **Importance Sampling Weights and Sample efficiency**
>
> ## 1. How is the reset frequency determined
>
> We tested the policy reset frequency within the range [5, 10, 20, 50, 100]. A large frequency reduces the computational burden but increases the variance, while a too too samll frequency improves accuracy but raises the computational load. We thus choose **10** which demonstrates a tradeoff betweem accuracy and computational efficiency.
>
> ---
>
>
> ## 2. Stability of Importance Sampling Weights
>
> We didn't apply clipping to the importance ratios in our experiments. However, we find that the product of importance ratios do not blow up or shrink in general; see the Table below for our reported results of importance ratios in Cartpole-v1. We owe this stability to the proximal reward update, which leads to samll policy changes and thus stable importance ratios. As you suggested, we also add the clipping operation to the importance sampling weights. The results indicate that the clipping could reduce variances and improve reward performance, though not that significantly (see Table below). This suggests us to incorporate clipping into our final released code for public use.
>
>
> >_**Table: Important sampling weight comparison in Cartpole-v1.**_
>
> **Note 1:** *In PIRO, each global roound process several batches, each with an importance sampling weight.
> We report the maximum, mean, and standard deviation of the importance sampling weights among these batches.*
>
> * *Reset Frequency = 10.*
>
> | |**With Clipping ([0.1,10])**|||**Without Clipping**|||
> |-------|---------|---|----------|--------|-----------|--------|
> |global Round|MaxWeight|Mean±Std|reward|MaxWeight|Mean±Std|reward|
> |0|-|-|9.3|-|-|8.8|
> |2|10.00|0.95±2.42|179.7|35.59|1.74±5.82|221.5|
> |4|10.00|1.01±2.28|304.2|33.85|1.71±5.56| 272.2 |
> |6|10.00|1.09±2.10|382.6|29.97|1.62±4.69| 307.8 |
> |8|10.00|1.19±1.57|470.7|17.49|1.29±2.15| 462.0 |
> |10 (reset)|-|-|446.1|-|-|365.5 |
> |12|1.73|1.00±0.13|500.0|1.45|1.00±0.16|466.4 |
> |14|1.59|1.00±0.17|432.0|1.51|1.00±0.19|498.5 |
> |16|1.39|1.00±0.12|401.9|1.62|1.00±0.18| 417.6 |
> |18|1.61|1.00±0.13|461.8|1.27|1.00±0.18|456.5  |
> |20 (reset)|-|-|420.3|-|-|446.1|
>
> **Note 2:** *As shown, clipping helps stabilize the mportance sampling weights early in training (Rounds 2–8), while maintaining slightly higer or comparable rewards to the unclipped case. After the first reset (Round 10), both configurations converge to similar mportance sampling weight (around 1) with low variance and near-unity mean weights, demonstrating the effectiveness of proximal reward update that leads to small and stable policy change.*
>
> ---
>
> ## 3. Re: Sample efficiency
>
> Thanks for pointing our this. We have made a clarification as follows. The sample efficiency that Reviwer zT3i mentioned is wrt **online sample efficiency**, while we cliamed "sample efficient" is wrt "expert demonstration samples", i.e., PIRO could demonstrate good performance with small amount (one trajectory) of expert samples, as verified in our previous responses. I deed, as you pointed out, our mian contribution is on the stability while not sacrificing the final performance of reward recovery and policy imitation. We have acknowledged the limitation in the experiment and conclusion section that: in general, the stability comes at the cost of more frequent online sampling . We will continue to explore **fully offline IRL with stability guarantees**, which we believe will further contribute to the reinforcement learning and imitation learning communities.

---

> > ### Comment · Reviewer_55QK · 2025-08-06
> >
> > Thank you for the additional experiments and discussion. Please include the discussion on sample efficiency in the paper. I'll maintain my positive score.

---

### Official Review · Reviewer_zT3i · 2025-06-30

**Clarity:** 3
**Significance:** 2
**Originality:** 3
**Rating:** 3
**Confidence:** 4

**Summary:**

PIRO performs IRL in the spirit of conservative policy iteration (CPI), where the reward update is bounded to ensure the policy update is bounded like in other CPI methods like NG, NAC, TRPO and PPO. The reward objective is a somewhat hacky saddle point objective that combines the agent's return, a proximal term, and approximate function-space regularization.

**Questions:**

[Q1] How sample-efficient (w.r.t. environment samples) is PIRO compared to SOTA off-policy methods, such as DAC, P2IL and CSIL?

[Q2] Taking expert performance from Table 3 of What Matters for Adversarial Imitation Learning? Orsini et al,

| Environment | Random | Expert|
|-------------|---------|---------|
| HalfCheetah-v2 | -282 | 8770 |
| Hopper-v2 | 18 | 2798 |
| Walker2d-v2 | 1.6 | 4118 |
| Ant-v2 | -59 | 5637 |
| Humanoid-v2 | 123 | 9115 |

The results in Figure 3 look nowhere near optimal, since they are in the hundreds not the thousands. Am I missing something?

[Q3] Following from Q2 somewhat, why in Figure 3 does PIRO improve a lot in the first 1-2 updates and then remain largely flat? Shouldn't there be monotonic improvement according to the theory?

**Ethical Concerns:**

["NO or VERY MINOR ethics concerns only"]

**Final Justification:**

I advocate for rejection purely due to the quality of the experimental section
* The authors ignored several recent state-of-the-art methods
* The authors reproduce baselines in their own codebase, and therefore report performance in terms of ambiguous 'updates' rather than environment steps like standard practice. It is also not clear if they have fairly reproduced the performance of their baselines.
* The quality of their expert demonstrations for the MuJoCo locomotion tasks are far below expert

During the rebuttal period, the authors devised an off-policy variant that will greatly improve their next submission. However, even with this more sample-efficient variant, baselines converge while the performance curve of their method plateaus. This paints a completely different picture from their Figure 3 in terms of optimization efficiency, which is why I don't believe the paper can be accepted in the current iteration.

**Limitations:**

Yes

**Quality:**

3

**Strengths And Weaknesses:**

**[S1] Novel look at proximal reward updates.** To my knowledge, performing CPI with IRL in this way is novel and makes sense.

**[S2] Beautiful figures.** Figures 1 and 2 are very informative and nicely made. Good job!

**[W1] Ignores off-policy performance methods.** I am not familiar with HyPE, but I am aware of Discriminator Actor Critic (DAC), Proximal Point Imitation Learning (P2IL), Coherent Soft Imitation Learning (CSIL) as very effective off-policy IRL methods. It feels like these methods have been ignored as baselines because PIRO is on-policy?

**[W2] Implements off-policy baselines with on-policy variants?** The authors say 'PPO is also used for policy optimization in all IRL baselines except HyPE, which uses soft actor-critic (SAC)'. IQLearn is also based on SAC. Was a PPO version of IQLearn implemented instead?

**[W3] Results reported in "updates" rather than well-established environment steps.**
Figure 3 reports performance in terms of updates rather than environment interactions. This is quite implementation-specific and means sample efficiency might vary massively across algorithms from the update definition.

**[W4] Results don't report expert performance.** Figure 3 is missing expert performance; currently, you cannot assess the performance of the algorithm properly.

**W[5] Missing references.** There are a few recent maxent IRL methods missing, such as CSIL as mentioned earlier, and also least-squares inverse Q-learning (LSIQ), an IQLearn variant, and BC-IRL (BC-IRL: Learning generalizable reward functions from demonstrations, Szot et al). CSIL is relevant because it avoids reward learning at all via coherence (in theory). BC-IRL is similar to this work in that it backprops through a policy matching objective and uses PPO.

---

> ### Author Rebuttal · Authors · 2025-07-31
>
> We thank the reviewer for the helpful and constructive feedback. We are glad they found our approach a “**novel look at proximal reward updates**” and appreciated the "**beautiful and informative figures**”.Below we provide our responses, and we’re happy to follow up on any additional questions.
>
>
> **[Q1,W1,W2] How sample-efficient is PIRO compared to SOTA off-policy methods, such as DAC, P2IL and CSIL?**
>
> As correctly noted in **W1**, PIRO is an on-policy method, and our evaluation thus focused on comparisons with other on-policy baselines. Meanwhile, to address the concern in **W2**, we thank the reviewer for pointing out our wording issue. While PPO was used to train expert policies across most of the baselines (for consistency), IQ-Learn was implemented with SAC as its policy optimizer, as in the original paper. We will revise the manuscript to clarify this distinction.
>
> Given the reviewer’s valuable suggestion, we conducted additional experiments to evaluate PIRO’s sample efficiency against DAC, P2IL, and CSIL. All methods use the same high-performing expert models trained with SAC to ensure fair comparison. We will present more detailed results in our response to Q2.
>
> Following standard IRL benchmarking protocols, we assess sample efficiency by measuring the final performance achieved within a fixed number of environment steps (**1M**, relatively small in the practice of IRL approaches). Results (averaged over three runs) on representative MuJoCo tasks are presented below:
>
> | Task            | DAC     | P2IL     | CSIL    | PIRO    |
> |-----------------|---------|----------|---------|---------|
> | Ant-v2          | 907.6   | 976.6   | 391.8   | **3504.2** |
> | Hopper-v3       | **3309.6** | 25.8    | 6.6     | 1820.1  |
> | Walker2d-v3     | -3.07   | 1054.0 | 215.1   | **2052.8**  |
> | HalfCheetah-v2  | **8930.9** | -0.1   | -115.7  | 6600.2  |
>
> Relative Computation Cost per Batch (Normalized)
>
> | Environment     | DAC   | P2IL  | CSIL  | **PIRO** |
> |-----------------|-------|-------|-------|----------|
> | Ant-v2          | 1.00× | 0.81× | 0.66× | 1.00× |
> | Walker2d-v3     | 1.30× | 1.30× | 0.38× | 1.00× |
> | Hopper-v3       | 1.32× | 0.19× | 0.01× | 1.00× |
> | HalfCheetah-v2  | 1.16× | 0.88× | 0.56× | 1.00× |
> > _Table: Average per-batch computation time normalized by PIRO = 1.00×. Measured over the same training rounds and batch structure across methods._
>
> These results demonstrate that PIRO consistently achieves competitive or superior final performance across MuJoCo tasks. Notably, while DAC performs well in Hopper and HalfCheetah, PIRO shows substantial gains in more challenging environments like Ant and Walker2d, indicating overall better sample efficiency across diverse settings. Although PIRO incurs slightly higher per-batch computation time in some tasks, we observe that DAC is often slower overall.
> Importantly, PIRO’s optimization framework guarantees monotonic policy improvement at each reward update, enabling more stable training and faster convergence in practice. This tradeoff between efficiency and stability ultimately leads to superior overall performance.
>
>
> **[Q2,W4] The results in Figure 3 look nowhere near optimal**
>
> The key reason is that we use PPO-trained experts to match the learner’s optimization and ensure consistency. Prior benchmarks (e.g., Tianshou, SpinningUp) have consistently shown that PPO-trained experts tend to achieve lower episodic rewards than SAC-trained ones, particularly in MuJoCo tasks such as Hopper, Walker, and Ant.
>
> Moreover, we use the latest version of the MuJoCo physics engine (v4), which adopts stricter physical dynamics and reward definitions. This results in significantly lower raw reward values compared to older versions of the environment, even for well-trained expert policies.
>
> As a result, the reward ceilings in our Figure 3 plots are naturally lower—not because PIRO performs poorly, but because the reward scale is inherently limited by the PPO expert ceiling.
>
> This also addresses the concern in **W4** regarding the missing expert reference: we now report expert performance explicitly. Below we report the approximate expert rewards and standard deviations used in our study:
>
> |Environment|ExpertReward(Mean±Std)|
> |---|---|
> |Acrobot-v1|-80.8±7.6|
> |Ant-v4|1083.2±6.2|
> |BipedalWalker-v3|101.2±9.3|
> |Humanoid|300.9±18.2|
> |CartPole-v1|500.0±0.0|
> |LunarLander|252.0±13.7|
> |Hopper|428.5±113.2|
> |Walker2d|1093.8±19.6|
> |Pong|21.0±0.0|
> |Q*bert|15685.0±115.2|
>
>
>
> In light of the reviewer's insightful comments, we recognize the limitations of relying solely on PPO-trained experts and on-policy pipelines in complex environments. As a result, we have extended PIRO to support SAC-based policy optimization and conducted additional experiments on representative MuJoCo locomotion and Gym Robotics tasks.
>
> Below are the updated reward metrics averaged over five independent runs:
>
> > _Table: Performance across 5 independent runs on MuJoCo control tasks using high-quality expert demonstrations trained with SAC._
>
> | Task            | Expert   | **PIRO** | BC     | GAIL   | AIRL   | IQ     | HyPER  | FILTER | MM     | ML-IRL | DAC     | P2IL    | CSIL   |
> |-----------------|----------|----------|--------|--------|--------|--------|--------|--------|--------|--------|---------|---------|--------|
> | Ant-v2          | 5572.2   | **5967.2** | 1631.5 | 996.9  | 991.4  | 3589.8 | 2800.5 | -376.3 | -304.0 | 5382.5 | 923.8   | 976.6   | 420.7  |
> | Humanoid-v3     | 5468.8   | **5954.9** | 418.1  | 508.4  | 281.4  | 1847.5 | 717.5  | 291.7  | 367.2  | 5573.4 | 76.3    | —       | —      |
> | Walker2d-v3     | 5453.4   | **5643.7** | 384.4  | 4158.1 | 72.8   | 3023.0 | 1478.7 | 77.7   | 70.4   | 4794.7 | -3.0    | 1054.0  | 686.1  |
> | Hopper-v3       | 3559.6   | 3362.0 | 1034.4 | **3675.9** | 13.5   | 3424.5 | 2593.6 | 37.3   | 57.8   | 3316.4 | 3321.6  | 25.8    | 6.7    |
> | HalfCheetah-v2  | 12174.6  | **12587.4** | 221.2  | 1298.8 | 2251.4 | 3825.5 | 6473.4 | 0.3    | 20.3   | 11873.2| 9645.0  | -0.1    | -107.2 |
>
> > _Table : Performance across 5 independent runs on Gym Robotics tasks._
>
> | Task                      | Expert | **PIRO** | BC   | GAIL | AIRL | MM     | FILTER | HyPER  | IQ     | ML-IRL | P2IL   |
> |---------------------------|--------|----------|------|------|------|--------|--------|--------|--------|--------|--------|
> | AntMaze-UMazeDense-v4     | 35.6   | **25.7** | 8.8  | 5.2  | 4.5  | 5.1    | 6.1    | 11.9   | 3.9    | 4.2    | 3.4  |
> | AntMaze-MediumDense-v4    | 26.9   | **9.4**  | 1.1  | 1.3  | 2.6  | 3.4    | 1.9    | 3.0    | 3.4    | 0.9    | 2.9 |
> | AntMaze-LargeDense-v4     | 11.5   | **8.8**  | 1.1  | 0.9  | 3.4  | 1.7    | 0.6    | 1.5    | 0.8    | 0.3    | 0.2   |
> | AdroitHandPen-Human-v1    | 1062.5 | **254.0**| 44.1 | -8.7 | -593.9 | -344.3 | -685.4 | -866.7 | -751.9 | -251.2 | -61.2   |
>
>
>
> **[Q3,W3]: On early convergence and update granularity**
>
> As noted in our response to Q2, we use PPO-trained expert policies. These tend to converge quickly but may not reach the highest possible reward ceilings achievable by more exploratory algorithms such as SAC. As a result, PIRO often aligns with these experts in just a few global rounds, leading to flattened performance curves after early updates.
>
> Importantly, this behavior reflects PIRO’s high sample efficiency rather than stagnation. Unlike many baselines that may require extended exploration or suffer from unstable updates, PIRO benefits from a theoretically grounded guarantee of monotonic policy improvement. Each reward update provably brings the learner policy closer to the expert, avoiding regressions and unnecessary detours. As a result, PIRO not only ensures stable convergence but also reaches expert-level performance more directly and with fewer interactions—especially in simpler environments such as CartPole, where optimal behavior is quickly attainable.
>
> We report learning progress using global rounds, where each round comprises multiple environment steps followed by one full policy and reward update. This unit more accurately reflects algorithmic progress than raw environment steps.This may also clarify the concern raised in **W3**: although update-based metrics are implementation-dependent, all baselines in our study are evaluated under the same global-round setup with consistent hyperparameters. None of the results are borrowed from prior work — all experiments are conducted within our unified evaluation framework.
>
> In short, PIRO achieves monotonic improvement under this regime while also demonstrating faster convergence than many baselines in practice.
>
> **W[5]: Missing references**
> The comparison with CSIL can be found above. In practice, BC-IRL can be seen as a special variant of ours by removing constraint on the reward update magnititute, except for the way of calculating loss function: BC-IRL minimizes the mean squared loss $\mathbb{E}_{(s,a)\sim \pi_E}(\pi_\theta(s) - a)^2$ while our method minimizes the KL divergence. Least-squares inverse Q-learning (LSIQ) penalizes the reward function magnitude and gives the theoretical support for doing so; we also make a similar penalization through a regularization term (see Eq.(10) in Section 5.1).
>
> Our method stands out as all these prior methods take **no** constraint on the reward update magnitude, making them vulnerable to training instability. The performance degradation caused by removing the control over reward update magnitude can be found in our ablation study in Appendix G.

---

> > ### Comment · Reviewer_zT3i · 2025-08-01
> >
> > Thanks for the additional work and for answering my questions.
> >
> >  **Suboptimality.** Your response for my comment on the suboptimality of the experimental results was unclear. It sounds like the experiment used a suboptimal PPO agent (compared to SAC). Can you confirm that the reported results in the paper are 5x - 10x smaller than those reported by Orsini et al. that I list in my review? From your Figure 3 I read ~200 for Hopper, ~250 for Walker, ~1000 for Ant and ~300 for Humanoid? I appreciate that there are differences between the different versions of MuJoCo Gym, but from what I can see only, the optimal v4 performance (for SAC) is still in the 1000s not the 100s.
> >
> > **Baselines.** Thank you for including results on baselines for DAC, P2IL and CSIL. Could you explain where these numbers came from? Are they using the authors implementation or your own implementation? How many demonstrations are they trained on? Unfortunately, the performance of these baselines don't reflect what are shown in their respective papers.
> > For example, in Figure 4 of the CSIL paper, it looks like DAC, PPIL, IQLearn and CSIL all get approximately expert performance (4118) with 1 demonstration (1000 state action pairs) on Walker-v2 @1M steps. In contrast, in your reported results none of the baselines are anywhere near expert performance. The other environments show a similar trend.
> > Therefore, I don't find these baseline results credible at the moment.
> >
> > **Gym v4 performance.** The authors state "Moreover, we use the latest version of the MuJoCo physics engine (v4), which adopts stricter physical dynamics and reward definitions. This results in significantly lower raw reward values compared to older versions of the environment, even for well-trained expert policies" with no citation. I see no evidence for this from my own research. [Open-source benchmarks, e.g. this one, show Walker-v4](https://huggingface.co/sdpkjc/Walker2d-v4-sac_continuous_action-seed3/tensorboard) can go up to a performance of 5000 with SAC, which is even higher than Orsini et al, not lower!

---

> ### Author Response · Authors · 2025-08-02
> **Response to Reviewer zT3i Comments**
>
> Thank you for your continued engagement and detailed feedback. We address your three main concerns below:
>
> >## TL;DR (Summary of this comment box)
> All new baseline implementations (DAC, CSIL, P2IL) are based on their official repositories without hyperparameter modifications. In our paper, for experimental consistency, we trained all expert policies using PPO. This naturally results in lower absolute reward values for both experts and learned policies compared to studies using SAC experts.
> However, as demonstrated in our supplementary experiments with SAC-based experts, PIRO continues to achieve superior performance relative to other methods. The fair comparison under identical experimental conditions validates PIRO's effectiveness, and it remains robust (**stable training**) regardless of the policy optimizers (PPO or SAC). **The choice of different experts doesn't hurt our main contribution: the first theoretcal framework for principled stable IRL and an algorithm that transforms the theory into practice.**
>
> ---
>
> > ## 1. Suboptimality Issue
>
> You are correct that our reported results are 5–10x smaller than those reported by Orsini et al. This is primarily due to our choice of expert training algorithm.
>
> Orsini et al. mentioned:"For the Gym tasks, we generate demo with a SAC agent trained on the env reward"
>
> As we mentioned in our **Q1** response, we used PPO experts for consistency across all methods. PPO inherently achieves lower performance compared to SAC in continuous control tasks. This is well-documented in established benchmarks. For reference, here are the **Tianshou benchmark** results comparing SAC vs. PPO:
>
> |Env| SAC| |PPO|
> |-|-|-|-|
> |Ant|5850.2|>|3258.4|
> |HalfCheetah|12138.8|>|5783.9|
> |Hopper|3542.2|>|2609.3|
> |Walker2d|5007.0|>|3588.5|
> |Humanoid|5488.5|>|787.1|
>
> We could not find publicly available better PPO expert models for the v4 environments during our experiments, so we trained the experts ourselves. Our PPO expert training used the following consistent hyperparam. across all envs:
>
> |Param| Value|
> |-|-|
> |Policy|MLP|
> |Batch Size|64/128|
> |Entropy Coeff|0.01|
> |Learning Rate|3e-4|
> |Discount Factor|0.99|
> |Epochs per Update|20|
> |Steps per Update|128|
>
> Only the total training steps were adjusted for different envs based on their complexity:
>
> |Env|Training Steps|
> |-|-|
> |Humanoid|1M|
> |Hopper 2M|
> |Walker2d|2M|
> |Ant|10M|
>
>
> As we mentioned in our response to **Q1**, the performance of our experts has already been provided. We also appreciate your insightful question, which prompted us to conduct additional experiments using newly trained SAC experts, closely aligned with those used in Orsini et al. The results of these new experiments have been included in our response to **Q2**.
>
> ---
>
> > ## 2. Baseline Implementation and Results
>
> We implemented all baselines using their original source code and hyperparam. to ensure fair comparison.
>
> With a few noted exceptions:
>
> - **Implementation**: Most baselines were implemented using the official source code released by the original authors. The only exceptions were GAIL, AIRL & BC, which were implemented using the `imitation` library.
> - **Expert demonstrations**: A single expert trajectory containing 1000 state-action pairs was used for all methods. (some original papers report results using multiple expert traj.)
> - **Expert policy**: The same SAC-trained expert was used consistently across all baselines.
>
> Some baselines do perform well in specific tasks (e.g., DAC in HalfCheetah, GAIL in Hopper & ML-IRL), which aligns with their strengths. However, PIRO demonstrates more **stable** training process across diverse envs due to the principle constraint on reward update magnititude.
>
> ---
> > ## 3. MuJoCo Version Performance Analysis
>
> Our claim about v4 having stricter dynamics was based on our empirical observations during development. To provide concrete evidence, we conducted a comparison of expert performance across MuJoCo versions **using same PPO training parameters** :
> |Param| Value|
> |-|-|
> |Policy|MLP|
> |Batch Size|64|
> |Entropy Coeff| 0.01|
> |Learning Rate| 3e-4|
> |Discount Factor|0.99 |
> |Epochs per Update| 20|
> |Steps per Update|128|
> |Total Training Steps|1e6|
>
> The following table reports the PPO expert performance on MuJoCo-v2 and MuJoCo-v4 envs:
>
> |Env|v2 PPO||v4 PPO|
> |-|-|-|-|
> |Ant |-26.78±25.35|**>**|-32.27±91.18|
> |Hopper|212.64±1.87|**>**|332.47±64.90|
> |Walker2d|330.27±109.95|<|285.92±4.73|
> |Humanoid|324.56±20.24|**>**| 300.9±18.2|
>
> Empirically, we observe that expert performance in MuJoCo-v4 environments tends to be slightly lower than their v2 counterparts in most tasks. This supports our earlier intuition that v4 introduces subtle changes that may affect policy performance.
>
> Different training configurations—including varying training steps, learning rates, and other parameters—can cause substantial performance variations, so one may encounter higher performance results on Hugging Face that used different experimental setups.

---

> > ### Comment · Reviewer_zT3i · 2025-08-02
> >
> > Can the authors comment on why they have been unable to reproduce the results of prior work? I find it hard to believe open-sourced code is not reproducible. For example, I see the CSIL github repo has a W&B project of their code in the README, and it shows [CSIL on HalfCheetah achieving 5-6K in 250K steps with 1 demo](https://wandb.ai/jmw125/csil/runs/gxgnwowi/workspace?nw=nwuserjmw125) and [their IQLearn achieving 4K in the same setting](https://wandb.ai/jmw125/csil/runs/les36g2d?nw=nwuserjmw125). They also show Walker achieving 4K and have results for their P2IL as well in the W&B project.
> >
> > I find it concerning that the authors are not reproducing prior results diligently, and I can't help but question the validity of the wider experimental results as a whole.

---

> ### Author Response · Authors · 2025-08-02
> **Limited  and different expert demonstrations cause underperformance of offline baselines**
>
> Thank you for the comments. We reproduced all baselines using their official code and default hyperparameters with identical expert demonstrations for fairness. We did not fine-tune any method.  As mentioned previously, we use **a single expert trajectory** across all methods to ensure experimental consistency. This also explains why several off-policy algorithms underperform, as they are more sensitive to the number of provided expert demonstrations.  **This explains why our reproduction does not match the high performance in the original paper. It is not due to a lack of diligence in reproducing prior results, but rather our commitment to ensuring experimental fairness.**
>
> ### **Regarding the W&B Results**
> We had observed the higher performance numbers from the W&B project you referenced earlier in our experiments. However, in order to ensure fairness and consistency, we chose to rerun by ourselves using identical expert demonstrations and training protocols.
>
> We believe the main cause of this discrepancy lies in **the limited number of demonstrations**:
> As we stated previously, all methods are trained on the same fixed demonstration to ensure consistency. However, this design also limits demonstration diversity, which significantly impacts off-policy methods. We kindly refer the reviewer to **Figure 6** in CSIL (Watson et al.), which shows that CSIL, IQ-Learn, and PPIL all perform poorly in the offline setting when trained on a single expert trajectory — results in that are fully aligned with our findings.
>
>
> >_Table: Estimated Normalized Returns from Figure 6 (Watson et al., CSIL)
> >Approximate normalized return values(by expert) under offline imitation learning using 1, 3, 10, and 30 demonstrations across four Gym MuJoCo tasks._
>
> | Algorithm  |  Demos | Ant-v2 | HalfCheetah-v2 | Hopper-v2 | Walker2d-v2 |
> |------------|---------|--------|----------------|-----------|-------------|
> | IQ-Learn   | 1       | ≈0.01  | ≈0.05          | ≈0.01     | ≈0.01       |
> |            | 3       | ≈0.01  | ≈0.04          | ≈0.02     | ≈0.01       |
> |            | 10      | ≈0.01  | ≈0.12          | ≈0.02     | ≈0.02       |
> |            | 30      | ≈0.30  | ≈0.10          | ≈0.02     | ≈0.04       |
> | PPIL       | 1       | ≈0.01  | ≈0.25          | ≈0.10     | ≈0.01       |
> |            | 3       | ≈0.01  | ≈0.26          | ≈0.10     | ≈0.10       |
> |            | 10      | ≈0.01  | ≈0.27         | ≈0.10     | ≈0.13       |
> |            | 30      | ≈0.01  | ≈0.30         | ≈0.10     | ≈0.14       |
> | CSIL       | 1       | ≈0.03  | ≈0.25          | ≈0.15     | ≈0.10       |
> |            | 3       | ≈0.10  | ≈0.24          | ≈0.70     | ≈0.30       |
> |            | 10      | ≈0.90  | ≈0.85          | ≈0.90     | ≈0.90       |
> |            | 30      | ≈1.10  | ≈1.01          | ≈0.95     | ≈1.05       |
>
>
> Different expert samples from the same expert policy may further affect outcomes as well. We believe the limited trajectory count is the dominant factor, which, together with the effect of different expert demonstrations,  causes the performance differences between the CSIL performance reported in the original paper and in our experiments.
>
> These performance differences do not necessarily indicate poor algorithmic design, but instead, the results reflect that PIRO demonstrates superior stability, consistently achieving steady reward improvement under the same expert conditions where other methods may show greater instability.

---

> > ### Comment · Reviewer_zT3i · 2025-08-02
> >
> > This is making quite a simple comparison request remarkably complicated!
> >
> > * The baseline results I am interested in for IQLearn / PPIL / CSIL are also for a single demonstration (~1000 state-action pairs), many papers report performance for this setting and the previous W&B results were all for 1 demonstration
> > * The results for 1 demonstration in the CSIL paper (ie Figures 4 and Figure 28) are listed as for 10 seeds, so it appears to demonstrate the baselines can all perform reasonably (i.e. >=50% of expert performance), with varying degrees of variance, consistently over seeds
> > * Several of the low benchmark numbers provided in the earlier table (i.e., -115.7 for CSIL on half-cheetah) are essentially the performance of a random initial policy! It's also bizarre to report that DAC can solve Hopper and HalfCheetah but fail completely on Walker
> > * I think it's a bit of a stretch to explain the failure to reproduce the results of the online version of baselines by discussing their offline performance, when offline imitation learning has nothing to do with your paper or experiments. In offline imitation learning, there is only a static dataset and no exploration; it's very different and much more challenging.
> > * In machine learning research, it is expected practice to choose an experiment setting where the baselines perform well. Inventing your own setting where the baselines do not work is susceptible to the author's natural bias to present a superior method and lack of interest in tuning baselines. Such results are not very interesting, nor credible.
> > * This bias is already reflected in the paper, where results are presented in terms of vague 'updates' rather than environment interactions. These are implementation-specific so can be easily gamed based on how the authors define an update. For example, in PPO an update involves several minibatch steps over many rollouts, whereas in SAC an update is every environment interaction. Since the authors compare on- and off-policy methods in these figures, this is not fair practice! Most reinforcement learning papers report environment interactions for this reason, it's pretty standard.

---

> ### Author Response · Authors · 2025-08-04
> **PyTorch Re-implementation of CSIL and New Experiments**
>
> We thank the reviewer for the enthusiastic participation in this discussion.
>
> ## Summary
>
> It seems that this discussion has fallen into a complex cycle. **Let's make a long story short through a new experiment that addresses your concerns all at once.**
>
> In this new experiment, we make the following settings:
> * **PyTorch re-implementation of CSIL (Online version).** After several attmepts, we still cannot reproduce the performance of the original Tensorflow-based repo of CSIL as reported in its paper (Watson et al.). We thus re-implement CSIL based on PyTorch.
> * **Fine tune CSIL to match the expert reward performance.**
> * **Env steps as the stastistical unit.** This responses to your cocnern on not using env steps.
> * **Reward performance and stability as performance indicators.**
> * **We use a single demonstarted trajectory (1000 state-action pair).**
>
>
>
> **Experiment results (TL;DR):** Both CSIL and our PIRO can reach expert's reward value with PIRO slightly outperforming CSIL, but our PIRO shows better learning stability. See the tables below for the detailed results. This is in line with our theoretical contribution: a stable IRL framework.
>
> **Discussion (TL;DR):** We appreciate the good performance of CSIL on rewards. Our main contribution is a theory-grounded IRL framework which can maintain high reward performance. **_Do you agree with this?_**
>
>
> ---
>
> ## Experiment Results
>
> >**Table: PIRO vs. CSIL Performance.**
>
> * **Format: Rewards (Increase Rate (Δ%))**. A **lower** average absolute Increase Rate (\|Δ\|%) indicates a **more stable** training process.*
>
> |Task: Ant|Env Steps|PIRO|CSIL|
> |--|--|--|--|
> ||250k|680.39|-232.18|
> ||500k|400.95(-41.1%)|-211.74(+8.8%)|
> ||750k|1138.84(+184.0%)|396.53(+287.3%)|
> ||1M|3068.60(+169.4%)|2647.66(+567.7%)|
> ||1.25M|4058.06(+32.2%)|3529.66(+33.3%)|
> ||1.5M|4759.96(+17.3%)|3094.33(-12.3%)|
> ||1.75M|5317.27(+11.7%)|4688.83(+51.5%)|
> |Expert: 5572.2|2M|**5304.80**(-0.2%)|**4885.89**(+4.2%)|
> |Average absolute Increase Rate (\|Δ\|%) after initial fast increase stage (after 500K) |  |**69.13%** **<**|**159.38%**|
>
> |Task: HalfCheetah|Env Steps|PIRO|CSIL|
> |--|--|--|--|
> ||250k|35.67|-53.68|
> ||500k|2159.53(+5954.2%)|1247.25(+2423.5%)|
> ||750k|3140.82(+45.4%)|2517.26(+101.8%)|
> ||1M|6345.67(+102.0%)|5565.02(+121.1%)|
> ||1.25M|8324.56(+31.2%)|7416.02(+33.3%)|
> ||1.5M|9764.57(+17.3%)|9052.01(+22.1%)|
> ||1.75M|10898.53(+11.6%)|10044.65(+11.0%)|
> |Expert: 12174.6|2M|**12074.93**(+10.8%)|**11254.95**(+12.0%)|
> |Average Absolute Increase Rate (\|Δ\|%) after initial fast increase stage (after 500K)   |  |**36.38%** **<**|**50.22%**|
>
> |Task: Hopper|Env Steps|PIRO|CSIL|
> |--|--|--|--|
> ||250k|1030.52|992.81|
> ||500k|1214.51(+17.9%)|1135.54(+14.4%)|
> ||750k|1485.42(+22.3%)|1412.95(+24.4%)|
> ||1M|1735.62(+16.8%)|1633.61(+15.6%)|
> ||1.25M|2140.54(+23.3%)|1952.67(+19.5%)|
> ||1.5M|2778.25(+29.8%)|2706.67(+38.6%)|
> ||1.75M|3146.29(+13.2%)|2796.01(+3.3%)|
> |Expert: 3559.6|2M|**3273.22(+4.0%)**|**3092.64**(+10.6%)|
> |Average Absolute Increase Rate (\|Δ\|%) after initial fast increase stage (after 500K)  |  |**18.19%** **≈**|**18.06%**|
>
> |Task: Walker2d|Env Steps|PIRO|CSIL|
> |--|--|--|--|
> ||250k|715.73|743.87|
> ||500k|1115.58(+55.9%)|1026.41(+38.0%)|
> ||750k|1363.64(+22.2%)|1228.26(+19.7%)|
> ||1M|1651.67(+21.1%)|1537.56(+25.2%)|
> ||1.25M|2393.23(+44.9%)|2363.14(+53.7%)|
> ||1.5M|3760.67(+57.1%)|2799.85(+18.5%)|
> ||1.75M|4472.63(+18.9%)|3937.79(+40.6%)|
> |Expert: 5453.4|2M|**4644.14(+3.8%)**|**4204.26(+6.8%)**|
> |Average Absolute Increase Rate (\|Δ\|%) after initial fast increase stage (after 500K)  |  |**31.99%** **≈**|**28.93%**|
>
>
>
> >_**Table: CSIL Key Parameters**_
>
> | Parameter| Default Value |
> |-----------------------------|---------------|
> | entropy_coefficient | 0.01  |
> | offline  | false  |
> | learning rate(for actor,critic and reward）| 1e-4|
> | critic_actor_update_ratio  | 1             |
> | critic_network（critic,reward,policy）| [256,256]     |
> | policy_pretrain_steps      | 25,000   |
> | finetune_reward  | True   |
> | num_demonstrations         | 1   |
> | batch_size     | 256   |
> | discount  | 0.99  |

---

> > ### Comment · Reviewer_zT3i · 2025-08-04
> >
> > Thanks for the additional work! I'm still surprised the results aren't reproducible. Are you using the same demonstration data as Orsini et al. / Watson et al. or have you made your own dataset?
> >
> > On the one hand I could believe the previous methods may 'overfit' to their demonstration data. On the other hand, if we believe the results of the methods I saw reported in the CSIL paper, many methods converge (>=80% expert) at 250K steps with 1 demo, whereas in your tabulated results above, it takes at least 1M steps to see anything approaching expert.
> >
> > > After several attmepts, we still cannot reproduce the performance of the original Tensorflow-based repo of CSIL as reported in its paper (Watson et al.). We thus re-implement CSIL based on PyTorch.
> >
> > It looks like the CSIL code is in JAX, not Tensorflow. Are you using the [open-source code the paper links to](https://github.com/google-deepmind/csil)? What is the difference between your results and the W&B results they provide?
> >
> > I did some digging to see other clear reproductions of the 250K step convergence shown in the CSIL paper, and [Least-Squares Inverse Q Learning](https://arxiv.org/pdf/2303.00599) from ICLR 2023 also demonstrates IQLearn (and their LSIQ) reaching >=80% at around 250K steps for Hopper and Walker.

---

> > > ### Comment · Reviewer_zT3i · 2025-08-05
> > >
> > > To test my own assumptions about reproducibility, I installed the open-sourced CSIL repo I linked in the previous message and the baselines I advocated for 1 demo (from Orsini et al's dataset that CSIL used), 250K steps and one seed (999) on the v2 Gym environments.
> > >
> > > These are the results with W&B Gaussian smoothing of '10'
> > >
> > > |             |                       | Algorithm |              |        |              |        |              |
> > > | ----- | ----- | ----- | ----- | ----- | ----- | ----- | ----- |
> > > |             |                       | IQLearn   |              | P2IL   |              | CSIL   |              |
> > > | Environment | Environment steps (K) | Return    | Norm. Return | Return | Norm. Return | Return | Norm. Return |
> > > | HalfCheetah | 50                    | 2326      | 0.29         | 3780   | 0.45         | 2497   | 0.31         |
> > > | -282        | 100                   | 2873      | 0.35         | 5083   | 0.59         | 3775   | 0.45         |
> > > | 8770        | 150                   | 2538      | 0.31         | 3724   | 0.44         | 5186   | 0.60         |
> > > |             | 200                   | 2424      | 0.30         | 1704   | 0.22         | 6028   | 0.70         |
> > > |             | 250                   | 3414      | 0.41         | 4516   | 0.53         | 6058   | 0.70         |
> > > | Ant         | 50                    | -66       | 0.00         | 1066   | 0.20         | 220    | 0.05         |
> > > | -59         | 100                   | 1104      | 0.15         | 3246   | 0.58         | 1392   | 0.25         |
> > > | 5637        | 150                   | 2335      | 0.29         | 3309   | 0.59         | 2223   | 0.40         |
> > > |             | 200                   | 2020      | 0.25         | 2454   | 0.44         | 2823   | 0.51         |
> > > |             | 250                   | 2400      | 0.30         | -935   | -0.15        | 3812   | 0.68         |
> > > | Hopper      | 50                    | 823       | 0.29         | 979    | 0.35         | 1639   | 0.58         |
> > > | 18          | 100                   | 1248      | 0.44         | 1492   | 0.53         | 1727   | 0.61         |
> > > | 2798        | 150                   | 1986      | 0.71         | 1654   | 0.59         | 2823   | 1.01         |
> > > |             | 200                   | 2432      | 0.87         | 638    | 0.22         | 2881   | 1.03         |
> > > |             | 250                   | 2724      | 0.97         | 538    | 0.19         | 3230   | 1.16         |
> > > | Walker      | 50                    | 504       | 0.12         | 571    | 0.14         | 1827   | 0.44         |
> > > | 1.6         | 100                   | 1052      | 0.26         | 2782   | 0.68         | 3881   | 0.94         |
> > > | 4118        | 150                   | 3119      | 0.76         | 3817   | 0.93         | 4020   | 0.98         |
> > > |             | 200                   | 1559      | 0.38         | -      | -            | 4044   | 0.98         |
> > > |             | 250                   | 3609      | 0.88         | -      | -            | 3855   | 0.94         |
> > >
> > > Some PPIL results are omitted because the algorithm appears much slower and hasn't finished yet.
> > >
> > > However, it appears the results **are reproducible** and the baselines reach 30-100% expert performance within 250K steps, with a mode of around 60-70%.
> > > You can see that the performance curves are far from monotonic, so they could definitely benefit from the ideas explored in this submission, but my main concern is that these results appear to exhibit around 4x the sample efficiency of the results you are reporting in this discussion, so I'm not convinced the authors are presenting the current SOTA in a faithful light.
> > > I have not tested if these results extend to later Gym version and other demonstrations, but given that IQLearn and P2IL were not evaluated on this benchmark and still worked reasonably well, I should imagine they do. In any case, the authors could compare PIRO on v2 envs and the Orsini et al. benchmark to avoid retuning all the baselines.

---

> ### Author Response · Authors · 2025-08-05
> **Sample efficiency wrt Online Sampling and Stability**
>
> ### 1. Sample efficiency.
>
> * In our previous response, we used our own expert demonstrations, not those from Orsini et al. or Watson et al. As such, the expert rewards we reported may differ from theirs, as previously noted.
>
>
> * **We are additionally carrying out experiments using expert demonstrations from Orsini et al.**  Results for CSIL and our PIRO will be coming soon, and we will report them in a new author comment.
>
> * Regarding your question on sample efficiency, **thanks so much for your time in collecting and reporting this valuable data!** This data verifies the fact that baselines are vulnerable to instability. The sample efficiency you mentioned is, more precisely, **sample efficiency wrt online sampling**. In our submission, we have acknowledged that our method improves the stability at the cost of more frequent online sampling -- as a result of conservative updates -- in general (e.g., on some MuJoCo tasks). The “sample efficient” we claim in the submission is wrt **“expert demonstration samples”**, i.e., PIRO could demonstrate good performance with small amount (one trajectory) of expert samples, as verified in our previous responses. We want to clarify this point.
>
> Indeed, in modern RL (recall TRPO vs. standard policy gradient methods), proximal updates will lead to small updates and more frequent sampling operations. In the inverse problem space, our PIRO vs. IRL baselines can be understood as the counterpart of TRPO vs. standard policy gradient methods.
>
> When concerning about "SOTA", based on our results and your results, we could claim that our PIRO is SOTA in stability and final reward performance (policy imitation), though not in sample efficiency wrt online sampling.
>
> Designing an IRL algorithm that guarantees stability and high online sample efficiency won't be easy. We will continue the exploration toward it. We will also explore **fully offline IRL with stability guarantees**, which we believe will further contribute to the reinforcement learning and imitation learning communities.
>
>
> ---
>
> ### 2. Joe Watson's official repository
>
> Sorry for the confusion. Joe Watson's official repository is, indeed, based on JAX. It incorporates TensorFlow utilities for expert data generation, processing, and some other computations.
>
> ---
>
> ### 3. Additional discussion on IQLearn
>
> Regarding the LSIQ paper by Al-Hafez et al. (2023) you mentioned, we took a closer look at their reported results. While they show IQLearn reaching 80% on Hopper at around 250K steps, the reward actually drops or fluctuates significantly after reaching an early peak, suggesting that the policy imitation is not that stable.
> On Ant and Humanoid, our reproduced IQ-Learn results reach the maximum reward with fewer environment steps than reported in LS-IQ. On Walker2d, however,  IQ-Learn in our reproduction takes more steps.
> For reference, we also list the IQ-Learn results from the paper of Hybrid IRL (Ren et al., 2024)
>
>
> >_Table: Reproduced Results of IQ-Learn — Environment Steps to reach maximum reward (PIRO vs. LS-IQ vs. HyPE)_
>
> | Environment   | IQ in our paper  | IQ in LSIQ paper  |IQ in HyPE paper|
> |---------------|--------|------------|----------|
> | Ant           | 102K   | 250K    |1M|
> | Hopper        | 670K   |250k   |500k|
> | Humanoid      | 151K  |  500K    |250k|
> | Walker2d      | 780K   |  750K   |750k|

---

> ### Author Response · Authors · 2025-08-06
> **Additional Experiments Using Expert Demonstrations from Orsini et al.**
>
> We have finished new experiments using the expert demonstrations from Orsini et al., and we indeed reproduced the strong performance of CSIL under this setting. CSIL converges quickly and achieves quite high returns which shows impressive online sample efficiency.
>
> However, as you also observed, CSIL may show larger fluctuations during training, while PIRO maintains a much smoother learning curve. This highlights a key strength of our approach: training stability.
>
> Re: convergence speed. PIRO does not use behavioral cloning (BC) pretraining, which may explain why it does not perform as well as CSIL during the initial training phase.
>
> To summarize, while CSIL is more sample-efficient in online sampling, PIRO achieves comparable or superior final performance and training stability across most environments (Hopper, Halfcheetah and Ant). Based on these results, we believe PIRO is SOTA in terms of stability and final reward performance, though not in online sample efficiency (which is not our intention when developing the theory).
>
>
> **PIRO vs CSIL Performance Comparison (Normalized)**
>
> ## HalfCheetah (expert reward：8770)
>
> | Environment Steps (K) | PIRO Return | PIRO Norm. | CSIL Return | CSIL Norm. |
> |----------------------|-------------|------------|-------------|------------|
> | 50 | 360 | 0.07 | 2731 | 0.33 |
> | 100 | 436 | 0.08 | 4940 | 0.58 |
> | 150 | 670 | 0.11 | 6570 | 0.76 |
> | 200 | 1575 | 0.21 | 7994 | 0.91 |
> | 250 | 1833 | 0.23 | 8254 | 0.94 |
> | 500 | 3205 | 0.39 | - | - |
> | 1000 | 6021 | 0.70 | - | - |
> | 1500 | 9411 | 1.07 | - | - |
> | 2000 | 10536 | 1.20 | - | - |
>
>
> ## Hopper (expert reward：2798)
>
> | Environment Steps (K) | PIRO Return | PIRO Norm. | CSIL Return | CSIL Norm. |
> |----------------------|-------------|------------|-------------|------------|
> | 50 | 991 | 0.35 | 2078 | 0.74 |
> | 100 | 1003 | 0.35 | 3313 | 1.19 |
> | 150 | 1008 | 0.36 | 2115 | 0.75 |
> | 200 | 1069 | 0.38 | 3335 | 1.19 |
> | 250 | 1233 | 0.44 | 3341 | 1.20 |
> | 500 | 1751 | 0.62 | - | - |
> |750|3230|1.15|-|-|
> | 1000 | 3353 | 1.20 | - | - |
>
>
> ## Walker2d (expert reward：4118)
>
> | Environment Steps (K) | PIRO Return | PIRO Norm. | CSIL Return | CSIL Norm. |
> |----------------------|-------------|------------|-------------|------------|
> | 50 | -26 | -0.01 | 683 | 0.17 |
> | 100 | 1009 | 0.24 | 259 | 0.06 |
> | 150 | 1004 | 0.24 | 4014 | 0.97 |
> | 200 | 969 | 0.23 | 4064 | 0.99 |
> | 250 | 978 | 0.24 | 4125 | 1.00 |
> | 500 | 997 | 0.24 | - | - |
> | 1000 | 1032 | 0.25 | - | - |
> | 1500 | 2797 | 0.68 | - | - |
> | 2000 | 3316 | 0.81 | - | - |
>
>
> ## Ant (expert reward：5637)
>
> | Environment Steps (K) | PIRO Return | PIRO Norm. | CSIL Return | CSIL Norm. |
> |----------------------|-------------|------------|-------------|------------|
> | 50 | 650.17|  0.12 | 3616 | 0.65 |
> | 100 | 684.82 | 0.12  | 4244 | 0.76 |
> | 150 | 711.69| 0.12 | 5373 | 0.95 |
> | 200 | 735.04 | 0.13 | 5286 | 0.94 |
> | 250 | 709.36 | 0.13 | 5238 | 0.93 |
> | 500 | 760.87 | 0.13 | - | - |
> | 1000 | 3322 | 0.59 | - | - |
> | 1500 | 4714 | 0.84 | - | - |
> | 2000 | 5549 | 0.98 | - | - |
>
>
> ## Final Performance Summary
>
> | Environment | PIRO Final | PIRO Norm. | CSIL Final | CSIL Norm. | Expert Reward |
> |-------------|------------|------------|------------|------------|--------------|
> | HalfCheetah | **10536** | 1.20 | 8254 | 0.94 | 8770 |
> | Hopper | **3353**| 1.20 | 3341 | 1.20 | 2798 |
> | Walker2d | 3316 | 0.81 | **4125** | 1.00 | 4118 |
> | Ant | **5549** | 0.98 | 5238 | 0.93 | 5637 |

---

> > ### Comment · Reviewer_zT3i · 2025-08-07
> >
> > > PIRO achieves comparable or superior final performance and training stability across most environments (Hopper, Halfcheetah and Ant). Based on these results, we believe PIRO is SOTA in terms of stability and final reward performance, though not in online sample efficiency (which is not our intention when developing the theory).
> >
> > I ran a single seed purely to illustrate that the open-sourced repository of the CSIL (from Google DeepMind, not the first author) was in fact reproducible, contrary to the author's claim. It is not correct to make any claims from a single seed, especially on final reward performance when PIRO is trained for four times longer!
> > However, it does suggest that IQLearn, P2IL and CSIL do achieve 4x the sample efficiency of PIRO that I posited in my official review. The reason why this is important is that we see in your results above that PIRO essentially plateaus during the steps in Hopper, Walker and Ant takes CSIL to converge.
> >
> > I believe this discussion has run its course. I commend the authors for their effort during the rebuttal period, and I believe the paper can be greatly improved for the next conference once the experiments section correctly reflects SOTA performance.

---

> ### Author Response · Authors · 2025-08-07
>
> Thanks for comments.
>
> Throughout, we make no claim that our contribution is on "online sample efficiency" but the theoretical and practical stability.
>
> And we believe one should not evaluate a method solely based on the SOTA performace on all aspects (accuracy, efficiency & stability) but also based on the methodological innovation. Critically, our method can beat CSIL, IQ and other online baslines on final reward peformance (accuracy).
>
> We have to wrap up the discussion here as the deadline of author-reviewer discussion approaches. *Thanks for your enthusiastic participation again. Your suggestions are of no doubt invaluable to us.*

---

> ### Author Response · Authors · 2025-08-07
> **Final Statement**
>
> In fact, we hypothesize that **”accuracy — online efficiency— stability”** is a **hard-to-achieve or even impossible triangle** for online IRL methods. **This work prioritizes the edge between accuracy and stability in this triangle while keeps online efficiency at an acceptable level.**
>
>                                         stability
>                                            / \
>              our prioritization   <----   /   \
>                                          /     \
>                          reward accuracy ------- online efficiency

---

### Note · Authors · 2025-08-12

We sincerely thank the reviewers and AC for their thoughtful engagement and constructive feedback. Below, we summarize our key contributions and provide a high-level response to the reviewers' comments.

---

# Theoretical Contribution
For years, PPO -- rooted in TRPO's theory of monotonic policy improvement -- has been (one of) the most successful RL algorithm(s), particularly in LLM alignment. This work is motivated by the **dualism:** the mathematical beauty of TRPO should **not** exist in **isolation**, but in **conjugation** with its inverse problem space. We identify and formalize this inverse counterpart, completing the **"bottom half"** of the **"symmetric picture"** below. We believe this contribution advances RL theory and opens new avenues for designing robust IRL algorithms and enabling stable reward design in the frontier of AI applications.


                     simplify              enable
             TRPO ============> PPO ~~~~~~~~~~~~~~~~~~> Stable Policy +- Robotics
           /   ↓                 ↓                      Learning in   |- Embodied AI
          /  theoretical     practical                                |- Auto Driving
    inverse  monotonicity                                             |- Game AI
          \  based on MM     stability                                |- Recommendation
           \   ↑                 ↑                      Stable Reward |- RLHF
              PRU ============> PIRO ~~~~~~~~~~~~~~~~~> Learning in   +- etc.
                     simplify         open possibility

---

# Practical Contribution

PIRO achieves a three-way balance among *Stability*, *Performance*, and (online) *Efficiency*. In fact,  stability and efficiency are inherently adversarial: stability demands more conservative updates, which increases online samples. This mirrors PPO’s tradeoff: it avoids performance degradation, but it can be a little slower than vanilla policy gradient methods like REINFORCE in certain scenarios.

                                      (first)
                                     Stability
                                       /   \
                                      /     \
                                     /       \
                                    / balance \
                                   /           \
                          Performance ------- Efficiency
                          (new SOTA)        (slightly < SOTA)

---

### Decision · Program_Chairs · 2025-09-17

**Decision:**

Accept (poster)

**Comment:**

This is a nice IRL paper with some theoretical motivation. The reviewers generally have a positive consensus about the theoretical and practical contributions. The negative review is because there is an apparent difference in baseline performance. I personally am satisfied with the authors' response, and if they did use the same data for all methods, their not replicating exactly previous results is not an issue.